# TOWARDS UNDERSTANDING FEATURE LEARNING IN PARAMETER TRANSFER

## ABSTRACT

Parameter transfer is a central paradigm in transfer learning, enabling knowledge reuse across tasks and domains by sharing model parameters between upstream and downstream models. However, when only a subset of parameters from the upstream model is transferred to the downstream model, there remains a lack of theoretical understanding of the conditions under which such partial parameter reuse is beneficial and of the factors that govern its effectiveness. To address this gap, we analyze a setting in which both the upstream and downstream models are ReLU convolutional neural networks (CNNs). Within this theoretical framework, we characterize how the inherited parameters act as carriers of universal knowledge and identify key factors that amplify their beneficial impact on the target task. Furthermore, our analysis provides insight into why, in certain cases, transferring parameters can lead to lower test accuracy on the target task than training a new model from scratch. To our best knowledge, our theory is the first to provide a dynamic analysis for parameter transfer and also the first to prove the existence of negative transfer theoretically. Numerical experiments and real-world data experiments are conducted to empirically validate our theoretical findings.

## 1 INTRODUCTION

Transfer learning has become the workhorse of modern deep learning, because it breaks the traditional curse of having to train a gigantic model from scratch for every new problem (Pan and Yang, 2009; Dai et al., 2009; Torrey and Shavlik, 2010; Imani et al., 2021). By reusing knowledge acquired in a source domain, practitioners can reach higher accuracy with orders-of-magnitude less labeled data and compute (Yosinski et al., 2014; Ruder et al., 2019). The dominant instantiation of this idea is the pre-train–fine-tune pipeline: an upstream model is first optimized on a large-scale, often self-supervised task and is subsequently adapted to a downstream objective (Devlin et al., 2019; Radford et al., 2021; He et al., 2020). Yet the real world seldom offers a perfect one-to-one architectural match between the two stages (Zhuang et al., 2020). Upstream backbones may be deeper, include modality-specific components, or be released as black-box feature extractors (Jiang et al., 2022), while downstream tasks can impose new input resolutions, output spaces, memory budgets, or even deployment hardware that forbid a literal copy of every weight (Bommasani et al., 2021). Parameter transfer emerges as an elegant remedy to this mismatch. Because it requires no raw data from the upstream domain and places almost no constraints on network topology, it combines the sample efficiency of transfer learning with the flexibility of modular design, fueling its rapid adoption across vision, speech, language, and multi-modal applications (Houlsby et al., 2019; Liu et al., 2022).

Despite these advances, existing theoretical studies have focused on static generalization bounds (Maurer et al., 2016; Kumagai, 2016; Wu et al., 2024), without addressing how transfer learning evolves during the training dynamics. Such a dynamic perspective is essential, since transfer is not only about the final generalization guarantee but also about the trajectory through which knowledge is acquired and reused across tasks. Parameter transfer is intrinsically a question of network dynamics. In particular, while empirical works have repeatedly reported the phenomenon of negative transfer (Wang et al., 2019; Zhang et al., 2022; Zu et al., 2025), a rigorous theoretical characterization has been missing. Our work fills this gap: we provide, to the best of our knowledge, the first theoretical analysis of training dynamics in parameter transfer. Importantly, our framework not only proves when and why transfer is beneficial, but also reveals, for the first time in theory, the precise conditions under which negative transfer arises. These findings significantly broaden the theoretical landscape of transfer learning and underscore the necessity of dynamic analysis for the principled design of parameter transfer.

More specifically, we aim to address two fundamental questions: (i) why parameter transfer can enhance test performance compared to random initialization, and (ii) why naive transfer learning may sometimes fail or even lead to negative transfer. In this paper, we conduct a theoretical analysis of parameter transfer within a nonlinear dynamical system (Huang et al., 2024; Zhang et al., 2025) where both the upstream model and the downstream model are two layer neural networks. We explicitly model the universal knowledge (also known as meta-knowledge) and the task-specific knowledge between the source task and the target task. It is assumed that an $\alpha$-proportion of the upstream model's weights are inherited by the downstream model. For the downstream model, the remaining weights are randomly initialized. To our best knowledge, we are the first one to give the training dynamics of parameter transfer and prove the existence of negative transfer in mathematics. Based on the above modeling, we analyze the roles of the three crucial factors: (1) the universal knowledge between the source task and the target tasks; (2) the training sample size for the upstream model; (3) the noise level in the source task. It shows that more inherited parameters, larger training sample size for the upstream model, and less noise in the upstream task can improve the performance of the downstream model. The results are consistent with the empirical performance of parameter transfer, providing theoretical support for its effectiveness. The contributions of our paper are as follows.

- To our best knowledge, this work is the first to give the training dynamics of parameter transfer. Specifically, we prove that when the training sample size, signal strength, noise level, and dimension of both the upstream and downstream models satisfy a certain condition, the test error rate approaches the Bayes optimal. The condition is tight. In opposite of this condition, we prove that the test error remains a constant away from the Bayes optimal. These results together demonstrate the sharpness of our theory and provide a rigorous explanation for the empirical success of parameter transfer.

- We provide theoretical explanation when parameter transfer outperforms direct training from random initialization. Specifically, we identify the critical roles of three factors in determining its effectiveness: the norm of the universal knowledge between the source task and target tasks, the sample size of the source task, and the noise level present in the source data. Our analysis reveals how these factors jointly influence the success of parameter transfer. In particular, we show that parameter transfer allows the downstream model to inherit universal knowledge of guaranteed strength, thereby improving generalization and mitigating the effect of noise memorization in the target tasks. These results offer a rigorous characterization of the advantage of inherited parameters over random initialization and provide practical guidance for their application.

- Our theoretical framework also sheds light on why parameter transfer can sometimes lead to a degradation in test accuracy compared to direct training. Recent studies have reported such phenomena (Zhang et al., 2022; Go et al., 2023; Zu et al., 2025), but the underlying mechanisms remain theoretically underexplored. In this work, we theoretically proved the existence of the negative transfer. Particularly, when the shared signal between the source and target tasks is very weak, even a well-trained upstream model with a large sample size or low noise level can harm the target task. The key mechanism is that the weight norm learned from the upstream model becomes excessively large. When transferred, these over-amplified weights fail to enhance the weak shared signal in the target task but instead magnify task-specific noise, hence degrading test performance. Our results thus offer rigorous theoretical guidance for the effective application of the parameter transfer methodology: parameter transfer should be designed to extract and transfer strong shared features, which necessitates careful selection of the source dataset to ensure sufficient relevance and signal quality.

## 2 RELATED WORK

**Transfer Learning Theory:** Transfer learning has long been the subject of rigorous theoretical scrutiny. The seminal bias-learning framework introduced by Baxter (2000) first quantified the benefits of a shared inductive bias across tasks. Later works refined this picture, establishing finite-sample guarantees for representation-based transfer (Maurer et al., 2016), information-theoretic upper bounds on the joint risk (Wang, 2018; Wu et al., 2024), and minimax-optimal sample-complexity characterisations in linear regimes (Tripuraneni et al., 2020). Yi et al. (2023) proves that conditional independence from spurious attributes given the label is sufficient for OOD robustness under correlation shift, and introduces the Conditional Spurious Variation (CSV) metric that directly controls the OOD generalization error. Besides, existing theoretical work on parameter transfer is quite limited, Kumagai (2016) assumes the parameter-transfer learnability of the parametric feature mapping and provides static generalization bounds without consideration of optimization for parameter transfer. Hu and Zhang (2023) assumes that different models may share common knowledge in their parameters and prove that transferring parameters via model averaging can improve the prediction performance of the target model. For discussion on transfer learning application, please refer to Section G.

**Neural Tangent Kernel and Feature Learning:** With the advancement of deep learning, analyzing the dynamics underlying neural networks has become increasingly meaningful. Jacot et al. (2018) introduce the Neural Tangent Kernel (NTK) regime, which effectively characterizes the dynamics of sufficiently over-parameterized neural networks and explains how they fit data during training. Building on this, Cao and Gu (2019; 2020) further investigated the generalization capabilities of neural networks in the over-parameterized regime. At the core of these studies is the observation that, under sufficiently over-parameterization, neural network weights can be well-approximated by a linear system (Yu et al., 2023; Benjamin et al., 2024; Fu and Wang, 2024) and remain close to their initialization throughout training. This phenomenon is known as lazy training (Chizat et al., 2019; Ghorbani et al., 2019; Zhu et al., 2023), which cannot explain the superior performance of neural networks well. Besides NTK regime, another line of studies explores benign overfitting in neural network, which is called feature learning (Zou et al., 2023; Cao et al., 2022; Meng et al., 2025). Feature learning theory typically assumes a specific data generation model and estimates how the weights learn the signals and noise present in the data. Feature learning theory differs from NTK in two key aspects: 1) Feature learning theory employs small initializations, which allow the learning process to dominate and avoid lazy training. 2) Feature learning system can be a highly nonlinear system, and its dynamics are closer to those of real neural networks. For example, Allen-Zhu and Li (2023) characterizes ensemble learning and knowledge distillation. Meng et al. (2024) investigates that CNNs can learn XOR problem efficiently. Shang et al. (2024) investigate the two layer neural networks and discover that the initialization of second layers matters in the generalization.

## 3 PROBLEM SETTING

**Notations.** For sequences $\{x_n\}$ and $\{y_n\}$, the relation $x_n = O(y_n)$ indicates the existence of absolute constants $C_1 > 0$ and $N > 0$ such that $|x_n| \leq C_1|y_n|$ holds uniformly for all $n \geq N$. Similarly, we write $x_n = \Omega(y_n)$ if $y_n = O(x_n)$, and we denote $x_n = \Theta(y_n)$ when both $x_n = O(y_n)$ and $x_n = \Omega(y_n)$ hold. We adopt $\widetilde{O}(\cdot)$, $\widetilde{\Omega}(\cdot)$, and $\widetilde{\Theta}(\cdot)$ to hide some logarithmic terms. For any event $\mathcal{E}$, we denote its indicator function by $\mathbf{1}(\mathcal{E})$, which equals 1 if $\mathcal{E}$ occurs and 0 otherwise. Furthermore, for non-negative quantities $x_1, \ldots, x_k$, we use the shorthand $y = \text{poly}(x_1, \ldots, x_k)$ to express that $y$ is bounded above by a positive power of $\max\{x_1, \ldots, x_k\}$, i.e., $y = O(\max\{x_1, \ldots, x_k\}^D)$ for some constant $D > 0$. $y = \text{polylog}(x)$ indicates that y grows polynomially with respect to $\log x$.

Then, we introduce the data generation model, the network model we adapt and the algorithm of parameter transfer. Let $\mathbf{u}, \mathbf{v}_1, \mathbf{v}_2 \in \mathbb{R}^d$ be three fixed signal vectors with $\mathbf{u} \perp \mathbf{v}_1$ and $\mathbf{u} \perp \mathbf{v}_2$. The data is given in the following definition.

**Definition 3.1** (Data in Task 1). *Each data point $(\mathbf{x}, y)$ with $\mathbf{x} = (\mathbf{x}^{(1)\top}, \mathbf{x}^{(2)\top})^\top \in \mathbb{R}^{2d}$ is generated from the following distribution $\mathcal{D}_1$: 1. The data label $y \in \{\pm 1\}$ is generated as a Rademacher random variable. 2. A noise vector $\boldsymbol{\xi}$ is generated from the Gaussian distribution $\mathcal{N}(\mathbf{0}, \sigma_{p,1}^2(\mathbf{I} - \mathbf{u}\mathbf{u}^\top/\|\mathbf{u}\|_2^2 - \mathbf{v}_1\mathbf{v}_1^\top/\|\mathbf{v}_1\|_2^2))$. 3. One of $\mathbf{x}^{(1)}$, $\mathbf{x}^{(2)}$ is randomly selected and assigned as $y \cdot (\mathbf{u} + \mathbf{v}_1)$ which is the signal part, and the other is assigned as $\boldsymbol{\xi}$ which is the noise part.*

**Definition 3.2** (Data in Task 2). *Each data point $(\mathbf{x}, y)$ with $\mathbf{x} = (\mathbf{x}^{(1)\top}, \mathbf{x}^{(2)\top})^\top \in \mathbb{R}^{2d}$ is generated from the following distribution $\mathcal{D}_2$: 1. The data label $y \in \{\pm 1\}$ is generated as a Rademacher random variable. 2. A noise vector $\boldsymbol{\xi}$ is generated from the Gaussian distribution $\mathcal{N}(\mathbf{0}, \sigma_{p,2}^2(\mathbf{I} - \mathbf{u}\mathbf{u}^\top/\|\mathbf{u}\|_2^2 - \mathbf{v}_2\mathbf{v}_2^\top/\|\mathbf{v}_2\|_2^2))$. 3. One of $\mathbf{x}^{(1)}$, $\mathbf{x}^{(2)}$ is randomly selected and assigned as $y \cdot (\mathbf{u} + \mathbf{v}_2)$ which is the signal part, and the other is assigned as $\boldsymbol{\xi}$ which is the noise part.*

We divide the data input into the signal and noise patch. Such data generation model has been widely used (Allen-Zhu and Li, 2023; Cao et al., 2022; Jelassi and Li, 2022; Kou et al., 2023; Meng et al., 2024). For the signal patch, the datasets in Task 1 and Task 2 share a universal signal vector denoted by $\mathbf{u}$, while also containing task-specific signal vectors $\mathbf{v}_1$ and $\mathbf{v}_2$ respectively. For the noise patch, we assume that it is orthogonal to the signal patch for simplicity. Although this orthogonality assumption simplifies the analysis, it can be naturally extended to more general cases where the noise may have a non-trivial correlation with the signal part. We show later that the universal knowledge is crucial for parameter transfer. In addition, the noise variances in Task 1 and Task 2 are $\sigma_{p,1}$ and $\sigma_{p,2}$; the sample sizes for Task 1 and Task 2 are $N_1$ and $N_2$; the data samples for Task 1 is denoted by $\{\mathbf{x}_{i,1}, y_{i,1}\}_{i=1}^{N_1}$ and the data samples for Task 2 is denoted by $\{\mathbf{x}_{i,2}, y_{i,2}\}_{i=1}^{N_2}$.

We consider adapt two-layer convolutional neural networks (CNN) for both the upstream model and the downstream model. The CNN filters are applied to both the signal part and the noise part. Specifically, the network is defined as

$$f(\mathbf{W}; \mathbf{x}) = F_{+1}(\mathbf{W}; \mathbf{x}) - F_{-1}(\mathbf{W}; \mathbf{x}), \ F_j(\mathbf{W}; \mathbf{x}) = \frac{1}{m}\sum_{r=1}^{m}[\sigma(\langle \mathbf{w}_{j,r}, \mathbf{x}^{(1)} \rangle) + \sigma(\langle \mathbf{w}_{j,r}, \mathbf{x}^{(2)} \rangle)].$$

---

**Algorithm 1:** Algorithm of Parameter Transfer.

---

**Input:** Data on Task 1 $\{\mathbf{x}_{i,1}, y_{i,1}\}_{i=1}^{N_1}$ and data on Task 2 $\{\mathbf{x}_{i,2}, y_{i,2}\}_{i=1}^{N_2}$. The upstream model $f^A$ and the downstream model $f^D$. The ratio of inherited parameters $\alpha$.

1 Initialize $f^A$: $\mathbf{w}_{j,r}^{A,(0)} \sim N(\mathbf{0}, \sigma_0^2), j \in \{+1, -1\}, r \in [m]$;
2 **for** $t \leq T^*$ **do**
3  Update $\mathbf{w}_{j,r}^{A,(t)}$ as: $\mathbf{w}_{j,r}^{A,(t+1)} = \mathbf{w}_{j,r}^{A,(t)} - \eta \nabla_{\mathbf{w}_{j,r}^A} L_{Task1}(\mathbf{W}^{A,(t)}); t = t + 1$;
4 **end**
5 Initialize $f^D$: $\mathbf{w}_{j,r}^{D,(0)} = \mathbf{w}_{j,r}^{A,(T^*)}$ if $1 \leq r \leq \alpha m$, and $\mathbf{w}_{j,r}^{D,(0)} \sim N(\mathbf{0}, \sigma_0^2)$ if $\alpha m < r \leq m$.
6 **for** $t \leq T^*$ **do**
7  Update $\mathbf{w}_{j,r}^{D,(t)}$ as: $\mathbf{w}_{j,r}^{D,(t+1)} = \mathbf{w}_{j,r}^{D,(t)} - \eta \nabla_{\mathbf{w}_{j,r}^D} L_{Task1}(\mathbf{W}^{D,(t)}); t = t + 1$;
8 **end**

---

**Algorithm 2:** Standard training.

---

**Input:** Data on Task 2 $\{\mathbf{x}_{i,2}, y_{i,2}\}_{i=1}^{N_2}$. The downstream model $f^D$.

1 Initialize $f^D$: $\mathbf{w}_{j,r}^{D,(0)} \sim N(\mathbf{0}, \sigma_0^2), j \in \{+1, -1\}, r \in [m]$;
2 **for** $t \leq T^*$ **do**
3  Update $\mathbf{w}_{j,r}^{D,(t)}$ as: $\mathbf{w}_{j,r}^{D,(t+1)} = \mathbf{w}_{j,r}^{D,(t)} - \eta \nabla_{\mathbf{w}_{j,r}^D} L_{Task1}(\mathbf{W}^{D,(t)}); t = t + 1$;
4 **end**

---

Here, $m$ is the number of convolutional filters, and $\sigma(z) = \max\{0, z\}$ is the activation function. Moreover, $\mathbf{w}_{j,r}$ denotes the weight for $r$-th filter, $\mathbf{W}_j$ is the weight matrices associated with $F_j$, and $\mathbf{W}$ collects all the weight matrices $\mathbf{w}_{j,r}$ for $j \in \{\pm 1\}$. Such convolutional neural network is widely used in feature learning theory. Then, define the cross-entropy loss function $\ell(z) = \log(1 + \exp(-z))$, the training loss for Task 1 and Task 2 can be written as

$$L_{Task1}(\mathbf{W}) = \frac{1}{N_1} \sum_{i \in [N_1]} \ell(y_{i,1} f(\mathbf{W}; \mathbf{x}_{i,1})); \quad L_{Task2}(\mathbf{W}) = \frac{1}{N_2} \sum_{i \in [N_2]} \ell(y_{i,2} f(\mathbf{W}; \mathbf{x}_{i,2})).$$

With a well-defined training objective, we present the parameter transfer training procedure in Algorithm 1, alongside the standard training baseline in Algorithm 2. The parameter transfer algorithm used in this work randomly sample weights from the upstream model. In contrast, most existing methods are typically designed to extract and transfer strong shared features. In addition, it is worth noting that in the upstream model, practitioners often leverage larger datasets and more complex model architectures to extract transferable knowledge. Such pretraining processes may incur substantial computational costs, sometimes exceeding the capacity of local computing resources. Furthermore, as we will discuss in the following section, transferring parameters from the upstream model to the downstream task is not universally beneficial. In some stringent scenarios, inappropriate inheritance of parameters can even degrade the test performance of the downstream model, which is also reported in literature.

## 4 MAIN RESULTS

In this section, we present our main results. Our main results aim to show the theoretical guarantees with probability at least $1 - \delta$ for some small $\delta > 0$. With such probability, we show that the training loss will converge below some arbitrarily small $\varepsilon > 0$, while the test accuracy can have different performance based on the training sample size $N_1$, $N_2$, the dimension $d$ and the inherited parameters $\alpha$ etc. We define $T^* = \eta^{-1}\text{poly}(n, d, \varepsilon, m)$ be the maximum admissible number of training iterations. To establish the results, we require several conditions that are summarized below.

**Condition 4.1.** *Define* $n = \max\{N_1, N_2\}$. *Suppose there exists a sufficiently large constant* $C$, *such that the following hold with* $\mathbf{v} = \mathbf{v}_1$ *or* $\mathbf{v}_2$, *and* $\sigma_p = \sigma_{p,1}$ *or* $\sigma_{p,2}$:

*1. Dimension $d$ satisfies:* $d = \widetilde{\Omega}(\max\{n\sigma_p^{-2}\|\mathbf{u} + \mathbf{v}\|_2^2, n^2\})$.
*2. Training sample size $n$ and neural network width satisfy:* $m \geq C \log(n/\delta)$, $n \geq C \log(m/\delta)$.

3. The norm of the signal satisfies $\|\mathbf{u} + \mathbf{v}\|_2^2 = \Omega(\sigma_p^2 \log(n/\delta))$.
4. The standard deviation of Gaussian initialization $\sigma_0$ is appropriately chosen such that

$$\sigma_0 = O\bigg( \Big( \max \big\{ \sigma_p d/\sqrt{n}, \sqrt{\log(m/\delta)} \cdot \|\mathbf{u} + \mathbf{v}\|_2 \big\} \Big)^{-1} \bigg).$$

5. The learning rate $\eta$ satisfies

$$\eta \leq O\bigg( \Big( \max \big\{ \sigma_p^2 d^{3/2}/(n^2 m \sqrt{\log(m/\delta)}), \sigma_p^2 d/n, \|\mathbf{u} + \mathbf{v}\|_2^2/m \big\} \Big)^{-1} \bigg).$$

The first two conditions on $d$, $n$, and $m$ are imposed to ensure the desired concentration results hold, accounting for randomness in both the data distribution and random initialization. The assumption on the width $d$ ensures that the learning dynamics operate in the over-parameterized regime. Similar assumptions have been adopted in a series of recent works (Allen-Zhu and Li, 2023; Cao et al., 2022; Kou et al., 2023; Meng et al., 2024). The condition on the initialization scale $\sigma_0$ requires it to be sufficiently small, so that the impact of initialization on training remains negligible. This allows the learning dynamics to dominate the training process, moving beyond the Neural Tangent Kernel (NTK) regime. Finally, the smallness condition on the learning rate $\eta$ is a standard technical assumption, ensuring the stability of the analysis. Under Condition 4.1, we have the following theorem.

**Theorem 4.2** (With parameter transfer). *Suppose that percentage $\alpha$ $(0 < \alpha \leq 1)$ of the upstream model's weights are inherited. For any $\varepsilon, \delta > 0$, if Condition 4.1 holds, then there exist constants $C_1, C_2, C_3 > 0$, such that with probability at least $1 - 2\delta$, the following results hold at $T = \Omega(N_2 m/(\eta \varepsilon \sigma_{p,2}^2))$:*

1. *The training loss is below $\varepsilon$: $L_S(\mathbf{W}^{(t)}) \leq \varepsilon$.*
2. *If $d \leq C_1(\frac{\alpha^2 N_1^2 \|\mathbf{u}\|_2^4}{\sigma_{p,1}^4} + \frac{N_2^2 \|\mathbf{u}+\mathbf{v}_2\|_2^4}{\sigma_{p,2}^4})/(\frac{\alpha^2 \sigma_{p,2}^2 N_1}{\sigma_{p,1}^2} + N_2)$, the test error is close to the optimum. For any new data $(\mathbf{x}, y)$*

$$\mathbb{P}(yf(\mathbf{W}^{(t)}; \mathbf{x}) < 0) \leq \exp\Big[ -C_2 \big(\frac{\alpha^2 N_1^2 \|\mathbf{u}\|_2^4}{\sigma_{p,1}^4} + \frac{N_2^2 \|\mathbf{u}+\mathbf{v}_2\|_2^4}{\sigma_{p,2}^4}\big)/\big(\frac{\alpha^2 \sigma_{p,2}^2 N_1 d}{\sigma_{p,1}^2} + N_2 d\big) \Big];$$

3. *If $d \geq C_3(\frac{\alpha^2 N_1^2 \|\mathbf{u}\|_2^4}{\sigma_{p,1}^4} + \frac{N_2^2 \|\mathbf{u}+\mathbf{v}_2\|_2^4}{\sigma_{p,2}^4})/(\frac{\alpha^2 \sigma_{p,2}^2 N_1}{\sigma_{p,1}^2} + N_2)$, the test error has a gap from the optimum: $\mathbb{P}(yf(\mathbf{W}^{(t)}; \mathbf{x}) < 0) \geq 0.1$.*

Theorem 4.2 reveals a phase transition of the generalization performance. It highlights the critical role of universal knowledge in parameter transfer, as well as the influence of inherited parameters, the sample size of the source task, and the signal-to-noise ratio. Specifically, the theorem shows that in the upstream model, generalization performance improves when the sample size of the source task, the amount of inherited parameters, and the strength of universal knowledge are sufficiently large, and when the noise level in the upstream model is small. Conversely, in the absence of universal knowledge, inherited parameters, or with a small sample size, such benefits do not emerge, regardless of other factors.

**Theorem 4.3** (Without parameter transfer, Previous results in Kou et al. (2023)). *For any $\varepsilon, \delta > 0$, if Condition 3.1 holds, then there exist constants $C_1', C_2', C_3' > 0$, such that with probability at least $1 - 2\delta$, the following results hold at $T = \Omega(N_2 m/(\eta \varepsilon \sigma_{p,2}^2))$:*

1. *The training loss converges below $\varepsilon$, i.e., $L(\mathbf{W}^{(T)}) \leq \varepsilon$.*
2. *If $N_2 \|\mathbf{u} + \mathbf{v}_2\|_2^4 \geq C_1' \sigma_{p,2}^4 d$, then the CNN trained by gradient descent can achieve near Bayes-optimal test error: $\mathbb{P}(yf(\mathbf{W}^{(t)}; \mathbf{x}) < 0) \leq \exp\left( -C_2' N_2 \|\mathbf{u} + \mathbf{v}_2\|_2^4/(\sigma_{p,2}^4 d) \right).$*
3. *If $N_2 \|\mathbf{u} + \mathbf{v}_2\|_2^4 \leq C_1' \sigma_{p,2}^4 d$, then the CNN trained by gradient descent can only achieve sub-optimal error rate: $\mathbb{P}(yf(\mathbf{W}^{(t)}; \mathbf{x}) < 0) \geq 0.1$.*

Theorem 4.3 characterizes the generalization performance of networks without parameter transfer. We define the following key quantity $\Gamma = \frac{\alpha^2 N_1 \|\mathbf{u}\|_2^4}{\sigma_{p,1}^2 \sigma_{p,2}^2 d}$. Under Condition 4.1, we observe in the theorem above that large value of $\Gamma$ is a sufficient condition in determining the success of parameter transfer. By comparing the conditions of the two theorems, we can draw the following conclusions.

**Proposition 4.4.** *Under the condition of Theorem 4.2 and 4.3:*

1. *If $\Gamma \geq C$ for some sufficient large $C > 0$, when $d > C'_1(N_2\|\mathbf{u} + \mathbf{v}_2\|_2^4)/(\sigma_{p,2}^4)$, **inherited parameters** improves the performance of downstream models:*

   - *Without **parameter transfer**, the error rate is sub-optimal: $\mathbb{P}(yf(\mathbf{W}^{(t)}; \mathbf{x}) < 0) \geq 0.1$;*
   - *With **parameter transfer**, the error rate is near optimal: $\mathbb{P}(yf(\mathbf{W}^{(t)}; \mathbf{x}) < 0) \leq c$ for $c$ small enough.*

   *When $d < C'_3(N_2\|\mathbf{u} + \mathbf{v}_2\|_2^4)/(\sigma_{p,2}^4)$, using parameter transfer or not both are near optimal error rate.*

2. *When $\frac{\|\mathbf{u}+\mathbf{v}_2\|_2^2}{\|\mathbf{u}\|_2^2} \geq \alpha N_1\sigma_{p,2}^2/(N_2\sigma_{p,1}^2) \geq C_4$ for $C_4$ large enough, which means that the norm of the universal signal is much smaller than that of the task-specific signal, parameter transfer is detrimental to the downstream model, i.e., **negative transfer**.*

For the first term, the value of $\Gamma$ should not be regarded as a necessary condition for determining the failure of parameter transfer. The key reason is that even when $\Gamma$ is small, a sufficiently large sample size $N_2$ or high data quality in Task 2 can still ensure the success of parameter transfer. As shown in Proposition 4.4, when $\Gamma$ is large, parameter transfer will not degrade performance if Task 2 itself achieves good generalization. Conversely, if Task 2 suffers from poor test performance, parameter transfer can leverage its knowledge transfer to improve overall accuracy. For the second term, theoretical analysis reveals that under very stringent conditions, parameter transfer can be detrimental to the performance of downstream models, i.e., negative transfer. The conditions indicate that negative transfer occurs only when the norm of the universal signal is much smaller than that of the task-specific signal.

## 5 PROOF SKETCH

In this section, we give a concise proof outline of Theorem 4.2 and full proof can be found in the appendix. Define the maximum admissible iterations for two training systems as $T^*, T^{**} = \eta^{-1}\text{poly}(n, d, \varepsilon, m)$, where $T^*$ is the maximum training iterations in the upstream model and $T^{**}$ is the maximum training iterations in the downstream model. The CNN filters' training dynamics are analyzed via the decomposition of weights:

$$\mathbf{w}_{j,r}^{(t)} = \mathbf{w}_{j,r}^{(0)} + j \cdot \gamma_{j,r}^{(t)} \cdot \|\mathbf{u}\|_2^{-2} \cdot \mathbf{u} + j \cdot \gamma_{j,r,1}^{(t)} \cdot \|\mathbf{v}_1\|_2^{-2} \cdot \mathbf{v}_1 + j \cdot \gamma_{j,r,2}^{(t)} \cdot \|\mathbf{v}_2\|_2^{-2} \cdot \mathbf{v}_2$$

$$+ \sum_{i=1}^{N_1} \rho_{j,r,i,1}^{(t)} \cdot \|\boldsymbol{\xi}_{i,1}\|_2^{-2} \cdot \boldsymbol{\xi}_{i,1} + \sum_{i=1}^{N_2} \rho_{j,r,i,2}^{(t)} \cdot \|\boldsymbol{\xi}_{i,2}\|_2^{-2} \cdot \boldsymbol{\xi}_{i,2}.$$

Here, $\gamma$ and $\rho$ track signal learning and noise memorization, respectively. The analysis proceeds in two systems (Task 1 and Task 2).

**System 1:** We define $\overline{x}_t^A, \underline{x}_t^A$ as solutions to:

$$\overline{x}_t^A + \overline{b}^A e^{\overline{x}_t^A} = \overline{c}^A t + \overline{b}^A, \quad \underline{x}_t^A + \underline{b}^A e^{\underline{x}_t^A} = \underline{c}^A t + \underline{b}^A,$$

with parameters $\overline{b}^A, \underline{b}^A$, and $\overline{c}^A, \underline{c}^A$ depending on $\eta, \sigma_{p,1}, d, N_1, m$. The key lemma bounds the coefficients:

**Lemma 5.1.** *Under Condition 4.1, it holds that*

$$\frac{\eta\|\mathbf{u}\|_2^2}{\overline{c}m}\overline{x}_{t-2}^A - \frac{2\eta\|\mathbf{u}\|_2^2}{m} \leq \gamma_{j,r}^{A,(t)} \leq \frac{\eta\|\mathbf{u}\|_2^2}{\underline{c}m}\underline{x}_{t-1}^A - \frac{2\eta\|\mathbf{u}\|_2^2}{m},$$

$$\frac{\eta\|\mathbf{v}_1\|_2^2}{\overline{c}m}\overline{x}_{t-2}^A - \frac{2\eta\|\mathbf{v}_1\|_2^2}{m} \leq \gamma_{j,r,1}^{A,(t)} \leq \frac{\eta\|\mathbf{v}_1\|_2^2}{\underline{c}m}\underline{x}_{t-1}^A - \frac{2\eta\|\mathbf{v}_1\|_2^2}{m}.$$

*Moreover, for the noise memorization it holds that*

$$\frac{N_1}{12}(\overline{x}_{t-2}^A - \overline{x}_1^A) \leq \sum_{i \in [N_1]} \overline{\rho}_{j,r,i,1}^{A,(t)} \leq 5N_1\underline{x}_{t-1}^A.$$

These bounds are established via the *balanced loss property* and continuous approximations.

**System 2:** We transfer the analysis by defining $\gamma_{j,r}^{D,(t)} - \gamma_{j,r}^{D,(T^*+1)}$, isolating the effect of new initialization. Define $\overline{x}_t^D, \underline{x}_t^D$ analogously, yielding:

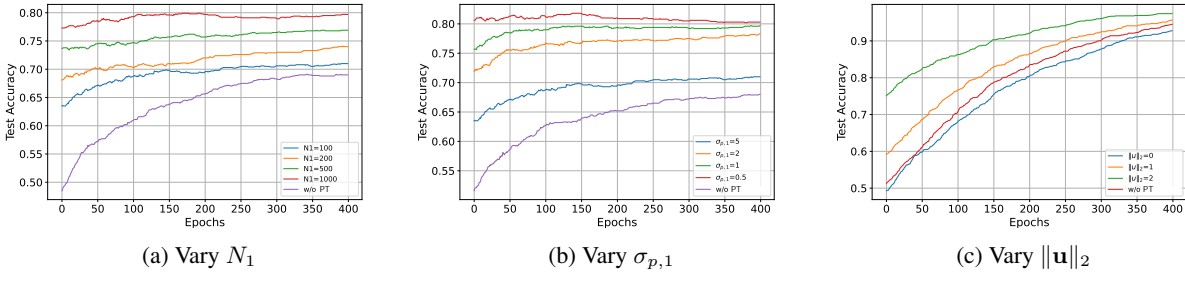

(a) Vary $N_1$        (b) Vary $\sigma_{p,1}$        (c) Vary $\|\mathbf{u}\|_2$

Figure 1: Test accuracy under varying conditions of the source task. "w/o PT" corresponds to standard training without parameter transfer. We compare three key factors that influence the effectiveness of parameter transfer: (a) training sample size of Task 1 $N_1$; (b) the noise level of Task 1; (c) the universal signal strength $\|\mathbf{u}\|_2$ while fixing $\|\mathbf{u} + \mathbf{v}_2\|_2$. All scenarios include a baseline setting without parameter transfer.

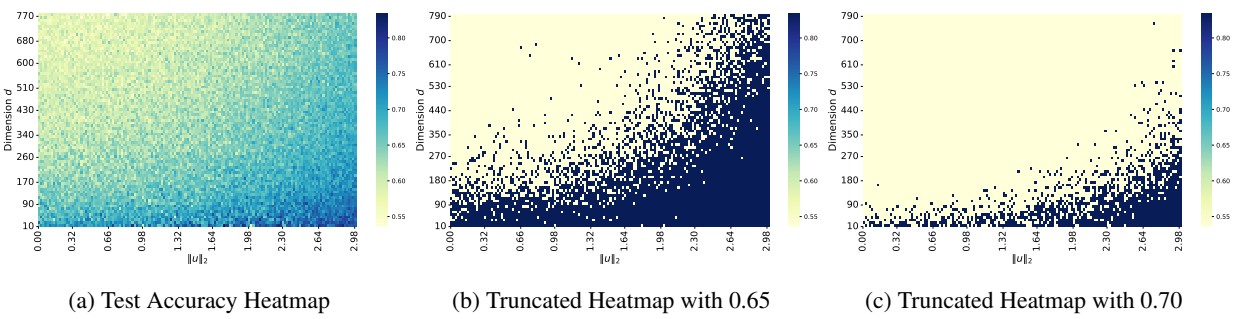

(a) Test Accuracy Heatmap     (b) Truncated Heatmap with 0.65     (c) Truncated Heatmap with 0.70

Figure 2: (a) is the heatmap of test accuracy under different dimensions $d$ and the universal signal strength $\|\mathbf{u}\|_2$ with fixex $\|\mathbf{u} + \mathbf{v}_2\|_2$. The x-axis is the value of $\|\mathbf{u}\|_2$ and the y-axis is the dimension $d$. (b) and (c) display the truncated heatmap of test accuracy. The accuracy smaller than 0.65 (0.70) is set as 0 (yellow) and the other is set as 1 (blue).

**Lemma 5.2.** *Under Condition 4.1, for $T^* + 1 \leq t \leq T^{**}$, it holds that*

$$\frac{\eta\|\mathbf{u}\|_2^2}{\underline{c}^D m}\underline{x}_{t-2}^D - \frac{2\eta\|\mathbf{u}\|_2^2}{m} \leq \gamma_{j,r}^{D,(t)} - \gamma_{j,r}^{D,(T^*+1)} \leq \frac{\eta\|\mathbf{u}\|_2^2}{\bar{c}^D m}\bar{x}_{t-1}^D - \frac{2\eta\|\mathbf{u}\|_2^2}{m},$$

$$\frac{\eta\|\mathbf{v}\|_2^2}{\underline{c}^D m}\underline{x}_{t-2}^D - \frac{2\eta\|\mathbf{v}_2\|_2^2}{m} \leq \gamma_{j,r,2}^{D,(t)} \leq \frac{\eta\|\mathbf{v}\|_2^2}{\bar{c}^D m}\bar{x}_{t-1}^D - \frac{2\eta\|\mathbf{v}_2\|_2^2}{m}.$$

*Moreover, for the noise memorization term, it holds that*

$$\frac{N_2}{12}(\underline{x}_{t-2}^D - \underline{x}_1^D) \leq \sum_{i \in [N_2]} \bar{\rho}_{j,r,i,2}^{D,(t)} \leq 5N_2\bar{x}_{t-1}^D.$$

Finally, test accuracy and training loss are evaluated by comparing inner products $\langle \mathbf{w}_{j,r}^{(t)}, \mathbf{u} + \mathbf{v}_2 \rangle$ and $\langle \mathbf{w}_{j,r}^{(t)}, \boldsymbol{\xi} \rangle$, leveraging the established bounds on $\gamma$ and $\rho$. This yields the desired generalization and convergence guarantees.

## 6 NUMERICAL EXPERIMENTS

In this section, we conduct experiments on the synthesized data. Our experiments choose training sample size $N_1, N_2$, noise level $\sigma_{p,1}, \sigma_{p,2}$, the universal signal strength $\|\mathbf{u}\|_2$. The test sample size is 1000 for all experiments. Given the dimension $d$ and the signal $\mathbf{u}, \mathbf{v}_1, \mathbf{v}_2$, the data in Task 1 and Task 2 is generated according to Definition 3.1 and 3.2. Specifically, We set $d = 2000$ and the signal are constructed via the Gram-Schmidt orthogonalization process to ensure mutual orthogonality in the vector space. Then, we generated the nosie vector $\boldsymbol{\xi}$ from Gaussion distribution.

We adapt the two-layer CNN model defined in section 3 for both upstream model and downstream model. The number of filters is $m = 40$. All models are trained with gradient descent with a learning rate $\eta = 0.01$. For all weights without

Table 1: **Effect of varying $N_1$ on CIFAR-10 and CIFAR-100.** "w/o PT" corresponds to standard training without parameter transfer, while "w/ PT" refers to the proposed parameter transfer methodology.

| | Upstream | Downstream | w/o PT | w/ PT (vary $N_1/N_2$) | | |
|---|---|---|---|---|---|---|
| | | | | 2 | 3 | 4 |
| CIFAR-10 | ResNet-101 | ResNet-34 | 90.80 | 94.20 | 96.90 | 97.20 |
| | | ResNet-50 | 89.25 | 94.25 | 97.25 | 97.85 |
| | VGG-16 | VGG-11 | 82.05 | 91.85 | 94.25 | 96.80 |
| | | VGG-13 | 85.90 | 89.80 | 92.65 | 95.20 |
| CIFAR-100 | ResNet-101 | ResNet-34 | 68.35 | 70.95 | 74.10 | 80.35 |
| | | ResNet-50 | 70.45 | 74.95 | 76.55 | 81.20 |
| | VGG-16 | VGG-11 | 62.05 | 64.30 | 65.65 | 66.60 |
| | | VGG-13 | 63.75 | 64.35 | 65.35 | 65.65 |

using parameter transfer, it is initialized as $N(0, \sigma_0^2)$, where $\sigma_0 = 0.01$. We set the learning rate as 0.01. The upstream models are trained for $T_1 = 800$ epochs while the downstream models are trained for $T_2 = 400$ epochs. Our goal is to explain the effect of parameter transfer under different settings.

1. In the first setting, we fix the noise level $\sigma_{p,1} = \sigma_{p,2} = 5$ and the sample size of the target dataset $N_2 = 100$. Then, we compare the test accuracy under different sample sizes of the target dataset $N_1$ and the results are shown in Figure 1a.
2. In the second setting, we fix $N_1 = N_2 = 100$ and the noise level of Task 2 $\sigma_{p,2} = 5$. Then, we compare the test accuracy under noise level of Task 1 $\sigma_{p,1}$ and the results are shown in Figure 1b.
3. In the third setting, we fix $N_1 = 1000, N_2 = 100$, the noise level of all data $\sigma_{p,1} = \sigma_{p,2} = 15$ and $\|\mathbf{u} + \mathbf{v}_2\|_2 = 3$, Then, we compare the test accuracy under different $\|\mathbf{u}\|_2$ and the results are shown in Figure 1c. Note that it is important to fix $\|\mathbf{u} + \mathbf{v}_2\|_2$ instead of $\|\mathbf{v}_2\|_2 = 3$. Otherwise, the performance improvement may be attributed to a stronger signal rather than parameter transfer.
4. In the fourth setting, we set $N_1 = 1000, N_2 = 100, \sigma_{p,1} = \sigma_{p,2} = 15, \alpha = 0.5$ so that the inherited weights plays a dominant role in Task 2. According to Theorem 4.2, the phase transition happens when $\|\mathbf{u}\|_2$ and $d$ break the balance. We plot the heatmap of test accuracy under different $d$ and $\|\mathbf{u}\|_2$ in Figure 2a. Moreover, the truncated heatmaps are also shown in Figure 2b and 2c.

Figure 1 demonstrates that increasing training sample size for the upstream model, reducing the noise in Task 1, or enhancing the universal knowledge in the signal can all improve the performance of parameter transfer. Especially, in Figure 1c, we find that when $\|\mathbf{u}\|_2 = 0$, parameter transfer lead to a degradation in test accuracy. This implies that there is few universal knowledge in the signal, it may lead to negative transfer, thereby impairing the model's performance on new tasks. As shown in Figure 2, increasing $\|\mathbf{u}\|_2$ or decreasing $d$ will improve the effect of parameter transfer. The universal knowledge in the signal is critical for the success of parameter transfer. These conclusions are intuitive and consistent with our theoretical analysis.

## 7 REAL DATA EXPERIMENTS

In this section, we perform real data experiments to show that parameter transfer is effective and is impacted by several factors: the training sample size of Task 1 and the noise level in Task 1.

**Experiments on Varying $N_1$.** We investigated the impact of the training sample size of Task 1 on the efficacy of the inherited parameters. Specifically, We randomly select 2 classes from CIFAR-10 (or 20 classes from CIFAR-100) as Task 2, and then randomly choose $k$ classes from the remaining categories as Task 1. For example, when $N_1/N_2 = 3$, we select 6 (or 60) classes from CIFAR-10 (or CIFAR-100) as Task 1. We use ResNet-101 as the upstream model and use ResNet-34 and ResNet-50 as the downstream models. As presented in Tab. 1, the results indicate that as the number of samples in Task 1 increases, parameter transfer demonstrates progressively greater performance improvements relative to a from-scratch training baseline. For example, employing a ResNet-101 upstream model and a ResNet-34 downstream model on CIFAR-100, the performance increment due is 2.6% when the source tasks are 40 classes. This increment rise to 12% when the source tasks are 80 classes.

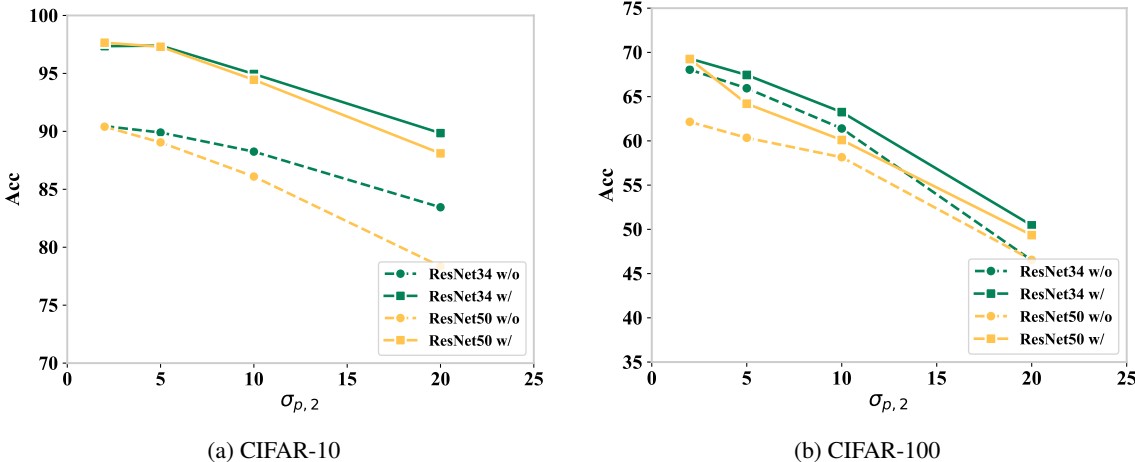

(a) CIFAR-10            (b) CIFAR-100

Figure 3: **Effect of varying $\sigma_{p,2}$ on CIFAR-10 and CIFAR-100.** Test accuracy of ResNet-34 and ResNet-50 as downstream models on (a) CIFAR-10 and (b) CIFAR-100 under different noise level $\sigma_{p,2}$. "w/" and "w/o" denote models trained with and without parameter transfer, respectively.

**Experiments on Varying $\sigma_{p,2}$.** Furthermore, we explore the effect of different proportions of added noise on the target tasks. Initially, both Task 1 and Task 2 inherently contain intrinsic noise. Subsequently, we designed an experiment where we progressively introduced noise into Task 2, as illustrated in Fig. 3. Specifically, we add Gaussian noise $\xi \sim N(0, \sigma_{p,2}^2)$ to the original image. We use ResNet-101 as the upstream model and use ResNet-34 and ResNet-50 as the downstream models. The experimental results indicate that as noise is continuously added to Task 2, the performance of inherited parameters consistently surpasses that of methods without parameter transfer. As presented in Fig. 3a, where the noise values gradually increase from 1 to 20, the advantage of parameter transfer not only persists but also tends to widen over time.

**Experiments on Vision Transformers.** We adopt DeiT (Touvron et al., 2021) as the architecture for both the upstream and downstream models. Specifically, both models are DeiT-Base, which consists of 12 multi-head attention blocks and 12 layers, totaling approximately 86M parameters. The upstream model is pretrained on ImageNet-2012 (Deng et al., 2009), achieving an accuracy of 81.8%. We select the 9th, 10th, and 11th layers from the upstream model as inherited parameters and transfer them to the downstream models. The downstream models are then fine-tuned on CIFAR-10 and CIFAR-100, respectively. We compare the performance of downstream models with parameter transfer against those with random initialization. The results are presented in Figure 4 in the appendix.

## 8 DISCUSSION

In this paper, we present a rigorous theoretical analysis of the parameter transfer mechanism within the framework of a two-layer ReLU convolutional neural network. Our analysis provides theoretical evidence that several key factors, such as the strength of universal signals shared between the upstream and downstream models, the sample size of the source task, and the noise level in the source task, play crucial roles in determining the effectiveness of parameter transfer. These theoretical findings are further supported by numerical simulations. Additionally, we conduct extensive real-world experiments on CIFAR-10 and CIFAR-100, employing modern neural architectures such as ResNet, VGG, and ViT, all of which consistently validate our theoretical predictions. A possible limitation of our theoretical framework is its focus on shallow neural networks. Nevertheless, even in this simplified setting, the theoretical understanding of parameter transfer remains highly non-trivial. Without first establishing a rigorous foundation for shallow networks, it would be challenging to develop solid theoretical insights for deeper and more complex architectures. This work thus serves as a necessary first step, and several promising directions remain for future research. One important direction is to extend our theoretical analysis to deep neural networks, which involves understanding more intricate dynamical systems arising from their training processes. Another interesting direction is to design regularization techniques that can guide the inherited model to select more effective weights rather than random transfer. Developing a theoretical framework to understand how regularization influences weight selection in parameter transfer remains an open and important question.

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

# A PROOF SKETCH

In this section, we briefly give the proof sketch of Theorem 4.2. We define $T^*, T^{**} = \eta^{-1}\text{poly}(n, d, \varepsilon, m)$ be the maximum admissible number of training iterations in system 1 and 2. Readers may refer to Section B for the calculation of gradient, and the meaning of the notations.

Our proof is based on a rigorous analysis of the training dynamics of CNN filters. Note that the activation functions are always non negative, hence $F_{+1}(\mathbf{W}; \mathbf{x})$ always contribute to the class $+1$, and $F_{-1}(\mathbf{W}; \mathbf{x})$ always contribute to the class $-1$. Our test error is calculated by rigorously comparing the output between $F_{+1}(\mathbf{W}; \mathbf{x})$ and $F_{-1}(\mathbf{W}; \mathbf{x})$. By the definition of $F_{+1}$ or $F_{-1}$, it is clear that the inner product of $\mathbf{w}_{j,r}$ and the signal $\mathbf{u} + \mathbf{v}_2$ in task 2 plays a key role in achieving high test accuracy.

Our analysis focused on the training dynamics of $\mathbf{w}_{j,r}^{(t)}$. By gradient calculation, $\mathbf{w}_{j,r}^{(t)}$ in the downstream model can be decomposed as

$$\mathbf{w}_{j,r}^{(t)} = \mathbf{w}_{j,r}^{(0)} + j \cdot \gamma_{j,r}^{(t)} \cdot \|\mathbf{u}\|_2^{-2} \cdot \mathbf{u} + j \cdot \gamma_{j,r,1}^{(t)} \cdot \|\mathbf{v}_1\|_2^{-2} \cdot \mathbf{v}_1 + j \cdot \gamma_{j,r,2}^{(t)} \cdot \|\mathbf{v}_2\|_2^{-2} \cdot \mathbf{v}_2$$
$$+ \sum_{i=1}^{N_1} \rho_{j,r,i,1}^{(t)} \cdot \|\boldsymbol{\xi}_{i,1}\|_2^{-2} \cdot \boldsymbol{\xi}_{i,1} + \sum_{i=1}^{N_2} \rho_{j,r,i,2}^{(t)} \cdot \|\boldsymbol{\xi}_{i,2}\|_2^{-2} \cdot \boldsymbol{\xi}_{i,2}.$$

This is because the update direction of $\mathbf{w}_{j,r}^{(t)}$ is in the space of $\text{span}\{\mathbf{u}, \mathbf{v}_1, \mathbf{v}_2, \boldsymbol{\xi}_{i,1}, \boldsymbol{\xi}_{i,2}\}$, Readers may refer to Section B for the detail. From the algorithm in Section 3, all the coefficients experienced two different systems. We proceed the analysis in the first system. With the precise characterization in the first system, we then transfer the whole analysis into the second system. readers may refer to Lemma B.2 for the two systems.

The following lemma constitutes the core technical results in our analysis of signal learning dynamics and noise memorization behavior in the first system. It is clear from the decomposition above that the coefficients $\gamma$ (i.e $\gamma_{j,r}$) are related to the growth of signal learning in the neural networks, and the coefficients $\rho$ (i.e $\rho_{j,r,i,1}$) are related to the growth of noise memorization. We would like to define $\overline{x}_t$ and $\underline{x}_t$ which help us give the precise characterization of signal learning and noise memorization. Let

$$\kappa_A = \frac{4C_2 N_1 \|\mathbf{u} + \mathbf{v}_1\|_2^2}{\sigma_{p,1}^2 d} \log(T^*) + (4C_1 + 64)N_1 \sqrt{\frac{\log(4N_1^2/\delta)}{d}} \log(T^*) + 8\sqrt{\log\left(\frac{12mN_1}{\delta}\right)} \cdot \sigma_0 \sigma_{p,1} \sqrt{d}.$$

and define $\overline{x}_t^A, \underline{x}_t^A$ be the unique solution of

$$\overline{x}_t^A + \overline{b}^A e^{\overline{x}_t^A} = \overline{c}^A t + \overline{b}^A,$$
$$\underline{x}_t^A + \underline{b}^A e^{\underline{x}_t^A} = \underline{c}^A t + \underline{b}^A,$$

where $\overline{b}^A = e^{-\kappa_A/2}, \overline{c}^A = \frac{3\eta\sigma_{p,1}^2 d}{2N_1 m}, \underline{b}^A = e^{\kappa_A/2}$ and $\underline{c}^A = \frac{\eta\sigma_{p,1}^2 d}{5N_1 m}$. We have the following lemmas.

**Lemma A.1.** *Under Condition 4.1, it holds that*

$$\frac{\eta\|\mathbf{u}\|_2^2}{\overline{c}m} \overline{x}_{t-2}^A - \frac{2\eta\|\mathbf{u}\|_2^2}{m} \leq \gamma_{j,r}^{A,(t)} \leq \frac{\eta\|\mathbf{u}\|_2^2}{\underline{c}m} \underline{x}_{t-1}^A - \frac{2\eta\|\mathbf{u}\|_2^2}{m},$$
$$\frac{\eta\|\mathbf{v}_1\|_2^2}{\overline{c}m} \overline{x}_{t-2}^A - \frac{2\eta\|\mathbf{v}_1\|_2^2}{m} \leq \gamma_{j,r,1}^{A,(t)} \leq \frac{\eta\|\mathbf{v}_1\|_2^2}{\underline{c}m} \underline{x}_{t-1}^A - \frac{2\eta\|\mathbf{v}_1\|_2^2}{m}.$$

*Moreover, for the noise memorization it holds that*

$$\frac{N_1}{12}(\overline{x}_{t-2}^A - \overline{x}_1^A) \leq \sum_{i\in[N_1]} \overline{\rho}_{j,r,i,1}^{A,(t)} \leq 5N_1\underline{x}_{t-1}^A.$$

The proof of Lemma A.1 is structured through Lemmas D.8 and D.9, which separately characterize the dynamics of signal learning and noise memorization. A key step in establishing Lemma A.1 lies in demonstrating the balanced nature of the per-sample training losses, namely that the ratio $\ell_i'^{(t)}/\ell_{i'}'^{(t)}$ remains uniformly bounded by a constant for all iterations $t$ and any $i, i' \in [N_1]$. Readers may refer to the proof of Proposition D.5 for a detailed argument on this balancing property. With the balanced loss established, we proceed to apply continuous approximation techniques, following a similar approach to that of Meng et al. (2024), and obtain the lemma above.

With the precise characterization of $\gamma$ and $\rho$ in system 1, we then transfer the analysis into the second system. The main challenges in second system are related to the analysis of the system with different initializations. In our analysis of the second system, for the universal part of $\gamma_{j,r}^{D,(t)}$, we directly define a new term $\gamma_{j,r}^{D,(t)} - \gamma_{j,r}^{D,(T^*+1)}$, and analysis is directly performed on this term. Combing the analysis in system 1, we define

$$\kappa_D = \frac{4C_2 N_2 \|\mathbf{u} + \mathbf{v}_2\|_2^2}{\sigma_{p,2}^2 d} \log(T^{**}) + \frac{4C_2 N_1 \|\mathbf{u}\|_2^2}{\sigma_{p,1}^2 d} \log(T^*) + 16\sqrt{\log(12mN_2/\delta)} \cdot \sigma_0 \sigma_{p,2} \sqrt{d}$$

$$+ (4C_1 + 64)(N_1 \frac{\sigma_{p,2}}{\sigma_{p,1}} + N_2)\sqrt{\frac{\log(4(N_1^2 + N_2^2)/\delta)}{d}} \log(T^{**}).$$

With the transfer from system 1 into system 2, we give the characterization of noise memorization and signal learning in the system 2. Let $\overline{x}_t^D, \underline{x}_t^D$ be the unique solution of

$$\overline{x}_t^D + \overline{b}^D e^{\overline{x}_t^D} = \overline{c}^D t + \overline{b}^D,$$

$$\underline{x}_t^D + \underline{b}^D e^{\underline{x}_t^D} = \underline{c}^D t + \underline{b}^D,$$

where $\overline{b}^D = e^{-\kappa_D/2}, \overline{c}^D = \frac{3\eta\sigma_{p,2}^2 d}{2N_2 m}, \underline{b}^D = e^{\kappa_D/2}$ and $\underline{c}^D = \frac{\eta\sigma_{p,2}^2 d}{5N_2 m}$. The coefficient in system 2 can be characterized as in the following lemma.

**Lemma A.2.** *Under Condition 4.1, for $T^* + 1 \leq t \leq T^{**}$, it holds that*

$$\frac{\eta\|\mathbf{u}\|_2^2}{\underline{c}^D m} \underline{x}_{t-2}^D - \frac{2\eta\|\mathbf{u}\|_2^2}{m} \leq \gamma_{j,r}^{D,(t)} - \gamma_{j,r}^{D,(T^*+1)} \leq \frac{\eta\|\mathbf{u}\|_2^2}{\overline{c}^D m} \overline{x}_{t-1}^D - \frac{2\eta\|\mathbf{u}\|_2^2}{m},$$

$$\frac{\eta\|\mathbf{v}\|_2^2}{\underline{c}^D m} \underline{x}_{t-2}^D - \frac{2\eta\|\mathbf{v}_2\|_2^2}{m} \leq \gamma_{j,r,2}^{D,(t)} \leq \frac{\eta\|\mathbf{v}\|_2^2}{\overline{c}^D m} \overline{x}_{t-1}^D - \frac{2\eta\|\mathbf{v}_2\|_2^2}{m}.$$

*Moreover, for the noise memorization term, it holds that*

$$\frac{N_2}{12}(\underline{x}_{t-2}^D - \underline{x}_1^D) \leq \sum_{i\in[N_2]} \overline{\rho}_{j,r,i,2}^{D,(t)} \leq 5N_2 \overline{x}_{t-1}^D.$$

With Lemma A.1 and A.2, our analysis then focuses on how much the training data noises $\boldsymbol{\xi}_i$ have been memorized by the CNN filters, and then the training loss and the test error can be calculated and bounded based on their definitions. Specifically, for the test accuracy, we can directly achieve the rate of $\langle \mathbf{w}_{j,r}^{D,(t)}, y_{\text{new}}(\mathbf{u} + \mathbf{v}_2)\rangle$ and $\langle \mathbf{w}_{j,r}^{D,(t)}, \boldsymbol{\xi}\rangle$ for the new data sample point $(\mathbf{u} + \mathbf{v}_2, \boldsymbol{\xi})$ by the expression of $\mathbf{w}_{j,r}^{(t)}$. Direct comparison will achieve our desired results. For the training loss, the inner product of $\mathbf{w}_{j,r}^{(t)}$ and $\boldsymbol{\xi}_{i,2}$ will make the output of neural networks large, leading to small training loss.

# B GRADIENT CALCULATION

In this section, we give the signal-noise decomposition of the weights and the update rule of each part in the weights. Moreover, we give the iterative equations for Task 1 and Task 2 separately. We use the superscript A for the upstream model in Task 1 and the superscript D for the downstream model in Task 2.

**Definition B.1.** *Let $\boldsymbol{w}_{j,r}^{(t)}$ for $j \in \{+1, -1\}$ and $r \in \{1, 2, \ldots, m\}$ be the convolution filters of the CNN at the $t$-th iteration of gradient descent. Then there exist unique coefficients $\gamma_{j,r}^{(t)}, \gamma_{j,r,1}^{(t)}, \gamma_{j,r,2}^{(t)} \geq 0, \rho_{j,r,i,1}^{(t)}$ and $\rho_{j,r,i,2}^{(t)}$ such that,*

$$\mathbf{w}_{j,r}^{(t)} = \mathbf{w}_{j,r}^{(0)} + j \cdot \gamma_{j,r}^{(t)} \cdot \|\mathbf{u}\|_2^{-2} \cdot \mathbf{u} + j \cdot \gamma_{j,r,1}^{(t)} \cdot \|\mathbf{v}_1\|_2^{-2} \cdot \mathbf{v}_1 + j \cdot \gamma_{j,r,2}^{(t)} \cdot \|\mathbf{v}_2\|_2^{-2} \cdot \mathbf{v}_2$$

$$+ \sum_{i=1}^{N_1} \rho_{j,r,i,1}^{(t)} \cdot \|\boldsymbol{\xi}_{i,1}\|_2^{-2} \cdot \boldsymbol{\xi}_{i,1} + \sum_{i=1}^{N_2} \rho_{j,r,i,2}^{(t)} \cdot \|\boldsymbol{\xi}_{i,2}\|_2^{-2} \cdot \boldsymbol{\xi}_{i,2}. \tag{B.1}$$

*Further denote*

$$\overline{\rho}_{j,r,i,s}^{(t)} := \rho_{j,r,i,s}^{(t)} \mathbf{1}\left(\rho_{j,r,i,s}^{(t)} \geq 0\right), \quad \underline{\rho}_{j,r,i,s}^{(t)} := \rho_{j,r,i,s}^{(t)} \mathbf{1}\left(\rho_{j,r,i,s}^{(t)} \leq 0\right).$$

*Then*

$$\mathbf{w}_{j,r}^{(t)} = \mathbf{w}_{j,r}^{(0)} + j \cdot \gamma_{j,r}^{(t)} \cdot \|\mathbf{u}\|_2^{-2} \cdot \mathbf{u} + j \cdot \gamma_{j,r,1}^{(t)} \cdot \|\mathbf{v}_1\|_2^{-2} \cdot \mathbf{v}_1 + j \cdot \gamma_{j,r,2}^{(t)} \cdot \|\mathbf{v}_2\|_2^{-2} \cdot \mathbf{v}_2$$

$$+ \sum_{i=1}^{N_1} \overline{\rho}_{j,r,i,1}^{(t)} \cdot \|\boldsymbol{\xi}_{i,1}\|_2^{-2} \cdot \boldsymbol{\xi}_{i,1} + \sum_{i=1}^{N_2} \overline{\rho}_{j,r,i,2}^{(t)} \cdot \|\boldsymbol{\xi}_{i,2}\|_2^{-2} \cdot \boldsymbol{\xi}_{i,2}$$

$$+ \sum_{i=1}^{N_1} \underline{\rho}_{j,r,i,1}^{(t)} \cdot \|\boldsymbol{\xi}_{i,1}\|_2^{-2} \cdot \boldsymbol{\xi}_{i,1} + \sum_{i=1}^{N_2} \underline{\rho}_{j,r,i,2}^{(t)} \cdot \|\boldsymbol{\xi}_{i,2}\|_2^{-2} \cdot \boldsymbol{\xi}_{i,2}. \tag{B.2}$$

Based on the above definition of the signal-noise decomposition of the weights, we will prove the unique of the coefficients and give the iterative equations in the next lemma.

**Lemma B.2** (Update Rule). *The coefficients are defined as Definition B.1. Note that We use the superscript A for the upstream model in Task 1 and the superscript D for the downstream model in Task 2. The coefficients in Task 1 are unique and satisfy the following iterative equations:*

$$\gamma_{j,r}^{A,(0)}, \gamma_{j,r,1}^{A,(0)}, \overline{\rho}_{j,r,i,1}^{A,(0)}, \underline{\rho}_{j,r,i,1}^{A,(0)}, \gamma_{j,r,2}^{A,(0)}, \overline{\rho}_{j,r,i,2}^{A,(0)}, \underline{\rho}_{j,r,i,2}^{A,(0)} = 0,$$

$$\gamma_{j,r}^{A,(t+1)} = \gamma_{j,r}^{A,(t)} - \frac{\eta}{N_1 m} \sum_{i \in [N_1]} \ell_i'^{(t)} \cdot \sigma'\big(\langle \mathbf{w}_{j,r}^{A,(t)}, y_{i,1} \cdot \mathbf{x}_1 \rangle\big) \cdot \|\mathbf{u}\|_2^2,$$

$$\gamma_{j,r,2}^{A,(t+1)} = \gamma_{j,r,2}^{A,(t)}, \quad \overline{\rho}_{j,r,i,2}^{A,(t+1)} = \overline{\rho}_{j,r,i,2}^{A,(t)}, \quad \underline{\rho}_{j,r,i,2}^{A,(t+1)} = \underline{\rho}_{j,r,i,2}^{A,(t)},$$

$$\gamma_{j,r,1}^{A,(t+1)} = \gamma_{j,r,1}^{A,(t)} - \frac{\eta}{N_1 m} \sum_{i \in [N_1]} \ell_i'^{(t)} \cdot \sigma'\big(\langle \mathbf{w}_{j,r}^{A,(t)}, y_{i,1} \cdot \mathbf{x}_1 \rangle\big) \cdot \|\mathbf{v}_1\|_2^2,$$

$$\overline{\rho}_{j,r,i,1}^{A,(t+1)} = \overline{\rho}_{j,r,i,1}^{A,(t)} - \frac{\eta}{N_1 m} \ell_i'^{(t)} \cdot \sigma'\big(\langle \mathbf{w}_{j,r}^{A,(t)}, \boldsymbol{\xi}_{i,2} \rangle\big) \cdot \|\boldsymbol{\xi}_{i,2}\|_2^2 \cdot \mathbf{1}\{y_{i,1} = j\},$$

$$\underline{\rho}_{j,r,i,1}^{A,(t+1)} = \underline{\rho}_{j,r,i,1}^{A,(t)} + \frac{\eta}{N_1 m} \ell_i'^{(t)} \cdot \sigma'\big(\langle \mathbf{w}_{j,r}^{A,(t)}, \boldsymbol{\xi}_{i,2} \rangle\big) \cdot \|\boldsymbol{\xi}_{i,2}\|_2^2 \cdot \mathbf{1}\{y_{i,1} = -j\},$$

*for all $r \in [m]$, $j \in \{\pm 1\}$ and $i \in [N_1]$. For the coefficients in task 2 are also unique and satisfy the following iterative equations:*

$$\gamma_{j,r}^{D,(t+1)} = \gamma_{j,r}^{D,(t)} - \frac{\eta}{N_2 m} \sum_{i \in [N_2]} \ell_i'^{D,(t)} \cdot \sigma'\big(\langle \mathbf{w}_{j,r}^{D,(t)}, y_{i,2} \cdot \mathbf{x}_1 \rangle\big) \cdot \|\mathbf{u}\|_2^2,$$

$$\gamma_{j,r,1}^{D,(t+1)} = \gamma_{j,r,1}^{D,(t)}, \quad \overline{\rho}_{j,r,i,1}^{D,(t+1)} = \overline{\rho}_{j,r,i,1}^{D,(t)}, \quad \underline{\rho}_{j,r,i,1}^{D,(t+1)} = \underline{\rho}_{j,r,i,1}^{D,(t)},$$

$$\gamma_{j,r,2}^{D,(t+1)} = \gamma_{j,r,2}^{D,(t)} - \frac{\eta}{N_2 m} \sum_{i \in [N_2]} \ell_i'^{D,(t)} \cdot \sigma'\big(\langle \mathbf{w}_{j,r}^{D,(t)}, y_{i,2} \cdot \mathbf{x}_1 \rangle\big) \cdot \|\mathbf{v}_2\|_2^2,$$

$$\overline{\rho}_{j,r,i,2}^{D,(t+1)} = \overline{\rho}_{j,r,i,2}^{D,(t)} - \frac{\eta}{N_2 m} \ell_i'^{D,(t)} \cdot \sigma'\big(\langle \mathbf{w}_{j,r}^{D,(t)}, \boldsymbol{\xi}_{i,2} \rangle\big) \cdot \|\boldsymbol{\xi}_{i,2}\|_2^2 \cdot \mathbf{1}\{y_{i,2} = j\},$$

$$\underline{\rho}_{j,r,i,2}^{D,(t+1)} = \underline{\rho}_{j,r,i,2}^{D,(t)} + \frac{\eta}{N_2 m} \ell_i'^{D,(t)} \cdot \sigma'\big(\langle \mathbf{w}_{j,r}^{D,(t)}, \boldsymbol{\xi}_{i,2} \rangle\big) \cdot \|\boldsymbol{\xi}_{i,2}\|_2^2 \cdot \mathbf{1}\{y_{i,2} = -j\},$$

*for all $r \in [m]$, $j \in \{\pm 1\}$ and $i \in [N_2]$.*

*Proof of Lemma B.2.* In Task 1, by the definition of data generation model in Definition 3.1 and the Gaussian initialization of the network weights, it is obvious that all the vectors (signals, noise and weights) are linearly independent with probability 1. So the decomposition equation B.2 is unique in Task 1. The update iterative equations can be calculated

directly by $\mathbf{w}_{j,r}^{A,(t+1)} = \mathbf{w}_{j,r}^{A,(t)} - \eta \nabla_{\mathbf{w}_{j,r}^A} L_{Task1}(\mathbf{W}^{A,(t)})$. That is shown as following

$$\gamma_{j,r}^{(t+1)} = \gamma_{j,r}^{(t)} - \frac{\eta}{N_1 m} \sum_{i \in [N_1]} \ell_i'^{(t)} \cdot \sigma'(\langle \mathbf{w}_{j,r}^{(t)}, y_{i,1} \cdot \mathbf{x}_1 \rangle) \cdot \|\mathbf{u}\|_2^2,$$

$$\gamma_{j,r,1}^{(t+1)} = \gamma_{j,r,1}^{(t)} - \frac{\eta}{N_1 m} \sum_{i \in [N_1]} \ell_i'^{(t)} \cdot \sigma'(\langle \mathbf{w}_{j,r}^{(t)}, y_{i,1} \cdot \mathbf{x}_1 \rangle) \cdot \|\mathbf{v}_1\|_2^2,$$

$$\rho_{j,r,i,1}^{(t+1)} = \rho_{j,r,i,1}^{(t)} - \frac{\eta}{N_1 m} \ell_i'^{(t)} \cdot \sigma'(\langle \mathbf{w}_{j,r}^{(t)}, \boldsymbol{\xi}_{i,1} \rangle) \cdot \|\boldsymbol{\xi}_{i,1}\|_2^2 \cdot j y_{i,1}.$$

Note that $\gamma_{j,r,2}^{(t)}$ and $\rho_{j,r,i,2}^{(t)}$ remain unchanged. Furthermore, denoted by $\overline{\rho}_{j,r,i,1}^{(t)} = \rho_{j,r,i,1}^{(t)} \mathbf{1}(\rho_{j,r,i,1}^{(t)} \geq 0)$ and $\underline{\rho}_{j,r,i,1}^{(t)} = \rho_{j,r,i,1}^{(t)} \mathbf{1}(\rho_{j,r,i,1}^{(t)} \leq 0)$, we have

$$\overline{\rho}_{j,r,i,1}^{(t+1)} = \overline{\rho}_{j,r,i,1}^{(t)} - \frac{\eta}{N_1 m} \ell_i'^{(t)} \cdot \sigma'(\langle \mathbf{w}_{j,r}^{(t)}, \boldsymbol{\xi}_{i,1} \rangle) \cdot \|\boldsymbol{\xi}_{i,1}\|_2^2 \cdot \mathbf{1}\{y_{i,1} = j\},$$

$$\underline{\rho}_{j,r,i,1}^{(t+1)} = \underline{\rho}_{j,r,i,1}^{(t)} + \frac{\eta}{N_1 m} \ell_i'^{(t)} \cdot \sigma'(\langle \mathbf{w}_{j,r}^{(t)}, \boldsymbol{\xi}_{i,1} \rangle) \cdot \|\boldsymbol{\xi}_{i,1}\|_2^2 \cdot \mathbf{1}\{y_{i,1} = -j\}.$$

Next, we prove the results for Task 2. Note that partial weights ($\alpha m \leq r \leq m$) are re-initialized at the start of Task 2. Then, by the definition of data generation model in Definition 3.2 and the Gaussian initialization of the re-initialized weights, it is obvious that all the vectors (signals, noise and weights) are also linearly independent with probability 1. So the decomposition equation B.2 is unique in Task 2. The update iterative equations can be calculated directly by $\mathbf{w}_{j,r}^{D,(t+1)} = \mathbf{w}_{j,r}^{D,(t)} - \eta \nabla_{\mathbf{w}_{j,r}^D} L_{Task2}(\mathbf{W}^{D,(t)})$. That is shown as following

$$\gamma_{j,r}^{(t+1)} = \gamma_{j,r}^{(t)} - \frac{\eta}{N_2 m} \sum_{i \in [N_2]} \ell_i'^{(t)} \cdot \sigma'(\langle \mathbf{w}_{j,r}^{(t)}, y_{i,2} \cdot \mathbf{x}_1 \rangle) \cdot \|\mathbf{u}\|_2^2,$$

$$\gamma_{j,r,2}^{(t+1)} = \gamma_{j,r,2}^{(t)} - \frac{\eta}{N_2 m} \sum_{i \in [N_2]} \ell_i'^{(t)} \cdot \sigma'(\langle \mathbf{w}_{j,r}^{(t)}, y_{i,2} \cdot \mathbf{x}_1 \rangle) \cdot \|\mathbf{v}_2\|_2^2,$$

$$\rho_{j,r,i,2}^{(t+1)} = \rho_{j,r,i,2}^{(t)} - \frac{\eta}{N_2 m} \ell_i'^{(t)} \cdot \sigma'(\langle \mathbf{w}_{j,r}^{(t)}, \boldsymbol{\xi}_{i,2} \rangle) \cdot \|\boldsymbol{\xi}_{i,2}\|_2^2 \cdot j y_{i,2}.$$

Note that $\gamma_{j,r,1}^{(t)}$ and $\rho_{j,r,i,1}^{(t)}$ remain unchanged in Task 2. Furthermore, denoted by $\overline{\rho}_{j,r,i,2}^{(t)} = \rho_{j,r,i,2}^{(t)} \mathbf{1}(\rho_{j,r,i,2}^{(t)} \geq 0)$ and $\underline{\rho}_{j,r,i,2}^{(t)} = \rho_{j,r,i,2}^{(t)} \mathbf{1}(\rho_{j,r,i,2}^{(t)} \leq 0)$, we have

$$\overline{\rho}_{j,r,i,2}^{(t+1)} = \overline{\rho}_{j,r,i,2}^{(t)} - \frac{\eta}{N_1 m} \ell_i'^{(t)} \cdot \sigma'(\langle \mathbf{w}_{j,r}^{(t)}, \boldsymbol{\xi}_{i,2} \rangle) \cdot \|\boldsymbol{\xi}_{i,2}\|_2^2 \cdot \mathbf{1}\{y_{i,2} = j\},$$

$$\underline{\rho}_{j,r,i,2}^{(t+1)} = \underline{\rho}_{j,r,i,2}^{(t)} + \frac{\eta}{N_1 m} \ell_i'^{(t)} \cdot \sigma'(\langle \mathbf{w}_{j,r}^{(t)}, \boldsymbol{\xi}_{i,2} \rangle) \cdot \|\boldsymbol{\xi}_{i,2}\|_2^2 \cdot \mathbf{1}\{y_{i,2} = -j\}.$$

Then, we complete the proof. $\qquad\square$

## C   PRELIMINARY LEMMAS

In this section, we introduce some basic technical lemmas, which can describe important properties of the data the weights at initialization.

**Lemma C.1.** *Suppose that $\delta > 0$ and $d = \Omega(\log(4 \max\{N_1, N_2\}/\delta))$, the following results hold with probability at least $1 - 3\delta$. In Task 1, for all $i, i' \in [N_1]$, we have*

$$\sigma_{p,1}^2 d/2 \leq \|\boldsymbol{\xi}_{i,1}\|_2^2 \leq 3\sigma_{p,1}^2 d/2,$$

$$|\langle \boldsymbol{\xi}_{i,1}, \boldsymbol{\xi}_{i',1} \rangle| \leq 2\sigma_{p,1}^2 \cdot \sqrt{d \log(4N_1^2/\delta)}.$$

*In Task 2, for all $i, i' \in [N_2]$, we have*

$$\sigma_{p,2}^2 d/2 \leq \|\boldsymbol{\xi}_{i,2}\|_2^2 \leq 3\sigma_{p,2}^2 d/2,$$

$$|\langle \boldsymbol{\xi}_{i,2}, \boldsymbol{\xi}_{i',2} \rangle| \leq 2\sigma_{p,2}^2 \cdot \sqrt{d \log(4N_2^2/\delta)}.$$

*Moreover, for all $i \in [N_1], i' \in [N_2]$, we have*

$$|\langle \boldsymbol{\xi}_{i,1}, \boldsymbol{\xi}_{i',2} \rangle| \le 2\sigma_{p,1}\sigma_{p,2} \cdot \sqrt{d \log(4 \max\{N_1^2, N_2^2\}/\delta)}.$$

*Proof of Lemma C.1.* For Task 1, by Bernstein's inequality, it holds with probability at least $1 - \delta/(2N_1)$

$$\left| \|\boldsymbol{\xi}_{i,1}\|_2^2 - \sigma_{p,1}^2 d \right| \le O(\sigma_{p,1}^2 \cdot \sqrt{d \log(4N_1/\delta)}).$$

By setting $d = \Omega(\log(4 \max\{N_1, N_2\}/\delta))$, we have

$$\sigma_{p,1}^2 d/2 \le \|\boldsymbol{\xi}_{i,1}\|_2^2 \le 3\sigma_{p,1}^2 d/2.$$

For the second result for Task 1, for $i \ne i'$, $\langle \boldsymbol{\xi}_{i,1}, \boldsymbol{\xi}_{i',1} \rangle$ has mean zero. Then by Bernstein's inequality, it holds with probability at least $1 - \delta/(2N_1^2)$

$$|\langle \boldsymbol{\xi}_{i,1}, \boldsymbol{\xi}_{i',1} \rangle| \le 2\sigma_{p,1}^2 \cdot \sqrt{d \log(4N_1^2/\delta)}.$$

The proof for Task 2 is similar and we omit it here. For $i \in [N_1], i' \in [N_2]$, by Bernstein's inequality, it holds with probability at least $1 - \delta/(2N_1 N_2)$

$$|\langle \boldsymbol{\xi}_{i,1}, \boldsymbol{\xi}_{i',2} \rangle| \le 2\sigma_{p,1}\sigma_{p,2} \cdot \sqrt{d \log(4 \max\{N_1^2, N_2^2\}/\delta)}.$$

Finally, by union bound, we complete the proof. $\qquad \square$

**Lemma C.2** (Meng et al. (2024)). *Suppose that $d = \Omega(\log(m \max N_1, N_2/\delta))$, $m = \Omega(\log(1/\delta))$. Then with probability at least $1 - \delta$,*

$$\sigma_0^2 d/2 \le \|\mathbf{w}_{j,r}^{(0)}\|_2^2 \le 3\sigma_0^2 d/2,$$

$$|\langle \mathbf{w}_{j,r}^{(0)}, \boldsymbol{\mu} \rangle| \le \sqrt{2 \log(12m/\delta)} \cdot \sigma_0 \|\boldsymbol{\mu}\|_2,$$

$$|\langle \mathbf{w}_{j,r}^{(0)}, \boldsymbol{\xi}_i \rangle| \le 2\sqrt{\log(12mn/\delta)} \cdot \sigma_0 \sigma_p \sqrt{d},$$

*for all $r \in [m], j \in \{\pm 1\}, i \in [n], \boldsymbol{\xi}_i \in \{\boldsymbol{\xi}_{i,1}, \boldsymbol{\xi}_{i,2}\}$ and $\boldsymbol{\mu} \in \{\mathbf{u}, \mathbf{v}_1, \mathbf{v}_2\}$. Moreover,*

$$\sigma_0 \|\boldsymbol{\mu}\|_2/2 \le \max_{r \in [m]} \langle \mathbf{w}_{j,r}^{(0)}, \boldsymbol{\mu} \rangle \le \sqrt{2 \log(12m/\delta)} \cdot \sigma_0 \|\boldsymbol{\mu}\|_2,$$

$$\sigma_0 \sigma_p \sqrt{d}/4 \le \max_{r \in [m]} \langle \mathbf{w}_{j,r}^{(0)}, \boldsymbol{\xi}_i \rangle \le 2\sqrt{\log(12mn/\delta)} \cdot \sigma_0 \sigma_p \sqrt{d},$$

*for all $j \in \{\pm 1\}, i \in [n], (\boldsymbol{\xi}_i, \sigma_p) \in \{(\boldsymbol{\xi}_{i,1}, \sigma_{p,1}), (\boldsymbol{\xi}_{i,2}, \sigma_{p,2})\}$ and $\boldsymbol{\mu} \in \{\mathbf{u}, \mathbf{v}_1, \mathbf{v}_2\}$.*

**Lemma C.3** (Kou et al. (2023)). *Suppose that $\delta > 0$, $m = \Omega(\log(2 \max\{N_1, N_2\}/\delta))$. Define $S_i^{A,(t)} = \{r \in [m] : \langle \mathbf{w}_{y_{i,1},r}^{(t)}, \boldsymbol{\xi}_{i,1} \rangle > 0\}$ and $S_i^{D,(t)} = \{r \in [m] : \langle \mathbf{w}_{y_{i,2},r}^{(t)}, \boldsymbol{\xi}_{i,2} \rangle > 0\}$. Then, with probability at least $1 - \delta$,*

$$|S_i^{A,(0)}| \ge 0.4m \quad and \quad |S_i^{D,(0)}| \ge 0.4m$$

*for all $i \in [n]$.*

*Proof of Lemma C.3.* By definition, we know that $S_i^{A,(0)} = \{r \in [m] : \langle \mathbf{w}_{y_{i,1},r}^{(0)}, \boldsymbol{\xi}_{i,1} \rangle > 0\}$. At initialization, it is obvious that $P(\langle \mathbf{w}_{y_{i,1},r}^{(0)}, \boldsymbol{\xi}_{i,1} \rangle > 0) = 0.5$. By Hoeffding's inequality, it holds with probability at least $1 - \delta/(2N_1)$ that

$$\left| \frac{|S_i^{A,(0)}|}{m} - 0.5 \right| \le \sqrt{\frac{\log(4N_1/\delta)}{2m}}.$$

So, the proof will be completed by applying union bound as if $\sqrt{\log(4N_1/\delta)/2m} \le 0.1$, i.e., $m \ge 50 \log(4N_1/\delta)$. The condition is satisfied. The proof for $|S_i^{D,(0)}| \ge 0.4m$ is similar and we omit it here. $\qquad \square$

**Lemma C.4** (Meng et al. (2024)). *Suppose that a sequence $a_t, t \ge 0$ follows the iterative formula*

$$a_{t+1} = a_t + \frac{c}{1 + be^{a_t}},$$

*for some $1 \ge c \ge 0$ and $b \ge 0$. Then it holds that*

$$x_t \le a_t \le \frac{c}{1 + be^{a_0}} + x_t$$

*for all $t \ge 0$. Here, $x_t$ is the unique solution of*

$$x_t + be^{x_t} = ct + a_0 + be^{a_0}.$$

# D THE FIRST SYSTEM

Note that the downstream model maintains an identical architecture to the upstream model but inherits only half of the first-layer parameters from the upstream model. The remaining half undergoes re-initialization, effectively creating a hybrid initialization scheme. To rigorously distinguish the training epochs between the upstream model's performance on Task 1 and the downstream model's performance on Task 2, we formally define $T^*$ as the transition point marking the boundary between the two tasks. The upstream model (Task 1): Training occurs over the interval $[0, T^*]$; the downstream model (Task 2): Training proceeds from $[T^*, T^{**}]$.

Lemma B.2 clearly gives us the update rule in both system. Note that in parameter transfer, some values of $\mathbf{w}_{j,r}$ are changed into the initialized normal distribution, we will later incoporate such change in the second system and analyze the test error.

## D.1 COEFFICIENT SCALE ANALYSIS

We denote the results from the upstream model (Task 1) with a superscript notation A.

**Proposition D.1.** *Under Condition 4.1, for $0 \le t \le T^*$, it holds that*

$$0 \le \bar{\rho}_{j,r,i,1}^{A,(t)} \le 4\log(T^*), \tag{D.1}$$

$$0 \ge \underline{\rho}_{j,r,i,1}^{A,(t)} \ge -2\sqrt{\log\left(\frac{12mN_1}{\delta}\right)} \cdot \sigma_0\sigma_{p,1}\sqrt{d} - C_1\sqrt{\frac{\log\left(\frac{4N_1^2}{\delta}\right)}{d}}N_1\log(T^*) \ge -4\log(T^*), \tag{D.2}$$

$$0 \le \gamma_{j,r}^{A,(t)} \le \frac{C_2 N_1 \|\mathbf{u}\|_2^2}{\sigma_{p,1}^2 d}\log(T^*), \tag{D.3}$$

$$0 \le \gamma_{j,r,1}^{A,(t)} \le \frac{C_2 N_1 \|\mathbf{v}_1\|_2^2}{\sigma_{p,1}^2 d}\log(T^*), \tag{D.4}$$

*for all $r \in [m], j \in \{\pm 1\}, i \in [N_1]$, where $C_1$ and $C_2$ are two absolute constant.*

We will prove Proposition D.1 by induction. Before that we give some important technical lemmas used in the proof.

**Lemma D.2.** *Under Condition 4.1, for $0 < t < T^*$, suppose equation D.1, equation D.2, equation D.3, equation D.4 hold at iteration t. Then, for all $r \in [m], j \in \{\pm 1\}, i \in [N_1]$, it holds that*

$$\left|\langle\mathbf{w}_{j,r}^{A,(t)} - \mathbf{w}_{j,r}^{A,(0)}, \boldsymbol{\xi}_{i,1}\rangle - \underline{\rho}_{j,r,i,1}^{A,(t)}\right| \le 16N_1\sqrt{\frac{\log(4N_1^2/\delta)}{d}}\log(T^*), \quad j \ne y_{i,1}; \tag{D.5}$$

$$\left|\langle\mathbf{w}_{j,r}^{A,(t)} - \mathbf{w}_{j,r}^{A,(0)}, \boldsymbol{\xi}_{i,1}\rangle - \bar{\rho}_{j,r,i,1}^{A,(t)}\right| \le 16N_1\sqrt{\frac{\log(4N_1^2/\delta)}{d}}\log(T^*), \quad j = y_{i,1}. \tag{D.6}$$

*Proof of Lemma D.2.* By equation B.2, we have

$$\langle\mathbf{w}_{j,r}^{A,(t)} - \mathbf{w}_{j,r}^{A,(0)}, \boldsymbol{\xi}_{i,1}\rangle = \sum_{i'=1}^{N_1}\bar{\rho}_{j,r,i',1}^{A,(t)} \cdot \|\boldsymbol{\xi}_{i',1}\|_2^{-2} \cdot \langle\boldsymbol{\xi}_{i',1}, \boldsymbol{\xi}_{i,1}\rangle + \sum_{i'=1}^{N_1}\underline{\rho}_{j,r,i',1}^{A,(t)} \cdot \|\boldsymbol{\xi}_{i',1}\|_2^{-2} \cdot \langle\boldsymbol{\xi}_{i',1}, \boldsymbol{\xi}_{i,1}\rangle. \tag{D.7}$$

When $j \ne y_{i,1}$, we have $\bar{\rho}_{j,r,i',1}^{A,(t)} = 0$ and the equation equation D.7 can be turned into

$$\langle\mathbf{w}_{j,r}^{A,(t)} - \mathbf{w}_{j,r}^{A,(0)}, \boldsymbol{\xi}_{i,1}\rangle = \underline{\rho}_{j,r,i,1}^{A,(t)} + \sum_{i'\ne i}\underline{\rho}_{j,r,i',1}^{A,(t)} \cdot \|\boldsymbol{\xi}_{i',1}\|_2^{-2} \cdot \langle\boldsymbol{\xi}_{i',1}, \boldsymbol{\xi}_{i,1}\rangle. \tag{D.8}$$

Then we bound the remainder as

$$\left|\sum_{i'\ne i}\underline{\rho}_{j,r,i',1}^{A,(t)} \cdot \|\boldsymbol{\xi}_{i',1}\|_2^{-2} \cdot \langle\boldsymbol{\xi}_{i',1}, \boldsymbol{\xi}_{i,1}\rangle\right| \le \sum_{i'\ne i}\left|\underline{\rho}_{j,r,i',1}^{A,(t)}\right| \cdot \|\boldsymbol{\xi}_{i',1}\|_2^{-2} \cdot \left|\langle\boldsymbol{\xi}_{i',1}, \boldsymbol{\xi}_{i,1}\rangle\right|$$

$$\le 16N_1\sqrt{\frac{\log(4N_1^2/\delta)}{d}}\log(T^*).$$

We finish the proof of equation D.5. When $j = y_{i,1}$, we have $\underline{\rho}_{j,r,i',1}^{A,(t)} = 0$ and the equation equation D.7 can be turned into

$$\langle \mathbf{w}_{j,r}^{A,(t)} - \mathbf{w}_{j,r}^{A,(0)}, \boldsymbol{\xi}_{i,1} \rangle = \bar{\rho}_{j,r,i,1}^{A,(t)} + \sum_{i' \neq i} \bar{\rho}_{j,r,i',1}^{A,(t)} \cdot \|\boldsymbol{\xi}_{i',1}\|_2^{-2} \cdot \langle \boldsymbol{\xi}_{i',1}, \boldsymbol{\xi}_{i,1} \rangle.$$

Then we bound the remainder as

$$\left| \sum_{i' \neq i} \bar{\rho}_{j,r,i',1}^{A,(t)} \cdot \|\boldsymbol{\xi}_{i',1}\|_2^{-2} \cdot \langle \boldsymbol{\xi}_{i',1}, \boldsymbol{\xi}_{i,1} \rangle \right| \leq \sum_{i' \neq i} \left| \bar{\rho}_{j,r,i',1}^{A,(t)} \right| \cdot \|\boldsymbol{\xi}_{i',1}\|_2^{-2} \cdot \left| \langle \boldsymbol{\xi}_{i',1}, \boldsymbol{\xi}_{i,1} \rangle \right|$$

$$\leq 16 N_1 \sqrt{\frac{\log(4N_1^2/\delta)}{d}} \log(T^*).$$

We finish the proof of equation D.6. $\qquad\qquad\square$

Next, we will give the bound for the output of the network. Before that, we define $\kappa_A$ as

$$\kappa_A = \frac{4C_2 N_1 \|\mathbf{u} + \mathbf{v}_1\|_2^2}{\sigma_{p,1}^2 d} \log(T^*) + (4C_1 + 64) N_1 \sqrt{\frac{\log(4N_1^2/\delta)}{d}} \log(T^*) + 8\sqrt{\log\left(\frac{12mN_1}{\delta}\right)} \cdot \sigma_0 \sigma_{p,1} \sqrt{d}.$$

By the condition of $d$ in Condition 4.1, we have $\kappa_A \leq 0.1$.

**Lemma D.3.** *Under Condition 4.1, for $0 < t < T^*$, suppose equation D.1, equation D.2, equation D.3, equation D.4 hold at iteration t. Then, it holds that*

$$F_{-y_{i,1}}(\mathbf{W}_{-y_{i,1}}^{A,(t)}, \mathbf{x}_{i,1}) \leq \kappa_A/4, \quad -\kappa_A/4 + \frac{1}{m}\sum_{r=1}^{m} \bar{\rho}_{j,r,i,1}^{A,(t)} \leq F_{y_{i,1}}(\mathbf{W}_{y_{i,1}}^{A,(t)}, \mathbf{x}_{i,1}) \leq \kappa_A/4 + \frac{1}{m}\sum_{r=1}^{m} \bar{\rho}_{j,r,i,1}^{A,(t)}$$

$$-\kappa_A/2 + \frac{1}{m}\sum_{r=1}^{m} \bar{\rho}_{j,r,i,1}^{A,(t)} \leq y_{i,1} f(\mathbf{W}^{A,(t)}, \mathbf{x}_{i,1}) \leq \kappa_A/2 + \frac{1}{m}\sum_{r=1}^{m} \bar{\rho}_{j,r,i,1}^{A,(t)}.$$

*Proof.* Recall that the definition of $F_j(\mathbf{W}_j^{A,(t)}, \mathbf{x}_{i,1})$ as

$$F_j(\mathbf{W}_j^{A,(t)}, \mathbf{x}_{i,1}) = \frac{1}{m}\sum_{r=1}^{m} \left[ \sigma\left(\langle \mathbf{w}_{j,r}, y_{i,1}(\mathbf{u} + \mathbf{v}_1) \rangle\right) + \sigma\left(\langle \mathbf{w}_{j,r}, \boldsymbol{\xi}_{i,1} \rangle\right) \right].$$

When $j = -y_{i,1}$, we have

$$F_{-y_{i,1}}(\mathbf{W}_{-y_{i,1}}^{A,(t)}, \mathbf{x}_{i,1}) \leq \frac{1}{m}\sum_{r=1}^{m} \left[ \left| \langle \mathbf{w}_{j,r}, y_{i,1}\mathbf{u} \rangle \right| + \left[ \left| \langle \mathbf{w}_{j,r}, y_{i,1}\mathbf{v}_1 \rangle \right| + \left| \langle \mathbf{w}_{j,r}, \boldsymbol{\xi}_{i,1} \rangle \right| \right] \right]$$

$$\leq \gamma_{j,r}^{A,(t)} + \gamma_{j,r,1}^{A,(t)} + \underline{\rho}_{j,r,i,1}^{A,(t)} + 16N_1 \sqrt{\frac{\log(4N_1^2/\delta)}{d}} \log(T^*)$$

$$\leq \frac{C_2 N_1 \|\mathbf{u} + \mathbf{v}_1\|_2^2}{\sigma_{p,1}^2 d} \log(T^*) + 2\sqrt{\log\left(\frac{12mN_1}{\delta}\right)} \cdot \sigma_0 \sigma_{p,1} \sqrt{d}$$

$$+ C_1 N_1 \sqrt{\frac{\log(4N_1^2/\delta)}{d}} \log(T^*) + 16N_1 \sqrt{\frac{\log(4N_1^2/\delta)}{d}} \log(T^*),$$

where the first inequality uses triangle inequality, the second inequality is by Lemma D.2, the third inequality is by equation D.2, equation D.3, equation D.4 and the fact that $\mathbf{u} \perp \mathbf{v}_1$. When $j = y_{i,1}$, we have

$$\left| F_{y_{i,1}}(\mathbf{W}_{y_{i,1}}^{A,(t)}, \mathbf{x}_{i,1}) - \frac{1}{m}\sum_{r=1}^{m} \bar{\rho}_{j,r,i,1}^{A,(t)} \right|$$

$$\leq \frac{1}{m}\sum_{r=1}^{m} \left[ \left| \langle \mathbf{w}_{j,r}, y_{i,1}\mathbf{u} \rangle \right| + \left[ \left| \langle \mathbf{w}_{j,r}, y_{i,1}\mathbf{v}_1 \rangle \right| + \left| \langle \mathbf{w}_{j,r}, \boldsymbol{\xi}_{i,1} \rangle - \bar{\rho}_{j,r,i,1}^{A,(t)} \right| \right] \right]$$

$$\leq \gamma_{j,r}^{A,(t)} + \gamma_{j,r,1}^{A,(t)} + 2\sqrt{\log\left(\frac{12mN_1}{\delta}\right)} \cdot \sigma_0 \sigma_{p,1} \sqrt{d} + 16N_1 \sqrt{\frac{\log(4N_1^2/\delta)}{d}} \log(T^*)$$

$$\leq \frac{C_2 N_1 \|\mathbf{u} + \mathbf{v}_1\|_2^2}{\sigma_{p,1}^2 d} \log(T^*) + 2\sqrt{\log\left(\frac{12mN_1}{\delta}\right)} \cdot \sigma_0 \sigma_{p,1} \sqrt{d} + 16N_1 \sqrt{\frac{\log(4N_1^2/\delta)}{d}} \log(T^*),$$

where the first inequality uses triangle inequality, the second inequality is by Lemma D.2, the third inequality is by equation D.2, equation D.3, equation D.4, and the last inequality uses the fact that $\mathbf{u} \perp \mathbf{v}_1$. At last, because

$$y_{i,1}f(\mathbf{W}^{A,(t)}, \mathbf{x}_{i,1}) = F_{y_{i,1}}(\mathbf{W}^{A,(t)}_{y_{i,1}}, \mathbf{x}_{i,1}) - F_{-y_{i,1}}(\mathbf{W}^{A,(t)}_{-y_{i,1}}, \mathbf{x}_{i,1}),$$

we complete the proof. □

**Lemma D.4.** *Under Condition 4.1, suppose equation D.1, equation D.2, equation D.3, equation D.4 hold for any iteration $0 < t < T^*$. Then, the following results hold for any iteration $t$:*

1. $\frac{1}{m} \sum_{r=1}^m \left[ \bar{\rho}^{A,(t)}_{y_{i,1},r,i,1} - \bar{\rho}^{A,(t)}_{r,k,i,1} \right] \le \log(12) + \kappa_A + \sqrt{\log(2N_1/\delta)/m}$ *for all* $i, k \in [N_1]$.

2. $S_i^{A,(0)} \subseteq S_i^{A,(t)}$, *where* $S_i^{A,(t)} = \{r \in [m] : \langle \mathbf{w}^{A,(t)}_{y_i,r}, \boldsymbol{\xi}_{i,1} \rangle > 0\}$.

3. $S_{j,r}^{A,(0)} \subseteq S_{j,r}^{A,(t)}$, *where* $S_{j,r}^{A,(t)} = \{i \in [N_1] : y_{i,1} = j, \langle \mathbf{w}^{A,(t)}_{j,r}, \boldsymbol{\xi}_{i,1} \rangle > 0\}$.

4. $\ell'^{(t)}_i / \ell'^{(t)}_k \le 13$.

5. *A refined estimation of* $\frac{1}{m} \sum_{r=1}^m \rho^{A,(t)}_{y_{i,1},r,i,1}$ *and* $\ell'^{(t)}_i$. *It holds that*

$$\underline{x}^A_t \le \frac{1}{m} \sum_{r=1}^m \bar{\rho}^{A,(t)}_{y_{i,1},r,i,1} \le \overline{x}^A_t + \overline{c}^A/(1 + \overline{b}^A),$$

$$\frac{1}{1 + \underline{b}^A e^{\underline{x}^A_t}} \le -\ell'^{(t)}_i \le \frac{1}{1 + \overline{b}^A e^{\overline{x}^A_t}},$$

*where* $\overline{x}^A_t, \underline{x}^A_t$ *are the the unique solution of*

$$\overline{x}^A_t + \overline{b}^A e^{\overline{x}^A_t} = \overline{c}^A t + \overline{b}^A,$$

$$\underline{x}^A_t + \underline{b}^A e^{\underline{x}^A_t} = \underline{c}^A t + \underline{b}^A,$$

*and* $\overline{b}^A = e^{-\kappa_A/2}, \overline{c}^A = \frac{3\eta\sigma^2_{p,1}d}{2N_1 m}, \underline{b}^A = e^{\kappa_A/2}$ *and* $\underline{c}^A = \frac{\eta\sigma^2_{p,1}d}{5N_1 m}$.

*Proof.* We prove it by induction. When $t = 0$, all results hold obviously. Now, we suppose there exists $\hat{t}$ and all the results hold for $t \le \hat{t} - 1$. Next, we prove these results hold at $t = \hat{t}$.

First, we prove the first result. With Lemma D.3, for $t \le \hat{t} - 1$, we have

$$-\kappa_A/2 \le y_{i,1}f(\mathbf{W}^{A,(t)}, \mathbf{x}_{i,1}) - \frac{1}{m} \sum_{r=1}^m \bar{\rho}^{A,(t)}_{j,r,i,1} \le \kappa_A/2,$$

$$-\kappa_A/2 \le y_{k,1}f(\mathbf{W}^{A,(t)}, \mathbf{x}_{k,1}) - \frac{1}{m} \sum_{r=1}^m \bar{\rho}^{A,(t)}_{j,r,k,1} \le \kappa_A/2.$$

By subtracting the two equations, we have

$$\left| \left[ y_{i,1}f(\mathbf{W}^{A,(t)}, \mathbf{x}_{i,1}) - y_{k,1}f(\mathbf{W}^{A,(t)}, \mathbf{x}_{k,1}) \right] - \left[ \frac{1}{m} \sum_{r=1}^m \bar{\rho}^{A,(t)}_{j,r,i,1} - \frac{1}{m} \sum_{r=1}^m \bar{\rho}^{A,(t)}_{j,r,k,1} \right] \right| \le \kappa_A. \tag{D.9}$$

When $\frac{1}{m}\sum_{r=1}^m \left[ \bar{\rho}_{y_{i,1},r,i,1}^{A,(\widehat{t}-1)} - \bar{\rho}_{r,k,i,1}^{A,(\widehat{t}-1)} \right] \leq \log(12) + \kappa_A$, we have

$$
\begin{aligned}
\frac{1}{m}\sum_{r=1}^m \left[ \bar{\rho}_{y_{i,1},r,i,1}^{A,(\widehat{t})} - \bar{\rho}_{y_{k,1},r,k,1}^{A,(\widehat{t})} \right] &= \frac{1}{m}\sum_{r=1}^m \left[ \bar{\rho}_{y_{i,1},r,i,1}^{A,(\widehat{t}-1)} - \bar{\rho}_{y_{k,1},r,k,1}^{A,(\widehat{t}-1)} \right] - \frac{\eta}{N_1 m}\cdot\frac{1}{m}\sum_{r=1}^m \left[ \ell_i'^{(\widehat{t}-1)}\cdot\sigma'\big(\langle \mathbf{w}_{y_{i,1},r}^{A,(\widehat{t}-1)}, \boldsymbol{\xi}_{i,1}\rangle\big) \right. \\
&\qquad \left. \cdot \|\boldsymbol{\xi}_{i,1}\|_2^2 - \ell_k'^{(\widehat{t}-1)}\cdot\sigma'\big(\langle \mathbf{w}_{y_{k,1},r}^{A,(\widehat{t}-1)}, \boldsymbol{\xi}_{k,1}\rangle\big)\cdot\|\boldsymbol{\xi}_{k,1}\|_2^2 \right] \qquad (D.10) \\
&\leq \frac{1}{m}\sum_{r=1}^m \left[ \bar{\rho}_{y_{i,1},r,i,1}^{A,(\widehat{t}-1)} - \bar{\rho}_{y_{k,1},r,k,1}^{A,(\widehat{t}-1)} \right] - \frac{\eta}{N_1 m}\cdot\frac{1}{m}\sum_{r=1}^m \ell_i'^{(\widehat{t}-1)} \\
&\qquad \cdot\sigma'\big(\langle \mathbf{w}_{y_{i,1},r}^{A,(\widehat{t}-1)}, \boldsymbol{\xi}_{i,1}\rangle\big)\cdot\|\boldsymbol{\xi}_{i,1}\|_2^2,
\end{aligned}
$$

where the first equality is by the update rule in Lemma B.2, the second inequality uses the fact $\ell_k'^{(\widehat{t}-1)} < 0$. Next, we bound the second term as

$$
\begin{aligned}
\left| \frac{\eta}{N_1 m}\cdot\frac{1}{m}\sum_{r=1}^m \ell_i'^{(\widehat{t}-1)}\cdot\sigma'\big(\langle \mathbf{w}_{y_{i,1},r}^{A,(\widehat{t}-1)}, \boldsymbol{\xi}_{i,1}\rangle\big)\cdot\|\boldsymbol{\xi}_{i,1}\|_2^2 \right| &\leq \frac{\eta}{N_1 m}\cdot\frac{1}{m}\sum_{r=1}^m |\ell_i'^{(\widehat{t}-1)}|\cdot\sigma'\big(\langle \mathbf{w}_{y_{i,1},r}^{A,(\widehat{t}-1)}, \boldsymbol{\xi}_{i,1}\rangle\big)\cdot\|\boldsymbol{\xi}_{i,1}\|_2^2 \\
&\leq \frac{\eta}{N_1 m^2}\cdot|S_i^{A,(\widehat{t}-1)}|\cdot\|\boldsymbol{\xi}_{i,1}\|_2^2 \\
&\leq \frac{\eta\sigma_{p,1}^2 d}{2N_1 m} \\
&\leq \sqrt{\log(2N_1/\delta)/m},
\end{aligned}
$$

where the first inequality is by triangle inequality, the second inequality uses the fact $-1 < \ell_i'^{(\widehat{t}-1)} < 0$ and the definition of $S_i^{A,(\widehat{t}-1)}$, the third inequality is by Lemma C.1, and the forth inequality is by the condition of $\eta$ in Condition 4.1. Therefore, we have

$$
\begin{aligned}
\frac{1}{m}\sum_{r=1}^m \left[ \bar{\rho}_{y_{i,1},r,i,1}^{A,(\widehat{t})} - \bar{\rho}_{y_{k,1},r,k,1}^{A,(\widehat{t})} \right] &\leq \frac{1}{m}\sum_{r=1}^m \left[ \bar{\rho}_{y_{i,1},r,i,1}^{A,(\widehat{t}-1)} - \bar{\rho}_{y_{k,1},r,k,1}^{A,(\widehat{t}-1)} \right] + \sqrt{\log(2N_1/\delta)/m} \\
&\leq \log(12) + \kappa_A + \sqrt{\log(2N_1/\delta)/m}.
\end{aligned}
$$

On the other side, When $\frac{1}{m}\sum_{r=1}^m \left[ \bar{\rho}_{y_{i,1},r,i,1}^{A,(\widehat{t}-1)} - \bar{\rho}_{r,k,i,1}^{A,(\widehat{t}-1)} \right] \geq \log(12) + \kappa_A$, with equation D.9, we have

$$
\begin{aligned}
y_{i,1}f(\mathbf{W}^{A,(\widehat{t}-1)}, \mathbf{x}_{i,1}) - y_{k,1}f(\mathbf{W}^{A,(\widehat{t}-1)}, \mathbf{x}_{k,1}) &\geq \frac{1}{m}\sum_{r=1}^m \left[ \bar{\rho}_{y_{i,1},r,i,1}^{A,(\widehat{t}-1)} - \bar{\rho}_{r,k,i,1}^{A,(\widehat{t}-1)} \right] - \kappa_A \\
&\geq \log(12),
\end{aligned}
$$

where the first inequality uses equation D.9. Then, it holds that

$$
\frac{-\ell_i'^{(\widehat{t}-1)}}{-\ell_k'^{(\widehat{t}-1)}} \leq e^{-y_{i,1}f(\mathbf{W}^{A,(\widehat{t}-1)}, \mathbf{x}_{i,1}) + y_{k,1}f(\mathbf{W}^{A,(\widehat{t}-1)}, \mathbf{x}_{k,1})} < \frac{1}{12}. \qquad (D.11)
$$

Then, we have

$$
\begin{aligned}
\frac{-\sum_{r=1}^m \ell_i'^{(\widehat{t}-1)}\cdot\sigma'\big(\langle \mathbf{w}_{y_{i,1},r}^{A,(\widehat{t}-1)}, \boldsymbol{\xi}_{i,1}\rangle\big)\cdot\|\boldsymbol{\xi}_{i,1}\|_2^2}{-\sum_{r=1}^m \ell_k'^{(\widehat{t}-1)}\cdot\sigma'\big(\langle \mathbf{w}_{y_{k,1},r}^{A,(\widehat{t}-1)}, \boldsymbol{\xi}_{k,1}\rangle\big)\cdot\|\boldsymbol{\xi}_{k,1}\|_2^2} &= \frac{-\ell_i'^{(\widehat{t}-1)}\cdot|S_i^{A,(\widehat{t}-1)}|\cdot\|\boldsymbol{\xi}_{i,1}\|_2^2}{-\ell_k'^{(\widehat{t}-1)}\cdot|S_k^{A,(\widehat{t}-1)}|\cdot\|\boldsymbol{\xi}_{k,1}\|_2^2} \\
&< \frac{1}{4}\cdot\frac{|S_i^{A,(\widehat{t}-1)}|}{|S_k^{A,(\widehat{t}-1)}|} \\
&\leq 1,
\end{aligned}
$$

where the first inequality uses equation D.11 and Lemma C.1, and the second inequality uses the fact that $|S_i^{\widehat{t}-1}| \leq m$, the induction $|S_k^0| \leq |S_k^{A,(\widehat{t}-1)}|$ and $|S_k^{A,(0)}| \geq m/4$. Then, with equation D.10, it holds that

$$
\begin{aligned}
\frac{1}{m}\sum_{r=1}^m \left[ \bar{\rho}_{y_{i,1},r,i,1}^{A,(\widehat{t})} - \bar{\rho}_{y_{k,1},r,k,1}^{A,(\widehat{t})} \right] &\leq \frac{1}{m}\sum_{r=1}^m \left[ \bar{\rho}_{y_{i,1},r,i,1}^{A,(\widehat{t}-1)} - \bar{\rho}_{y_{k,1},r,k,1}^{A,(\widehat{t}-1)} \right] \\
&\leq \log(12) + \kappa_A + \sqrt{\log(2N_1/\delta)/m}.
\end{aligned}
$$

Next, we prove the second result and the third result together. When $j = y_{i,1}$, by Lemma B.2, it hols that

$$\langle \mathbf{w}_{j,r}^{A,(\widehat{t})}, \boldsymbol{\xi}_{i,1} \rangle = \langle \mathbf{w}_{j,r}^{A,(\widehat{t}-1)}, \boldsymbol{\xi}_{i,1} \rangle - \frac{\eta}{N_1 m} \sum_{i' \in [N_1]} \ell_{i'}^{\prime(\widehat{t}-1)} \cdot \sigma'\big(\langle \mathbf{w}_{j,r}^{A,(\widehat{t}-1)}, \boldsymbol{\xi}_{i',1} \rangle\big) \cdot \langle \boldsymbol{\xi}_{i',1}, \boldsymbol{\xi}_{i,1} \rangle$$

$$= \langle \mathbf{w}_{j,r}^{A,(\widehat{t}-1)}, \boldsymbol{\xi}_{i,1} \rangle - \frac{\eta}{N_1 m} \ell_i^{\prime(\widehat{t}-1)} \cdot \sigma'\big(\langle \mathbf{w}_{j,r}^{A,(\widehat{t}-1)}, \boldsymbol{\xi}_{i,1} \rangle\big) \cdot \|\boldsymbol{\xi}_{i,1}\|_2^2$$

$$- \frac{\eta}{N_1 m} \sum_{i' \neq i} \ell_{i'}^{\prime(\widehat{t}-1)} \cdot \sigma'\big(\langle \mathbf{w}_{j,r}^{A,(\widehat{t}-1)}, \boldsymbol{\xi}_{i',1} \rangle\big) \cdot \langle \boldsymbol{\xi}_{i',1}, \boldsymbol{\xi}_{i,1} \rangle$$

$$\geq \langle \mathbf{w}_{j,r}^{A,(\widehat{t}-1)}, \boldsymbol{\xi}_{i,1} \rangle + \frac{\eta \sigma_{p,1}^2 d}{2 N_1 m} \ell_i^{\prime(\widehat{t}-1)} - \frac{26 \eta \sigma_{p,1}^2 \sqrt{d \log(4 N_1^2/\delta)}}{m} \ell_i^{\prime(\widehat{t}-1)}$$

$$\geq \langle \mathbf{w}_{j,r}^{A,(\widehat{t}-1)}, \boldsymbol{\xi}_{i,1} \rangle,$$

where the first inequality is by Lemma C.1 and the induction $\ell_k^{\prime(\widehat{t}-1)}/\ell_i^{\prime(\widehat{t}-1)} \leq 13$, and the second inequality is by the condition of $d$ in Condition 4.1. Then, we know that $S_i^{A,(0)} \subseteq S_i^{A,(\widehat{t}-1)} \subseteq S_i^{A,(\widehat{t})}$ and $S_{j,r}^{A,(0)} \subseteq S_{j,r}^{A,(\widehat{t}-1)} \subseteq S_{j,r}^{A,(\widehat{t})}$ by induction.

Next, we prove the forth result. With equation D.9, it holds that

$$\frac{\ell_i^{\prime(\widehat{t})}}{\ell_k^{\prime(\widehat{t})}} \leq e^{-y_{i,1} f(\mathbf{W}^{A,(\widehat{t})}, \mathbf{x}_{i,1}) + y_{k,1} f(\mathbf{W}^{A,(\widehat{t})}, \mathbf{x}_{k,1})}$$

$$\leq e^{-\frac{1}{m} \sum_{r=1}^m \bar{\rho}_{j,r,i,1}^{A,(\widehat{t})} + \frac{1}{m} \sum_{r=1}^m \bar{\rho}_{j,r,k,1}^{A,(\widehat{t})} + \kappa_A}$$

$$\leq e^{\log(12) + 2\kappa_A + \sqrt{\log(2 N_1/\delta)/m}} = 12 + o(1) \leq 13.$$

Next, we prove the fifth result. From Lemma B.2, we know that

$$\frac{1}{m} \sum_{r=1}^m \overline{\rho}_{y_{i,1},r,i,1}^{A,(\widehat{t})} = \frac{1}{m} \sum_{r=1}^m \overline{\rho}_{y_{i,1},r,i,1}^{A,(\widehat{t}-1)} - \frac{\eta}{N_1 m} \cdot \frac{1}{m} \sum_{r=1}^m \ell_i^{\prime(\widehat{t}-1)} \cdot \sigma'\big(\langle \mathbf{w}_{y_{i,1},r}^{A,(t)}, \boldsymbol{\xi}_{i,1} \rangle\big) \cdot \|\boldsymbol{\xi}_{i,1}\|_2^2$$

$$= \frac{1}{m} \sum_{r=1}^m \overline{\rho}_{y_{i,1},r,i,1}^{A,(\widehat{t}-1)} - \frac{\eta}{N_1 m} \cdot \frac{|S_i^{A,(\widehat{t}-1)}|}{m} \cdot \ell_i^{\prime(\widehat{t}-1)} \cdot \|\boldsymbol{\xi}_{i,1}\|_2^2.$$

Here, with Lemma D.3, the gradient $\ell_i^{\prime(\widehat{t}-1)}$ can be bounded as

$$\frac{-1}{1 + e^{\frac{1}{m} \sum_{r=1}^m \bar{\rho}_{y_{i,1},r,i,1}^{A,(\widehat{t}-1)} - \kappa_A/2}} \leq \ell_i^{\prime(\widehat{t}-1)} = \frac{-1}{1 + e^{y_{i,1} f(\mathbf{W}^{A,(\widehat{t}-1)}, \mathbf{x}_{i,1})}} \leq \frac{-1}{1 + e^{\frac{1}{m} \sum_{r=1}^m \bar{\rho}_{y_{i,1},r,i,1}^{A,(\widehat{t}-1)} + \kappa_A/2}}.$$

Then, we have

$$\frac{1}{m} \sum_{r=1}^m \overline{\rho}_{y_{i,1},r,i,1}^{A,(\widehat{t})} \leq \frac{1}{m} \sum_{r=1}^m \overline{\rho}_{y_{i,1},r,i,1}^{A,(\widehat{t}-1)} + \frac{\eta}{N_1 m} \cdot \frac{|S_i^{A,(\widehat{t}-1)}|}{m} \cdot \frac{1}{1 + e^{\frac{1}{m} \sum_{r=1}^m \bar{\rho}_{y_{i,1},r,i,1}^{A,(\widehat{t}-1)} - \kappa_A/2}} \cdot \|\boldsymbol{\xi}_{i,1}\|_2^2$$

$$\leq \frac{1}{m} \sum_{r=1}^m \overline{\rho}_{y_{i,1},r,i,1}^{A,(\widehat{t}-1)} + \frac{3\eta \sigma_{p,1}^2 d}{2 N_1 m} \cdot \frac{1}{1 + e^{\frac{1}{m} \sum_{r=1}^m \bar{\rho}_{y_{i,1},r,i,1}^{A,(\widehat{t}-1)} - \kappa_A/2}};$$

$$\frac{1}{m} \sum_{r=1}^m \overline{\rho}_{y_{i,1},r,i,1}^{A,(\widehat{t})} \geq \frac{1}{m} \sum_{r=1}^m \overline{\rho}_{y_{i,1},r,i,1}^{A,(\widehat{t}-1)} + \frac{\eta}{N_1 m} \cdot \frac{|S_i^{A,(\widehat{t}-1)}|}{m} \cdot \frac{1}{1 + e^{\frac{1}{m} \sum_{r=1}^m \bar{\rho}_{y_{i,1},r,i,1}^{A,(\widehat{t}-1)} + \kappa_A/2}} \cdot \|\boldsymbol{\xi}_{i,1}\|_2^2$$

$$\geq \frac{1}{m} \sum_{r=1}^m \overline{\rho}_{y_{i,1},r,i,1}^{A,(\widehat{t}-1)} + \frac{\eta \sigma_{p,1}^2 d}{5 N_1 m} \cdot \frac{1}{1 + e^{\frac{1}{m} \sum_{r=1}^m \bar{\rho}_{y_{i,1},r,i,1}^{A,(\widehat{t}-1)} + \kappa_A/2}}.$$

So, the estimation of $\frac{1}{m} \sum_{r=1}^m \overline{\rho}_{y_{i,1},r,i,1}^{A,(\widehat{t})}$ can be approximated by solving the continuous-time iterative equation

$$\frac{dx_t^A}{dt} = \frac{a}{1 + b e^{x_t^A}} \quad and \quad x_0 = 0.$$

The result is shown in Lemma C.4. For the gradient counterparts, with Lemma D.3, the gradient $\ell_i'^{(\widehat{t}-1)}$ can be bounded as

$$\frac{1}{1 + e^{\frac{1}{m}\sum_{r=1}^{m}\bar{\rho}_{y_{i,1},r,i,1}^{A,(\widehat{t}-1)}+\kappa_A/2}} \leq -\ell_i'^{(\widehat{t}-1)} = \frac{1}{1 + e^{y_{i,1}f(\mathbf{W}^{A,(\widehat{t}-1)},\mathbf{x}_{i,1})}} \leq \frac{1}{1 + e^{\frac{1}{m}\sum_{r=1}^{m}\bar{\rho}_{y_{i,1},r,i,1}^{A,(\widehat{t}-1)}-\kappa_A/2}}.$$

The result is obvious since that $1/m\sum_{r=1}^{m}\bar{\rho}_{y_{i,1},r,i,1}^{A,(\widehat{t}-1)}$ is bounded. Since then we complete the proof. $\qquad\square$

*Proof of Proposition D.1.* We prove it by induction. When $t = 0$, all results hold obviously. Now, we suppose there exists $\widehat{t}$ and all the results hold for $t \leq \widehat{t} - 1$. Next, we prove these results hold at $t = \widehat{t}$.

First, for the first result, when $j \neq y_{i,1}$, we have $\bar{\rho}_{j,r,i,1}^{A,(\widehat{t})} = 0$. When $j = y_{i,1}$, by the update rule, it holds that

$$\bar{\rho}_{j,r,i,1}^{A,(\widehat{t})} = \bar{\rho}_{j,r,i,1}^{A,(\widehat{t}-1)} - \frac{\eta}{N_1 m}\ell_i'^{(\widehat{t}-1)} \cdot \sigma'\big(\langle\mathbf{w}_{j,r}^{A,(\widehat{t}-1)}, \boldsymbol{\xi}_{i,1}\rangle\big) \cdot \|\boldsymbol{\xi}_{i,1}\|_2^2. \tag{D.12}$$

If $\bar{\rho}_{j,r,i,1}^{A,(\widehat{t}-1)} \leq 2\log(T^*)$, we have

$$\begin{aligned}
\bar{\rho}_{j,r,i,1}^{A,(\widehat{t})} &\leq \bar{\rho}_{j,r,i,1}^{A,(\widehat{t}-1)} + \frac{\eta}{N_1 m}\frac{3\sigma_{p,1}^2 d}{2} \\
&\leq 2\log(T^*) + \log(T^*) \leq 4\log(T^*),
\end{aligned}$$

where the first inequality uses the fact $-1 \leq \ell_i'^{(\widehat{t}-1)} \leq 0$ and Lemma C.1, and the second inequality is by the condition of $\eta$ in Condition 4.1. If $\bar{\rho}_{j,r,i,1}^{A,(\widehat{t}-1)} \geq 2\log(T^*)$, from equation D.12 we know that $\bar{\rho}_{j,r,i,1}^{A,(t)}$ increases with $t$. Therefore, suppose that $t_{j,r,i,1}$ is the last time satisfying $\bar{\rho}_{j,r,i,1}^{A,(t_{j,r,i,1})} \leq 2\log(T^*)$. Now, we want to show that the increment of $\bar{\rho}$ from $t_{j,r,i,1}$ to $\widehat{t}$ does not exceed $2\log(T^*)$.

$$\begin{aligned}
\bar{\rho}_{j,r,i,1}^{A,(\widehat{t})} = \bar{\rho}_{j,r,i,1}^{A,(t_{j,r,i,1})} &- \frac{\eta}{N_1 m}\ell_i'^{(t_{j,r,i,1})} \cdot \sigma'\big(\langle\mathbf{w}_{j,r}^{A,(t_{j,r,i,1})}, \boldsymbol{\xi}_{i,1}\rangle\big) \cdot \|\boldsymbol{\xi}_{i,1}\|_2^2 \\
&- \sum_{t_{j,r,i,1}<t\leq\widehat{t}-1} \frac{\eta}{N_1 m}\ell_i'^{(t)} \cdot \sigma'\big(\langle\mathbf{w}_{j,r}^{A,(t)}, \boldsymbol{\xi}_{i,1}\rangle\big) \cdot \|\boldsymbol{\xi}_{i,1}\|_2^2.
\end{aligned} \tag{D.13}$$

Here, the second term can be bounded as

$$\left|\frac{\eta}{N_1 m}\ell_i'^{(t_{j,r,i,1})} \cdot \sigma'\big(\langle\mathbf{w}_{j,r}^{A,(t_{j,r,i,1})}, \boldsymbol{\xi}_{i,1}\rangle\big) \cdot \|\boldsymbol{\xi}_{i,1}\|_2^2\right| \leq \frac{3\eta\sigma_{p,1}^2 d}{2N_1 m} \leq \log(T^*),$$

where the first inequality is by Lemma C.1 and the second inequality is by the condition of $\eta$ in Condition 4.1. For the third term, note that when $t > t_{j,r,i,1}$,

$$\begin{aligned}
\langle\mathbf{w}_{y_{i,1},r}^{A,(t)}, \boldsymbol{\xi}_{i,1}\rangle &\geq \langle\mathbf{w}_{y_{i,1},r}^{A,(0)}, \boldsymbol{\xi}_{i,1}\rangle + \bar{\rho}_{j,r,i,1}^{A,(\widehat{t})} - 4N_1\sqrt{\frac{\log(4N_1^2/\delta)}{d}}\log(T^*) \\
&\geq -2\sqrt{\log(12mN_1/\delta)} \cdot \sigma_0\sigma_{p,1}\sqrt{d} + 2\log(T^*) - 4N_1\sqrt{\frac{\log(4N_1^2/\delta)}{d}}\log(T^*) \\
&\geq 1.8\log(T^*),
\end{aligned} \tag{D.14}$$

where the first inequality is by Lemma D.2, the second inequality is by Lemma C.2 and the third inequality is by $\sqrt{\log(12mN_1/\delta)} \cdot \sigma_0\sigma_{p,1}\sqrt{d} \leq 0.1\log(T^*), 4N_1\sqrt{\frac{\log(4N_1^2/\delta)}{d}}\log(T^*) \leq 0.1\log(T^*)$ from the Condition 4.1. Then, the gradient can be bounded as

$$\begin{aligned}
|\ell_i^{(t)}| &= \frac{1}{1 + e^{-y_{i,1}[F_{+1}(\mathbf{W}_{+1}^{A,(t)},\mathbf{x}_{i,1})-F_{-1}(\mathbf{W}_{-1}^{A,(t)},\mathbf{x}_{i,1})]}} \\
&\leq e^{-y_{i,1}F_{y_{i,1}}(\mathbf{W}_{+1}^{A,(t)},\mathbf{x}_{i,1})+0.1} \\
&= e^{-\frac{1}{m}\sum_{r=1}^{m}\sigma(\langle\mathbf{w}_{y_{i,1},r}^{A,(t)},\boldsymbol{\xi}_{i,1}\rangle)+0.1} \\
&\leq e^{0.1} \cdot e^{-1.8\log(T^*)} \leq 2e^{-1.8\log(T^*)},
\end{aligned}$$

where the first inequality is by Lemma D.3 that $\kappa_A \le 0.2$, the second inequality is by equation D.14. Based on these results, we can bound the third term in equation D.13 as

$$\left| \sum_{t_{j,r,i,1} < t \le \hat{t}-1} \frac{\eta}{N_1 m} \ell_i'^{(t)} \cdot \sigma'\big(\langle \mathbf{w}_{j,r}^{A,(t)}, \boldsymbol{\xi}_{i,1}\rangle\big) \cdot \|\boldsymbol{\xi}_{i,1}\|_2^2 \right| \le \frac{\eta T^*}{N_1 m} \cdot 2e^{-1.8\log(T^*)} \cdot \frac{3\sigma_{p,1}^2 d}{2}$$

$$\le \frac{T^*}{(T^*)^{1.8}} \cdot \frac{3\eta \sigma_{p,1}^2 d}{N_1 m}$$

$$\le 1 \le \log(T^*),$$

where the first inequality is by the bound of $|\ell_i^{(t)}|$ and Lemma C.1, the second inequality is by the fact that $e^{-x} \le 1/x, x > 0$ and the third inequality is by the selection of $\eta$ in Condition 4.1. Since then, we prove that $\overline{\rho}_{j,r,i,1}^{A,(\hat{t})} \le 4\log(T^*)$.

Next, we prove the second result. When $j = y_{i,2}$, we have $\underline{\rho}_{-j,r,i,1}^{A,(\hat{t})} = 0$. If $\underline{\rho}_{-j,r,i,1}^{A,(\hat{t}-1)} \le -2\sqrt{\log\left(\frac{12mN_1}{\delta}\right)} \cdot \sigma_0\sigma_{p,1}\sqrt{d} - (C_1 - 4)N_1\sqrt{\frac{\log\left(\frac{4N_1^2}{\delta}\right)}{d}}\log(T^*)$, by Lemma D.2, it holds that

$$\left| \langle \mathbf{w}_{j,r}^{A,(\hat{t}-1)} - \mathbf{w}_{j,r}^{A,(0)}, \boldsymbol{\xi}_{i,1}\rangle - \underline{\rho}_{-j,r,i,1}^{A,(\hat{t}-1)} \right| \le 4N_1\sqrt{\frac{\log(4N_1^2/\delta)}{d}}\log(T^*).$$

Rearrange the inequality, we get

$$\langle \mathbf{w}_{j,r}^{A,(\hat{t}-1)}, \boldsymbol{\xi}_{i,1}\rangle \le \langle \mathbf{w}_{j,r}^{A,(0)}, \boldsymbol{\xi}_{i,1}\rangle + \underline{\rho}_{-j,r,i,1}^{A,(\hat{t}-1)} + 4N_1\sqrt{\frac{\log(4N_1^2/\delta)}{d}}\log(T^*)$$

$$\le 0.$$

Then, by the update rule, it holds that

$$\rho_{-j,r,i,1}^{A,(\hat{t})} = \rho_{-j,r,i,1}^{A,(\hat{t}-1)} + \frac{\eta}{N_1 m}\ell_i'^{(\hat{t}-1)} \cdot \sigma'\big(\langle \mathbf{w}_{j,r}^{A,(\hat{t}-1)}, \boldsymbol{\xi}_{i,1}\rangle\big) \cdot \|\boldsymbol{\xi}_{i,1}\|_2^2$$

$$= \underline{\rho}_{-j,r,i,1}^{A,(\hat{t}-1)} \ge -2\sqrt{\log\left(\frac{12mN_1}{\delta}\right)} \cdot \sigma_0\sigma_{p,1}\sqrt{d} - C_1 N_1\sqrt{\frac{\log\left(\frac{4N_1^2}{\delta}\right)}{d}}\log(T^*).$$

If $\underline{\rho}_{-j,r,i,1}^{A,(\hat{t}-1)} \ge -2\sqrt{\log\left(\frac{12mN_1}{\delta}\right)} \cdot \sigma_0\sigma_{p,1}\sqrt{d} - (C_1 - 4)N_1\sqrt{\frac{\log\left(\frac{4N_1^2}{\delta}\right)}{d}}e\log(T^*)$, by the update rule, it holds that

$$\rho_{-j,r,i,1}^{A,(\hat{t})} = \rho_{-j,r,i,1}^{A,(\hat{t}-1)} + \frac{\eta}{N_1 m}\ell_i'^{(\hat{t}-1)} \cdot \sigma'\big(\langle \mathbf{w}_{j,r}^{A,(\hat{t}-1)}, \boldsymbol{\xi}_{i,1}\rangle\big) \cdot \|\boldsymbol{\xi}_{i,1}\|_2^2$$

$$\ge \rho_{-j,r,i,1}^{A,(\hat{t}-1)} - \frac{3\eta\sigma_{p,1}^2 d}{2N_1 m}$$

$$\ge -2\sqrt{\log\left(\frac{12mN_1}{\delta}\right)} \cdot \sigma_0\sigma_{p,1}\sqrt{d} - C_1 N_1\sqrt{\frac{\log\left(\frac{4N_1^2}{\delta}\right)}{d}}\log(T^*),$$

where the first inequality uses the fact $-1 \le \ell_i'^{(\hat{t}-1)} \le 0$ and Lemma C.1, and the second inequality is by the condition of $\eta$ in Condition 4.1.

Next, we prove the third result. We prove a stronger conclusion that for any $i^* \in S_{j,r}^{A,(0)}$, it holds that

$$\frac{\gamma_{j,r}^{A,(t)}}{\overline{\rho}_{j,r,i^*}^{A,(t)}} \le \frac{26N_1\|\mathbf{u}\|_2^2}{\sigma_{p,1}^2 d}.$$

Recall the update rule that

$$\gamma_{j,r}^{A,(\hat{t})} = \gamma_{j,r}^{A,(\hat{t}-1)} - \frac{\eta}{N_1 m} \sum_{i \in [N_1]} \ell_i'^{(\hat{t}-1)} \cdot \sigma'\big(\langle \mathbf{w}_{j,r}^{A,(\hat{t}-1)}, y_{i,1} \cdot (\mathbf{u} + \mathbf{v}_1)\rangle\big) \cdot \|\mathbf{u}\|_2^2$$

$$\leq \gamma_{j,r}^{A,(\hat{t}-1)} - \frac{\eta}{N_1 m} \cdot 13n \cdot \ell_i'^{A,(\hat{t}-1)} \cdot \sigma'\big(\langle \mathbf{w}_{j,r}^{A,(\hat{t}-1)}, y_{i,1} \cdot (\mathbf{u} + \mathbf{v}_1)\rangle\big) \cdot \|\mathbf{u}\|_2^2,$$

where the inequality follows by $\ell_i'^{(t)}/\ell_k'^{(t)} \leq 13$ in Lemma D.4, and

$$\bar{\rho}_{j,r,i^*,1}^{A,(\hat{t})} = \bar{\rho}_{j,r,i^*,1}^{A,(\hat{t}-1)} - \frac{\eta}{N_1 m} \ell_{i^*}'^{A,(\hat{t}-1)} \cdot \sigma'\big(\langle \mathbf{w}_{j,r}^{A,(\hat{t}-1)}, \boldsymbol{\xi}_{i^*,1}\rangle\big) \cdot \|\boldsymbol{\xi}_{i^*,1}\|_2^2 \cdot \mathbf{1}\{y_{i^*,1} = j\}.$$

Compare the gradient, we have

$$\frac{\gamma_{j,r}^{A,(\hat{t})}}{\bar{\rho}_{j,r,i^*,1}^{A,(\hat{t})}} \leq \max\left\{ \frac{\gamma_{j,r}^{A,(\hat{t}-1)}}{\bar{\rho}_{j,r,i^*,1}^{A,(\hat{t}-1)}}, \frac{13N_1 \cdot \ell_{i^*}'^{(\hat{t}-1)} \cdot \sigma'\big(\langle \mathbf{w}_{j,r}^{A,(\hat{t}-1)}, y_{i^*,1} \cdot (\mathbf{u} + \mathbf{v}_1)\rangle\big) \cdot \|\mathbf{u}\|_2^2}{\ell_{i^*}'^{(\hat{t}-1)} \cdot \sigma'\big(\langle \mathbf{w}_{j,r}^{A,(\hat{t}-1)}, \boldsymbol{\xi}_{i^*,1}\rangle\big) \cdot \|\boldsymbol{\xi}_{i^*,1}\|_2^2} \right\}$$

$$\leq \max\left\{ \frac{\gamma_{j,r}^{A,(\hat{t}-1)}}{\bar{\rho}_{j,r,i^*,1}^{A,(\hat{t}-1)}}, \frac{13N_1 \|\mathbf{u}\|_2^2}{\|\boldsymbol{\xi}_{i^*,1}\|_2^2} \right\}$$

$$\leq \max\left\{ \frac{\gamma_{j,r}^{A,(\hat{t}-1)}}{\bar{\rho}_{j,r,i^*,1}^{A,(\hat{t}-1)}}, \frac{26N_1 \|\mathbf{u}\|_2^2}{\sigma_{p,1}^2 d} \right\}$$

$$\leq \frac{26N_1 \|\mathbf{u}\|_2^2}{\sigma_{p,1}^2 d},$$

where the first inequality is from two update rules, the second inequality is by $i^* \in S_{j,r}^{A,(0)}$, the third inequality is by Lemma C.1 and the last inequality use the induction $\frac{\gamma_{j,r}^{A,(\hat{t}-1)}}{\bar{\rho}_{j,r,i^*,1}^{A,(\hat{t}-1)}} \leq \frac{26N_1 \|\mathbf{u}\|_2^2}{\sigma_{p,1}^2 d}$. Similarly, it holds that $\frac{\gamma_{j,r,1}^{A,(\hat{t})}}{\bar{\rho}_{j,r,i^*,1}^{A,(\hat{t})}} \leq \frac{26N_1 \|\mathbf{v}_1\|_2^2}{\sigma_{p,1}^2 d}$. $\qquad\square$

**Proposition D.5.** *Under Condition 4.1, for $0 \leq t \leq T^*$, it holds that*

$$0 \leq \bar{\rho}_{j,r,i,1}^{A,(t)} \leq 4\log(T^*), \tag{D.15}$$

$$0 \geq \underline{\rho}_{j,r,i,1}^{A,(t)} \geq -2\sqrt{\log\left(\frac{12mN_1}{\delta}\right)} \cdot \sigma_0 \sigma_{p,1} \sqrt{d} - C_1 N_1 \sqrt{\frac{\log\left(\frac{4N_1^2}{\delta}\right)}{d}} \log(T^*) \geq -4\log(T^*), \tag{D.16}$$

$$0 \leq \gamma_{j,r}^{A,(t)} \leq \frac{C_2 N_1 \|\mathbf{u}\|_2^2}{\sigma_{p,1}^2 d} \log(T^*), \tag{D.17}$$

$$0 \leq \gamma_{j,r,1}^{A,(t)} \leq \frac{C_2 N_1 \|\mathbf{v}_1\|_2^2}{\sigma_{p,1}^2 d} \log(T^*), \tag{D.18}$$

*for all $r \in [m], j \in \{\pm 1\}, i \in [N_1]$, where $C_1$ and $C_2$ are two absolute constant. Besides, we also have the following results:*

1. $\frac{1}{m} \sum_{r=1}^m \left[ \rho_{y_{i,1},r,i,1}^{A,(t)} - \bar{\rho}_{r,k,i,1}^{A,(t)} \right] \leq \log(12) + \kappa_A + \sqrt{\log(2N_1/\delta)/m}$ *for all $i, k \in [N_1]$.*

2. $S_i^{A,(0)} \subseteq S_i^{A,(t)}$, *where $S_i^{A,(t)} = \{r \in [m] : \langle \mathbf{w}_{y_i,r}^{A,(t)}, \boldsymbol{\xi}_{i,1}\rangle > 0\}$.*

3. $S_{j,r}^{A,(0)} \subseteq S_{j,r}^{A,(t)}$, *where $S_{j,r}^{A,(t)} = \{i \in [N_1] : y_{i,1} = j, \langle \mathbf{w}_{j,r}^{A,(t)}, \boldsymbol{\xi}_{i,1}\rangle > 0\}$.*

4. $\ell_i'^{(t)}/\ell_k'^{(t)} \leq 13$.

5. *A refined estimation of $\frac{1}{m}\sum_{r=1}^{m}\rho_{y_{i,1},r,i,1}^{A,(t)}$ and $\ell_i'^{(t)}$. It holds that*

$$\underline{x}_t^A \leq \frac{1}{m}\sum_{r=1}^{m}\bar{\rho}_{y_{i,1},r,i,1}^{A,(t)} \leq \overline{x}_t^A + \overline{c}^A/(1+\overline{b}^A),$$

$$\frac{1}{1+\underline{b}^A e^{\underline{x}_t^A}} \leq -\ell_i'^{(t)} \leq \frac{1}{1+\overline{b}^A e^{\overline{x}_t^A}},$$

*where $\overline{x}_t^A, \underline{x}_t^A$ are the unique solution of*

$$\overline{x}_t^A + \overline{b}^A e^{\overline{x}_t^A} = \overline{c}^A t + \overline{b}^A,$$

$$\underline{x}_t^A + \underline{b}^A e^{\underline{x}_t^A} = \underline{c}^A t + \underline{b}^A,$$

*and $\overline{b}^A = e^{-\kappa_A/2}, \overline{c}^A = \frac{3\eta\sigma_{p,1}^2 d}{2N_1 m}, \underline{b}^A = e^{\kappa_A/2}$ and $\underline{c}^A = \frac{\eta\sigma_{p,1}^2 d}{5N_1 m}$.*

**Lemma D.6** (Meng et al. (2024)). *It holds that*

$$\log\left(\frac{\eta\sigma_{p,1}^2 d}{8N_1 m}t + \frac{2}{3}\right) \leq \overline{x}_t^A \leq \log\left(\frac{2\eta\sigma_{p,1}^2 d}{N_1 m}t + 1\right),$$

$$\log\left(\frac{\eta\sigma_{p,1}^2 d}{8N_1 m}t + \frac{2}{3}\right) \leq \underline{x}_t^A \leq \log\left(\frac{2\eta\sigma_{p,1}^2 d}{N_1 m}t + 1\right),$$

*for the defined $\overline{b}^A, \overline{c}^A, \underline{b}^A, \underline{c}^A$.*

## D.2 SIGNAL LEARNING AND NOISE MEMORIZATION

In this part, we will give detailed analysis of signal learning and noise memorization.

**Lemma D.7.** *Under Condition 4.1, for $0 \leq t \leq T^*$, $\langle \mathbf{w}_{j,r}^{A,(t)}, j(\mathbf{u}+\mathbf{v})\rangle$ increases with $t$.*

*Proof.* By Lemma B.1, it holds that

$$\langle \mathbf{w}_{j,r}^{A,(t)}, j(\mathbf{u}+\mathbf{v})\rangle = \gamma_{j,r}^{A,(t)} + \gamma_{j,r,1}^{A,(t)}.$$

By the update rule in Lemma B.2, we know that $\gamma_{j,r}^{A,(t)}$ and $\gamma_{j,r,1}^{A,(t)}$ increase with $t$. So $\langle \mathbf{w}_{j,r}^{A,(t)}, j(\mathbf{u}+\mathbf{v})\rangle$ increases with $t$. $\square$

**Lemma D.8.** *Under Condition 4.1, for $0 \leq t \leq T^*$, it holds that*

$$\frac{\eta\|\mathbf{u}\|_2^2}{\overline{c}m}\overline{x}_{t-2}^A - \frac{2\eta\|\mathbf{u}\|_2^2}{m} \leq \gamma_{j,r}^{A,(t)} \leq \frac{\eta\|\mathbf{u}\|_2^2}{\underline{c}m}\underline{x}_{t-1}^A - \frac{2\eta\|\mathbf{u}\|_2^2}{m},$$

$$\frac{\eta\|\mathbf{v}_1\|_2^2}{\overline{c}m}\overline{x}_{t-2}^A - \frac{2\eta\|\mathbf{v}_1\|_2^2}{m} \leq \gamma_{j,r,1}^{A,(t)} \leq \frac{\eta\|\mathbf{v}_1\|_2^2}{\underline{c}m}\underline{x}_{t-1}^A - \frac{2\eta\|\mathbf{v}_1\|_2^2}{m}.$$

*Proof.* By the update rule, it holds that

$$\gamma_{j,r}^{A,(t+1)} + \gamma_{j,r,1}^{A,(t+1)} = \gamma_{j,r}^{A,(t)} + \gamma_{j,r,1}^{A,(t)} - \frac{\eta}{N_1 m} \sum_{i'=1}^{N_1} \ell_{i'}^{\prime(t)} \cdot \sigma'\big(\langle \mathbf{w}_{j,r}^{A,(t)}, y_i(\mathbf{u}+\mathbf{v}_1)\rangle\big)\|\mathbf{u}+\mathbf{v}_1\|_2^2$$

$$\leq \gamma_{j,r}^{A,(t)} + \gamma_{j,r,1}^{A,(t)} + \frac{\eta\|\mathbf{u}+\mathbf{v}_1\|_2^2}{m} \frac{1}{1+\underline{b}^A e^{\underline{x}_t^A}}$$

$$\leq \gamma_{j,r}^{A,(0)} + \gamma_{j,r,1}^{A,(0)} + \frac{\eta\|\mathbf{u}+\mathbf{v}_1\|_2^2}{m} \sum_{s=0}^{t} \frac{1}{1+\underline{b}^A e^{\underline{x}_s^A}}$$

$$\leq \gamma_{j,r}^{A,(0)} + \gamma_{j,r,1}^{A,(0)} + \frac{\eta\|\mathbf{u}+\mathbf{v}_1\|_2^2}{m} \int_{s=0}^{t} \frac{1}{1+\underline{b}^A e^{\underline{x}_s^A}} ds$$

$$\leq \gamma_{j,r}^{A,(0)} + \gamma_{j,r,1}^{A,(0)} + \frac{\eta\|\mathbf{u}+\mathbf{v}_1\|_2^2}{m} \int_{s=0}^{t} \frac{1}{\underline{c}^A} d\underline{x}_s^A$$

$$\leq \gamma_{j,r}^{A,(0)} + \gamma_{j,r,1}^{A,(0)} + \frac{\eta\|\mathbf{u}+\mathbf{v}_1\|_2^2}{\underline{c}^A m} \underline{x}_t^A - \frac{2\eta\|\mathbf{u}+\mathbf{v}_1\|_2^2}{m}$$

$$\leq \frac{\eta\|\mathbf{u}+\mathbf{v}_1\|_2^2}{\underline{c}^A m} \underline{x}_t^A - \frac{2\eta\|\mathbf{u}+\mathbf{v}_1\|_2^2}{m},$$

where the first inequality is by the fifth result in Lemma D.4, the second inequality is by summation and the forth inequality is by the definition of $\underline{x}_s^A$. On the other side, we have

$$\gamma_{j,r}^{A,(t+1)} + \gamma_{j,r,1}^{A,(t+1)} = \gamma_{j,r}^{A,(t)} + \gamma_{j,r,1}^{A,(t)} - \frac{\eta}{N_1 m} \sum_{i'=1}^{N_1} \ell_{i'}^{\prime(t)} \cdot \sigma'\big(\langle \mathbf{w}_{j,r}^{A,(t)}, y_i(\mathbf{u}+\mathbf{v}_1)\rangle\big)\|\mathbf{u}+\mathbf{v}_1\|_2^2$$

$$\geq \gamma_{j,r}^{A,(t)} + \gamma_{j,r,1}^{A,(t)} + \frac{\eta\|\mathbf{u}+\mathbf{v}_1\|_2^2}{m} \frac{1}{1+\overline{b}^A e^{\overline{x}_t^A}}$$

$$\geq \gamma_{j,r}^{A,(0)} + \gamma_{j,r,1}^{A,(0)} + \frac{\eta\|\mathbf{u}+\mathbf{v}_1\|_2^2}{m} \sum_{s=0}^{t} \frac{1}{1+\overline{b}^A e^{\overline{x}_s^A}}$$

$$\geq \gamma_{j,r}^{A,(0)} + \gamma_{j,r,1}^{A,(0)} + \frac{\eta\|\mathbf{u}+\mathbf{v}_1\|_2^2}{m} \int_{s=0}^{t-1} \frac{1}{1+\overline{b}^A e^{\overline{x}_s^A}} ds$$

$$\geq \gamma_{j,r}^{A,(0)} + \gamma_{j,r,1}^{A,(0)} + \frac{\eta\|\mathbf{u}+\mathbf{v}_1\|_2^2}{m} \int_{s=0}^{t-1} \frac{1}{\overline{c}^A} d\overline{x}_s^A$$

$$\geq \gamma_{j,r}^{A,(0)} + \gamma_{j,r,1}^{A,(0)} + \frac{\eta\|\mathbf{u}+\mathbf{v}_1\|_2^2}{\overline{c} m} \overline{x}_{t-1}^A - \frac{2\eta\|\mathbf{u}+\mathbf{v}_1\|_2^2}{m}$$

$$\geq \frac{\eta\|\mathbf{u}+\mathbf{v}_1\|_2^2}{\overline{c}^A m} \overline{x}_{t-1}^A - \frac{2\eta\|\mathbf{u}+\mathbf{v}_1\|_2^2}{m},$$

where the first inequality is by the fifth result in Lemma D.4, the second inequality is by summation and the forth inequality is by the definition of $\overline{x}_s^A$. Since that $\mathbf{u} \perp \mathbf{v}_1$, we have

$$\gamma_{j,r}^{A,(t)} = \frac{\|\mathbf{u}\|_2^2}{\|\mathbf{u}+\mathbf{v}_1\|_2^2}(\gamma_{j,r}^{A,(t)} + \gamma_{j,r,1}^{A,(t)}),$$

$$\gamma_{j,r,1}^{A,(t)} = \frac{\|\mathbf{v}_1\|_2^2}{\|\mathbf{u}+\mathbf{v}_1\|_2^2}(\gamma_{j,r}^{A,(t)} + \gamma_{j,r,1}^{A,(t)}).$$

Then, we complete the proof. □

**Lemma D.9.** *Under Condition 4.1, for $0 \leq t \leq T^*$, it holds that*

$$\frac{N_1}{12}(\overline{x}_{t-2}^A - \overline{x}_1^A) \leq \sum_{i \in [N_1]} \bar{\rho}_{j,r,i}^{A,(t)} \leq 5N_1 \underline{x}_{t-1}^A.$$

*Proof.* For $j = y_i$, it holds that

$$\sum_{i \in [N_1]} \overline{\rho}_{j,r,i,1}^{A,(t+1)} = \sum_{i \in [N_1]} \overline{\rho}_{j,r,i,1}^{A,(t)} - \sum_{i \in [N_1]} \frac{\eta}{N_1 m} \ell_i'^{A,(t)} \cdot \sigma'\left(\langle \mathbf{w}_{j,r}^{A,(t)}, \boldsymbol{\xi}_{i,1} \rangle\right) \cdot \|\boldsymbol{\xi}_{i,1}\|_2^2$$

$$= \sum_{i \in [N_1]} \overline{\rho}_{j,r,i,1}^{A,(t)} - \sum_{i \in S_{j,r}^{A,(t)}} \frac{\eta}{N_1 m} \ell_i'^{(t)} \cdot \|\boldsymbol{\xi}_{i,1}\|_2^2$$

$$\geq \sum_{i \in [N_1]} \overline{\rho}_{j,r,i,1}^{A,(t)} + |S_{j,r}^{A,(0)}| \frac{\eta}{N_1 m} \frac{1}{1 + \overline{b}^A \overline{x}_t^A} \cdot \|\boldsymbol{\xi}_{i,1}\|_2^2$$

$$\geq \sum_{s=1}^{t} |S_{j,r}^{A,(0)}| \frac{\eta}{N_1 m} \frac{1}{1 + \overline{b}^A \overline{x}_s^A} \cdot \|\boldsymbol{\xi}_{i,1}\|_2^2$$

$$\geq \int_{s=1}^{t-1} |S_{j,r}^{A,(0)}| \frac{\eta}{N_1 m} \frac{1}{1 + \overline{b}^A \overline{x}_s^A} \cdot \|\boldsymbol{\xi}_{i,1}\|_2^2 ds$$

$$\geq \frac{N_1}{12} (\overline{x}_{t-1}^A - \overline{x}_1^A),$$

where the first inequality is by $|S_{j,r}^{A,(t)}| \geq |S_{j,r}^{A,(0)}|$, the second inequality is by rearranging the summation and the last inequality is by the definition of $\overline{x}_s^A$. On the other side, it holds that

$$\sum_{i \in [N_1]} \overline{\rho}_{j,r,i,1}^{A,(t+1)} \leq \sum_{i \in [N_1]} \overline{\rho}_{j,r,i,1}^{A,(t)} + |S_{j,r}^{A,(t)}| \frac{\eta}{N_1 m} \frac{1}{1 + \underline{b}^A \underline{x}_t^A} \cdot \|\boldsymbol{\xi}_{i,1}\|_2^2$$

$$\leq \sum_{s=1}^{t} N_1 \frac{\eta}{N_1 m} \frac{1}{1 + \underline{b}^A \underline{x}_s^A} \cdot \|\boldsymbol{\xi}_{i,1}\|_2^2$$

$$\leq \int_{s=1}^{t} N_1 \frac{\eta}{N_1 m} \frac{1}{1 + \underline{b}^A \underline{x}_s^A} \cdot \|\boldsymbol{\xi}_{i,1}\|_2^2 ds$$

$$\leq 5 N_1 (\underline{x}_t^A - \underline{x}_1^A)$$

$$\leq 5 N_1 \underline{x}_t^A,$$

where the second inequality is by $|S_{j,r}^{A,(t)}| \leq N_1$ and rearranging the summation and the forth inequality is by the definition of $\underline{x}_s^A$. Then, we complete the proof. $\qquad\square$

## E  THE SECOND SYSTEM

To clearly distinguish the processes of Task 1 and Task 2, we assume that the upstream model is trained on Task 1 for $T^*$ epochs. At this point, a subset of the weights (i.e. inherited parameters, $1 \leq r \leq \alpha m$) is transferred to the downstream model, while the remaining weights ($\alpha m \leq r \leq m$) are randomly initialized. For simplicity, we assume that at $t = T^* + 1$, the downstream model has completed initialization and begins training on Task 2. So we have

$$\mathbf{w}_{j,r}^{D,(T^*+1)} = \begin{cases} \mathbf{w}_{j,r}^{A,(T^*)} & \text{if } 1 \leq r \leq \alpha m, \\ \widetilde{\mathbf{w}}_{j,r}^{D,(T^*)} & \text{if } \alpha m < r \leq m, \end{cases}$$

where $\widetilde{\mathbf{w}}_{j,r}^{D,(T^*)}, \alpha m < r \leq m$ is the re-initialized weights. To distinguish the weights used in Task 1 from those in Task 2, we use the superscript $D$ to denote the weights and coefficients of the downstream model on Task 2. Specially, because the coefficients $\gamma_{j,r,1}^{(t)}, \overline{\rho}_{j,r,i,1}^{(t)}$ and $\underline{\rho}_{j,r,i,1}^{(t)}$ are updated only on Task 1, we keep the superscript $A$ for them so that the readers can find the results of system 1 easily.

### E.1  COEFFICIENT SCALE ANALYSIS

In this section, we give the analysis of coefficient scale on Task 2 for $T^* + 1 \leq t \leq T^{**}$.

**Proposition E.1.** *Under Condition 4.1, and define $n = \max\{N_1, N_2\}$, for $T^* + 1 \leq t \leq T^{**}$, it holds that*

$$0 \leq \bar{\rho}_{j,r,i,2}^{D,(t)} \leq 4\log(T^{**}), \tag{E.1}$$

$$0 \geq \underline{\rho}_{j,r,i,2}^{D,(t)} \geq -2\sqrt{\log(12mN_2/\delta)} \cdot \sigma_0\sigma_{p,2}\sqrt{d} - C_1(N_1\frac{\sigma_{p,2}}{\sigma_{p,1}} + N_2)\sqrt{\frac{\log(4(N_1^2 + N_2^2)/\delta)}{d}}\log(T^{**}) \geq -4\log(T^{**}), \tag{E.2}$$

$$0 \leq \gamma_{j,r}^{D,(t)} - \gamma_{j,r}^{D,(T^*+1)} \leq \frac{C_2 N_2 \|\mathbf{u}\|_2^2}{\sigma_{p,2}^2 d}\log(T^{**}), \tag{E.3}$$

$$0 \leq \gamma_{j,r,2}^{D,(t)} \leq \frac{C_2 N_2 \|\mathbf{v}_2\|_2^2}{\sigma_{p,2}^2 d}\log(T^{**}), \tag{E.4}$$

*for all $r \in [m], j \in \{\pm 1\}, i \in [n]$, where $C_1$ and $C_2$ are two absolute constant.*

We will prove Proposition E.1 by induction. Before that we give some important technical lemmas used in the proof.

**Lemma E.2.** *Under Condition 4.1, for $T^* + 1 \leq t \leq T^{**}$, suppose equation E.1, equation E.2, equation E.3, equation E.4 hold at iteration t. Then, for all $j \in \{\pm 1\}, i \in [N_2]$, it holds that for $1 \leq r \leq \alpha m$*

$$\left|\langle \mathbf{w}_{j,r}^{D,(T^*+1)}, \boldsymbol{\xi}_{i,2}\rangle\right| \leq 2\sqrt{\log(12mN_2/\delta)} \cdot \sigma_0\sigma_{p,2}\sqrt{d} + 16N_1\frac{\sigma_{p,2}}{\sigma_{p,1}}\sqrt{\frac{\log(4(N_1^2 + N_2^2)/\delta)}{d}}\log(T^{**}), \tag{E.5}$$

*and for $\alpha m < r \leq m$*

$$\left|\langle \mathbf{w}_{j,r}^{D,(T^*+1)}, \boldsymbol{\xi}_{i,2}\rangle\right| \leq 2\sqrt{\log(12mN_2/\delta)} \cdot \sigma_0\sigma_{p,2}\sqrt{d} \tag{E.6}$$

*Proof of Lemma E.2.* When $\alpha m < r \leq m$, because these weights are re-initialized, the result can be directly derived from Lemma C.2. When $1 \leq r \leq \alpha m$, we have

$$\left|\langle \mathbf{w}_{j,r}^{D,(T^*+1)}, \boldsymbol{\xi}_{i,2}\rangle\right| = \left|\langle \mathbf{w}_{j,r}^{A,(T^*)}, \boldsymbol{\xi}_{i,2}\rangle\right|$$

$$\leq \left|\langle \mathbf{w}_{j,r}^{A,(0)}, \boldsymbol{\xi}_{i,2}\rangle\right| + \left|\sum_{i'=1}^{N_1} \rho_{j,r,i',1}^{A,(t)} \cdot \|\boldsymbol{\xi}_{i',1}\|_2^{-2} \cdot \langle \boldsymbol{\xi}_{i',1}, \boldsymbol{\xi}_{i,2}\rangle\right|$$

$$\leq 2\sqrt{\log(12mN_2/\delta)} \cdot \sigma_0\sigma_{p,2}\sqrt{d} + \left|\sum_{i'=1}^{N_1} \|\boldsymbol{\xi}_{i',1}\|_2^{-2} \cdot \langle \boldsymbol{\xi}_{i',1}, \boldsymbol{\xi}_{i,2}\rangle\right| 4\log(T^{**})$$

$$\leq 2\sqrt{\log(12mN_2/\delta)} \cdot \sigma_0\sigma_{p,2}\sqrt{d}$$
$$+ N_1 \cdot \frac{2}{\sigma_{p,1}^2 d} \cdot 2\sigma_{p,1}\sigma_{p,2} \cdot \sqrt{d\log(4(N_1^2 + N_2^2)/\delta)}4\log(T^{**})$$

$$\leq 2\sqrt{\log(12mN_2/\delta)} \cdot \sigma_0\sigma_{p,2}\sqrt{d} + 16N_1\frac{\sigma_{p,2}}{\sigma_{p,1}}\sqrt{\frac{\log(4(N_1^2 + N_2^2)/\delta)}{d}}\log(T^{**}),$$

where the first inequality is by triangle inequality, the second inequality is by Lemma C.2, Lemma D.5 and $T^* \leq T^{**}$ and the third inequality is by Lemma C.1. $\square$

**Lemma E.3.** *Under Condition 4.1, for $T^* + 1 \leq t \leq T^{**}$, suppose equation E.1, equation E.2, equation E.3, equation E.4 hold at iteration t. Then, for all $r \in [m], j \in \{\pm 1\}, i \in [N_2]$, it holds that*

$$\left|\langle \mathbf{w}_{j,r}^{D,(t)} - \mathbf{w}_{j,r}^{D,(T^*+1)}, \boldsymbol{\xi}_{i,2}\rangle - \underline{\rho}_{j,r,i,2}^{D,(t)}\right| \leq 16N_2\sqrt{\frac{\log(4(N_1^2 + N_2^2)/\delta)}{d}}\log(T^{**}), \quad j \neq y_{i,1}; \tag{E.7}$$

$$\left|\langle \mathbf{w}_{j,r}^{D,(t)} - \mathbf{w}_{j,r}^{D,(T^*+1)}, \boldsymbol{\xi}_{i,2}\rangle - \bar{\rho}_{j,r,i,2}^{D,(t)}\right| \leq 16N_2\sqrt{\frac{\log(4(N_1^2 + N_2^2)/\delta)}{d}}\log(T^{**}), \quad j = y_{i,1}.; \tag{E.8}$$

*Proof.* The proof is similar to that in Lemma D.2 and uses the fact $N_1^2 + N_2^2 > N_2^2$. So we omit it here. $\square$

Before we give the next result, we need to define

$$\kappa_D = \frac{4C_2 N_2 \|\mathbf{u} + \mathbf{v}_2\|_2^2}{\sigma_{p,2}^2 d} \log(T^{**}) + \frac{4C_2 N_1 \|\mathbf{u}\|_2^2}{\sigma_{p,1}^2 d} \log(T^*) + 16\sqrt{\log(12mN_2/\delta)} \cdot \sigma_0 \sigma_{p,2} \sqrt{d}$$

$$+ (4C_1 + 64)(N_1 \frac{\sigma_{p,2}}{\sigma_{p,1}} + N_2) \sqrt{\frac{\log(4(N_1^2 + N_2^2)/\delta)}{d}} \log(T^{**}).$$

By the condition of $d$ in Condition 4.1, we have $\kappa_D \leq 0.1$.

**Lemma E.4.** *Under Condition 4.1, for $T^* + 1 \leq t \leq T^{**}$, suppose equation E.1, equation E.2, equation E.3, equation E.4 hold at iteration t. Then, it holds that*

$$F_{-y_{i,2}}(\mathbf{W}_{-y_{i,2}}^{D,(t)}, \mathbf{x}_{i,2}) \leq \kappa_D/4, \quad -\kappa_D/4 + \frac{1}{m} \sum_{r=1}^m \bar{\rho}_{j,r,i,2}^{D,(t)} \leq F_{y_{i,2}}(\mathbf{W}_{y_{i,2}}^{D,(t)}, \mathbf{x}_{i,2}) \leq \kappa_D/4 + \frac{1}{m} \sum_{r=1}^m \bar{\rho}_{j,r,i,2}^{D,(t)},$$

$$-\kappa_D/2 + \frac{1}{m} \sum_{r=1}^m \bar{\rho}_{j,r,i,2}^{D,(t)} \leq y_{i,2} f(\mathbf{W}^{D,(t)}, \mathbf{x}_{i,2}) \leq \kappa_D/2 + \frac{1}{m} \sum_{r=1}^m \bar{\rho}_{j,r,i,2}^{D,(t)}.$$

*Proof.* Recall that the definition of $F_j(\mathbf{W}_j^{D,(t)}, \mathbf{x}_{i,2})$ as

$$F_j(\mathbf{W}_j^{D,(t)}, \mathbf{x}_{i,2}) = \frac{1}{m} \sum_{r=1}^m \left[ \sigma\left(\langle \mathbf{w}_{j,r}^{D,(t)}, y_{i,2}(\mathbf{u} + \mathbf{v}_2)\rangle\right) + \sigma\left(\langle \mathbf{w}_{j,r}^{D,(t)}, \boldsymbol{\xi}_{i,2}\rangle\right) \right].$$

When $j = -y_{i,2}$, we have

$$F_{-y_{i,2}}(\mathbf{W}_{-y_{i,2}}^{D,(t)}, \mathbf{x}_{i,2}) \leq \frac{1}{m} \sum_{r=1}^m \left[ |\langle \mathbf{w}_{j,r}^{D,(t)}, y_{i,2}\mathbf{u}\rangle| + \left[ |\langle \mathbf{w}_{j,r}^{D,(t)}, y_{i,2}\mathbf{v}_2\rangle| + |\langle \mathbf{w}_{j,r}^{D,(t)}, \boldsymbol{\xi}_{i,2}\rangle| \right] \right]$$

$$\leq \frac{1}{m} \sum_{r=1}^m \left[ \gamma_{j,r}^{D,(t)} + \gamma_{j,r,2}^{D,(t)} + |\langle \mathbf{w}_{j,r}^{D,(T^*+1)}, \boldsymbol{\xi}_{i,2}\rangle| + |\varrho_{j,r,i,2}^{D,(t)}| \right.$$

$$\left. + 16N_2 \sqrt{\frac{\log(4(N_1^2 + N_2^2)/\delta)}{d}} \log(T^{**}) \right]$$

$$\leq \frac{C_2 N_2 \|\mathbf{u} + \mathbf{v}_2\|_2^2}{\sigma_{p,2}^2 d} \log(T^{**}) + \frac{1}{m} \sum_{r=1}^m \gamma_{j,r}^{D,(T^*+1)} + 4\sqrt{\log(12mN_2/\delta)} \cdot \sigma_0 \sigma_{p,2} \sqrt{d}$$

$$+ (C_1 + 16)(N_1 \frac{\sigma_{p,2}}{\sigma_{p,1}} + N_2) \sqrt{\frac{\log(4(N_1^2 + N_2^2)/\delta)}{d}} \log(T^{**})$$

$$\leq \frac{C_2 N_2 \|\mathbf{u} + \mathbf{v}_2\|_2^2}{\sigma_{p,2}^2 d} \log(T^{**}) + \frac{C_2 N_1 \|\mathbf{u}\|_2^2}{\sigma_{p,1}^2 d} \log(T^*) + 4\sqrt{\log(12mN_2/\delta)} \cdot \sigma_0 \sigma_{p,2} \sqrt{d}$$

$$+ (C_1 + 16)(N_1 \frac{\sigma_{p,2}}{\sigma_{p,1}} + N_2) \sqrt{\frac{\log(4(N_1^2 + N_2^2)/\delta)}{d}} \log(T^{**})$$

$$\leq \kappa_D/4,$$

where the first inequality uses triangle inequality, the second inequality is by Lemma E.3 and triangle inequality, the third inequality is by equation E.2, equation E.3, equation E.4, Lemma E.2 and $0 \leq \alpha \leq 1$, the forth inequality is by

equation D.17 and $0 \leq \alpha \leq 1$, and the last inequality is by the definition of $\kappa_D$. When $j = y_{i,2}$, we have

$$\left| F_{y_{i,2}}(\mathbf{W}_{y_{i,2}}^{D,(t)}, \mathbf{x}_{i,1}) - \frac{1}{m} \sum_{r=1}^{m} \bar{\rho}_{j,r,i,2}^{D,(t)} \right| \leq \frac{1}{m} \sum_{r=1}^{m} \left[ |\langle \mathbf{w}_{j,r}^{D,(t)}, y_{i,2}\mathbf{u} \rangle| + \left[ |\langle \mathbf{w}_{j,r}^{D,(t)}, y_{i,2}\mathbf{v}_2 \rangle| + |\langle \mathbf{w}_{j,r}^{D,(t)}, \boldsymbol{\xi}_{i,2} \rangle - \bar{\rho}_{j,r,i,2}^{D,(t)}| \right] \right.$$

$$\leq \frac{1}{m} \sum_{r=1}^{m} \left[ \gamma_{j,r}^{D,(t)} + \gamma_{j,r,1}^{D,(t)} + 16N_2 \sqrt{\frac{\log(4(N_1^2 + N_2^2)/\delta)}{d}} \log(T^{**}) \right.$$

$$\left. + |\langle \mathbf{w}_{j,r}^{D,(T^*+1)}, \boldsymbol{\xi}_{i,2} \rangle| \right]$$

$$\leq \frac{C_2 N_2 \|\mathbf{u} + \mathbf{v}_2\|_2^2}{\sigma_{p,2}^2 d} \log(T^{**}) + \frac{C_2 N_1 \|\mathbf{u}\|_2^2}{\sigma_{p,1}^2 d} \log(T^*)$$

$$+ 16N_2 \sqrt{\frac{\log(4(N_1^2 + N_2^2)/\delta)}{d}} \log(T^{**})$$

$$+ 2\sqrt{\log\left(\frac{12mN_2}{\delta}\right)} \cdot \sigma_0 \sigma_{p,2} \sqrt{d}$$

$$\leq \kappa_D/4,$$

where the first inequality uses triangle inequality, the second inequality is by Lemma E.3, the third inequality is by equation E.2, equation E.3, equation E.4, equation D.17 and Lemma E.2, and the last inequality is by the definition of $\kappa_D$. At last, because

$$y_{i,2} f(\mathbf{W}^{D,(t)}, \mathbf{x}_{i,2}) = F_{y_{i,2}}(\mathbf{W}_{y_{i,2}}^{D,(t)}, \mathbf{x}_{i,2}) - F_{-y_{i,2}}(\mathbf{W}_{-y_{i,2}}^{D,(t)}, \mathbf{x}_{i,2}),$$

we complete the proof. $\qquad\square$

**Lemma E.5.** *Under Condition 4.1, and define* $n = \max\{N_1, N_2\}$, *for* $T^* \leq t \leq T^{**}$, *suppose equation E.1, equation E.2, equation E.3, equation E.4 hold at iteration t. Then, the following results hold for any iteration t:*

1. $\frac{1}{m} \sum_{r=1}^{m} \left[ \bar{\rho}_{y_{i,2},r,i,2}^{D,(t)} - \bar{\rho}_{y_{k,2},r,k,2}^{D,(t)} \right] \leq \log(12) + \kappa_D + \sqrt{\log(2N_2/\delta)/m}$ *for all* $i, k \in [N_2]$.

2. $S_i^{D,(0)} \subseteq S_i^{D,(t)}$, *where* $S_i^{D,(t)} = \{r \in [m] : \langle \mathbf{w}_{y_{i,2},r}^{D,(t)}, \boldsymbol{\xi}_{i,2} \rangle > 0\}$.

3. $S_{j,r}^{D,(0)} \subseteq S_{j,r}^{D,(t)}$, *where* $S_{j,r}^{D,(t)} = \{i \in [N_2] : y_{i,2} = j, \langle \mathbf{w}_{j,r}^{D,(t)}, \boldsymbol{\xi}_{i,2} \rangle > 0\}$.

4. $\ell_i'^{(t)}/\ell_k'^{(t)} \leq 13$.

5. *A refined estimation of* $\frac{1}{m} \sum_{r=1}^{m} \rho_{y_{i,2},r,i,2}^{D,(t)}$ *and* $\ell_i'^{(t)}$. *It holds that*

$$\underline{x}_t^D \leq \frac{1}{m} \sum_{r=1}^{m} \bar{\rho}_{y_{i,2},r,i,2}^{D,(t)} \leq \overline{x}_t^D + \overline{c}^D/(1 + \overline{b}^D),$$

$$\frac{1}{1 + \underline{b}^D e^{\underline{x}_t^D}} \leq -\ell_i'^{(t)} \leq \frac{1}{1 + \overline{b}^D e^{\overline{x}_t^D}},$$

*where* $\overline{x}_t^D, \underline{x}_t^D$ *are the unique solution of*

$$\overline{x}_t^D + \overline{b}^D e^{\overline{x}_t^D} = \overline{c}^D t + \overline{b}^D,$$

$$\underline{x}_t^D + \underline{b}^D e^{\underline{x}_t^D} = \underline{c}^D t + \underline{b}^D,$$

*and* $\overline{b}^D = e^{-\kappa_D/2}, \overline{c}^D = \frac{3\eta\sigma_{p,2}^2 d}{2N_2 m}, \underline{b}^D = e^{\kappa_D/2}$ *and* $\underline{c}^D = \frac{\eta\sigma_{p,2}^2 d}{5N_2 m}$.

*Proof.* We prove it by induction. When $t = 0$, all results hold obviously. Now, we suppose there exists $\hat{t}$ and all the results hold for $t \leq \hat{t} - 1$. Next, we prove these results hold at $t = \hat{t}$.

First, we prove the first result. With Lemma E.4, for $t \le \hat{t} - 1$, we have

$$-\kappa_D/2 \le y_{i,2} f(\mathbf{W}^{D,(t)}, \mathbf{x}_{i,2}) - \frac{1}{m} \sum_{r=1}^{m} \bar{\rho}_{j,r,i,2}^{D,(t)} \le \kappa_D/2,$$

$$-\kappa_D/2 \le y_{k,2} f(\mathbf{W}^{D,(t)}, \mathbf{x}_{k,2}) - \frac{1}{m} \sum_{r=1}^{m} \bar{\rho}_{j,r,k,2}^{D,(t)} \le \kappa_D/2.$$

By subtracting the two equations, we have

$$\left| \left[ y_{i,2} f(\mathbf{W}^{D,(t)}, \mathbf{x}_{i,2}) - y_{k,2} f(\mathbf{W}^{D,(t)}, \mathbf{x}_{k,2}) \right] - \left[ \frac{1}{m} \sum_{r=1}^{m} \bar{\rho}_{j,r,i,2}^{D,(t)} - \frac{1}{m} \sum_{r=1}^{m} \bar{\rho}_{j,r,k,2}^{D,(t)} \right] \right| \le \kappa_D. \qquad \text{(E.9)}$$

When $\frac{1}{m} \sum_{r=1}^{m} \left[ \bar{\rho}_{y_{i,2},r,i,2}^{D,(\hat{t}-1)} - \bar{\rho}_{r,k,i,2}^{D,(\hat{t}-1)} \right] \le \log(12) + \kappa_D$, we have

$$\frac{1}{m} \sum_{r=1}^{m} \left[ \bar{\rho}_{y_{i,2},r,i,2}^{D,(\hat{t})} - \bar{\rho}_{y_{k,2},r,k,2}^{D,(\hat{t})} \right] = \frac{1}{m} \sum_{r=1}^{m} \left[ \bar{\rho}_{y_{i,2},r,i,2}^{D,(\hat{t}-1)} - \bar{\rho}_{y_{k,2},r,k,2}^{D,(\hat{t}-1)} \right] - \frac{\eta}{N_2 m} \cdot \frac{1}{m} \sum_{r=1}^{m} \left[ \ell_i'^{(\hat{t}-1)} \cdot \sigma'\left( \langle \mathbf{w}_{y_{i,2},r}^{D,(\hat{t}-1)}, \boldsymbol{\xi}_{i,2} \rangle \right) \right.$$

$$\left. \cdot \|\boldsymbol{\xi}_{i,2}\|_2^2 - \ell_k'^{(\hat{t}-1)} \cdot \sigma'\left( \langle \mathbf{w}_{y_{k,2},r}^{D,(\hat{t}-1)}, \boldsymbol{\xi}_{k,2} \rangle \right) \cdot \|\boldsymbol{\xi}_{k,2}\|_2^2 \right] \qquad \text{(E.10)}$$

$$\le \frac{1}{m} \sum_{r=1}^{m} \left[ \bar{\rho}_{y_{i,2},r,i,2}^{D,(\hat{t}-1)} - \bar{\rho}_{y_{k,2},r,k,2}^{D,(\hat{t}-1)} \right] - \frac{\eta}{N_2 m} \cdot \frac{1}{m} \sum_{r=1}^{m} \ell_i'^{(\hat{t}-1)} \cdot \sigma'\left( \langle \mathbf{w}_{y_{i,2},r}^{D,(\hat{t}-1)}, \boldsymbol{\xi}_{i,2} \rangle \right)$$

$$\cdot \|\boldsymbol{\xi}_{i,2}\|_2^2,$$

where the first equality is by the update rule, the second inequality uses the fact $\ell_k'^{(\hat{t}-1)} < 0$. Next, we bound the second term as

$$\left| \frac{\eta}{N_2 m} \cdot \frac{1}{m} \sum_{r=1}^{m} \ell_i'^{(\hat{t}-1)} \cdot \sigma'\left( \langle \mathbf{w}_{y_{i,2},r}^{D,(\hat{t}-1)}, \boldsymbol{\xi}_{i,2} \rangle \right) \cdot \|\boldsymbol{\xi}_{i,2}\|_2^2 \right| \le \frac{\eta}{N_2 m} \cdot \frac{1}{m} \sum_{r=1}^{m} |\ell_i'^{(\hat{t}-1)}| \cdot \sigma'\left( \langle \mathbf{w}_{y_{i,2},r}^{D,(\hat{t}-1)}, \boldsymbol{\xi}_{i,2} \rangle \right) \cdot \|\boldsymbol{\xi}_{i,2}\|_2^2$$

$$\le \frac{\eta}{N_2 m^2} \cdot |S_i^{D,(\hat{t}-1)}| \cdot \|\boldsymbol{\xi}_{i,2}\|_2^2$$

$$\le \frac{\eta \sigma_{p,2}^2 d}{2 N_2 m}$$

$$\le \sqrt{\log(2N_2/\delta)/m},$$

where the first inequality is by triangle inequality, the second inequality uses the fact $-1 < \ell_i'^{(\hat{t}-1)} < 0$ and the definition of $S_i^{D,(\hat{t}-1)}$, the third inequality is by Lemma C.1 and $|S_i^{D,(\hat{t}-1)}| \le m$, and the forth inequality is by the condition of $\eta$ in Condition 4.1. Therefore, we have

$$\frac{1}{m} \sum_{r=1}^{m} \left[ \bar{\rho}_{y_{i,2},r,i,2}^{D,(\hat{t})} - \bar{\rho}_{y_{k,2},r,k,2}^{D,(\hat{t})} \right] \le \frac{1}{m} \sum_{r=1}^{m} \left[ \bar{\rho}_{y_{i,2},r,i,2}^{D,(\hat{t}-1)} - \bar{\rho}_{y_{k,2},r,k,2}^{D,(\hat{t}-1)} \right] + \sqrt{\log(2N_2/\delta)/m}$$

$$\le \log(12) + \kappa_D + \sqrt{\log(2N_2/\delta)/m}.$$

On the other side, When $\frac{1}{m} \sum_{r=1}^{m} \left[ \bar{\rho}_{y_{i,2},r,i,2}^{D,(\hat{t}-1)} - \bar{\rho}_{r,k,i,2}^{D,(\hat{t}-1)} \right] \ge \log(12) + \kappa_D$, with equation E.9, we have

$$y_{i,2} f(\mathbf{W}^{D,(\hat{t}-1)}, \mathbf{x}_{i,2}) - y_{k,2} f(\mathbf{W}^{D,(\hat{t}-1)}, \mathbf{x}_{k,2}) \ge \frac{1}{m} \sum_{r=1}^{m} \left[ \bar{\rho}_{y_{i,2},r,i,2}^{D,(\hat{t}-1)} - \bar{\rho}_{r,k,i,2}^{D,(\hat{t}-1)} \right] - \kappa_D$$

$$\ge \log(12),$$

where the first inequality uses equation E.9. Then, it holds that

$$\frac{-\ell_i'^{(\hat{t}-1)}}{-\ell_k'^{(\hat{t}-1)}} \le e^{-y_{i,2} f(\mathbf{W}^{D,(\hat{t}-1)}, \mathbf{x}_{i,2}) + y_{k,2} f(\mathbf{W}^{D,(\hat{t}-1)}, \mathbf{x}_{k,2})} < \frac{1}{12}. \qquad \text{(E.11)}$$

Then, we have

$$
\frac{-\sum_{r=1}^{m} \ell_i'^{(\hat{t}-1)} \cdot \sigma'\big(\langle \mathbf{w}_{y_{i,2},r}^{D,(\hat{t}-1)}, \boldsymbol{\xi}_{i,2}\rangle\big) \cdot \|\boldsymbol{\xi}_{i,2}\|_2^2}{-\sum_{r=1}^{m} \ell_k'^{(\hat{t}-1)} \cdot \sigma'\big(\langle \mathbf{w}_{y_{k,2},r}^{D,(\hat{t}-1)}, \boldsymbol{\xi}_{k,2}\rangle\big) \cdot \|\boldsymbol{\xi}_{k,2}\|_2^2} = \frac{-\ell_i'^{(\hat{t}-1)} \cdot |S_i^{D,(\hat{t}-1)}| \cdot \|\boldsymbol{\xi}_{i,2}\|_2^2}{-\ell_k'^{(\hat{t}-1)} \cdot |S_k^{D,(\hat{t}-1)}| \cdot \|\boldsymbol{\xi}_{k,2}\|_2^2}
$$

$$
< \frac{1}{4} \cdot \frac{|S_i^{D,(\hat{t}-1)}|}{|S_k^{D,(0)}|}
$$

$$
\leq 1,
$$

where the first inequality uses equation E.11 and Lemma C.1, and the second inequality uses the fact that $|S_i^{D,(\hat{t}-1)}| \leq m$, the induction $|S_k^{D,(0)}| \leq |S_k^{D,(\hat{t}-1)}|$ and $|S_k^{D,(0)}| \geq m/4$. Then, with equation E.10, it holds that

$$
\frac{1}{m} \sum_{r=1}^{m} \Big[ \rho_{y_{i,2},r,i,2}^{D,(\hat{t})} - \bar{\rho}_{y_{k,2},r,k,2}^{D,(\hat{t})} \Big] \leq \frac{1}{m} \sum_{r=1}^{m} \Big[ \rho_{y_{i,2},r,i,2}^{D,(\hat{t}-1)} - \bar{\rho}_{y_{k,2},r,k,2}^{D,(\hat{t}-1)} \Big]
$$

$$
\leq \log(12) + \kappa_D + \sqrt{\log(2N_2/\delta)/m}.
$$

Next, we prove the second result and the third result together. When $j = y_{i,2}$, by the update rule in Task 2 in Lemma B.2, it hols that

$$
\langle \mathbf{w}_{j,r}^{D,(\hat{t})}, \boldsymbol{\xi}_{i,2}\rangle = \langle \mathbf{w}_{j,r}^{D,(\hat{t}-1)}, \boldsymbol{\xi}_{i,2}\rangle - \frac{\eta}{N_2 m} \sum_{i' \in [N_2]} \ell_{i'}'^{(\hat{t}-1)} \cdot \sigma'\big(\langle \mathbf{w}_{j,r}^{D,(\hat{t}-1)}, \boldsymbol{\xi}_{i'}\rangle\big) \cdot \langle \boldsymbol{\xi}_{i'}, \boldsymbol{\xi}_i\rangle
$$

$$
= \langle \mathbf{w}_{j,r}^{D,(\hat{t}-1)}, \boldsymbol{\xi}_{i,2}\rangle - \frac{\eta}{N_2 m} \ell_i'^{(\hat{t}-1)} \cdot \sigma'\big(\langle \mathbf{w}_{j,r}^{D,(\hat{t}-1)}, \boldsymbol{\xi}_{i,2}\rangle\big) \cdot \|\boldsymbol{\xi}_{i,2}\|_2^2
$$

$$
\quad - \frac{\eta}{N_2 m} \sum_{i' \neq i} \ell_{i'}'^{(\hat{t}-1)} \cdot \sigma'\big(\langle \mathbf{w}_{j,r}^{D,(\hat{t}-1)}, \boldsymbol{\xi}_{i'}\rangle\big) \cdot \langle \boldsymbol{\xi}_{i'}, \boldsymbol{\xi}_i\rangle
$$

$$
\geq \langle \mathbf{w}_{j,r}^{D,(\hat{t}-1)}, \boldsymbol{\xi}_{i,2}\rangle + \frac{\eta\sigma_{p,2}^2 d}{2N_2 m} \ell_i'^{(\hat{t}-1)} - \frac{26\eta\sigma_{p,2}^2\sqrt{d\log(4N_2^2/\delta)}}{m} \ell_i'^{(\hat{t}-1)}
$$

$$
\geq \langle \mathbf{w}_{j,r}^{D,(\hat{t}-1)}, \boldsymbol{\xi}_{i,2}\rangle,
$$

where the first inequality is by Lemma C.1 and the induction $\ell_k'^{(\hat{t}-1)}/\ell_i'^{(\hat{t}-1)} \leq 13$, and the second inequality is by the condition of $d$ in Condition 4.1. Then, we know that $S_i^{D,(0)} \subseteq S_i^{D,(\hat{t}-1)} \subseteq S_i^{D,(\hat{t})}$ and $S_{j,r}^{D,(0)} \subseteq S_{j,r}^{D,(\hat{t}-1)} \subseteq S_{j,r}^{D,(\hat{t})}$ by induction.

Next, we prove the forth result. With equation E.9, it holds that

$$
\frac{\ell_i'^{(\hat{t})}}{\ell_k'^{(\hat{t})}} \leq e^{-y_{i,2}f(\mathbf{W}^{D,(\hat{t})}, \mathbf{x}_{i,2}) + y_{k,2}f(\mathbf{W}^{D,(\hat{t})}, \mathbf{x}_{k,2})}
$$

$$
\leq e^{-\frac{1}{m}\sum_{r=1}^{m} \bar{\rho}_{j,r,i,2}^{D,(\hat{t})} + \frac{1}{m}\sum_{r=1}^{m} \bar{\rho}_{j,r,k,2}^{D,(\hat{t})} + \kappa_D}
$$

$$
\leq e^{\log(12) + 2\kappa_D + \sqrt{\log(2N_2/\delta)/m}} = 12 + o(1) \leq 13,
$$

where the second inequality is by equation E.9, the third inequality is by the first result of the induction, and the equation is by the selection of $\kappa_D$ and $m$. Next, we prove the fifth result. From Lemma B.2, we know that

$$
\frac{1}{m} \sum_{r=1}^{m} \bar{\rho}_{y_{i,2},r,i,2}^{D,(\hat{t})} = \frac{1}{m} \sum_{r=1}^{m} \bar{\rho}_{y_{i,2},r,i,2}^{D,(\hat{t}-1)} - \frac{\eta}{N_2 m} \cdot \frac{1}{m} \sum_{r=1}^{m} \ell_i'^{(\hat{t}-1)} \cdot \sigma'\big(\langle \mathbf{w}_{y_{i,2},r}^{D,(t)}, \boldsymbol{\xi}_{i,2}\rangle\big) \cdot \|\boldsymbol{\xi}_{i,2}\|_2^2
$$

$$
= \frac{1}{m} \sum_{r=1}^{m} \bar{\rho}_{y_{i,2},r,i,2}^{D,(\hat{t}-1)} - \frac{\eta}{N_2 m} \cdot \frac{|S_i^{D,(\hat{t}-1)}|}{m} \cdot \ell_i'^{(\hat{t}-1)} \cdot \|\boldsymbol{\xi}_{i,2}\|_2^2.
$$

Here, with Lemma E.4, the gradient $\ell_i'^{(\hat{t}-1)}$ can be bounded as

$$
\frac{-1}{1 + e^{\frac{1}{m}\sum_{r=1}^{m} \bar{\rho}_{y_{i,2},r,i,2}^{D,(\hat{t}-1)} - \kappa_D/2}} \leq \ell_i'^{(\hat{t}-1)} = \frac{-1}{1 + e^{y_{i,2}f(\mathbf{W}^{D,(\hat{t}-1)}, \mathbf{x}_{i,2})}} \leq \frac{-1}{1 + e^{\frac{1}{m}\sum_{r=1}^{m} \bar{\rho}_{y_{i,2},r,i,2}^{D,(\hat{t}-1)} + \kappa_D/2}}. \tag{E.12}
$$

Then, by the update rule of $\overline{\rho}_{j,r,i,2}^{D,(\widehat{t})}$ in Lemma B.2, we have

$$\frac{1}{m}\sum_{r=1}^{m}\overline{\rho}_{y_{i,2},r,i,2}^{D,(\widehat{t})} \leq \frac{1}{m}\sum_{r=1}^{m}\overline{\rho}_{y_{i,2},r,i,2}^{D,(\widehat{t}-1)} + \frac{\eta}{N_2 m}\cdot\frac{|S_i^{D,(\widehat{t}-1)}|}{m}\cdot\frac{1}{1+e^{\frac{1}{m}\sum_{r=1}^{m}\rho_{y_{i,2},r,i,2}^{D,(\widehat{t}-1)}-\kappa_D/2}}\cdot\|\boldsymbol{\xi}_{i,2}\|_2^2;$$

$$\leq \frac{1}{m}\sum_{r=1}^{m}\overline{\rho}_{y_{i,2},r,i,2}^{D,(\widehat{t}-1)} + \frac{3\eta\sigma_p^2 d}{2N_2 m}\cdot\frac{1}{1+e^{\frac{1}{m}\sum_{r=1}^{m}\rho_{y_{i,2},r,i,2}^{D,(\widehat{t}-1)}-\kappa_D/2}};$$

$$\frac{1}{m}\sum_{r=1}^{m}\overline{\rho}_{y_{i,2},r,i,2}^{D,(\widehat{t})} \geq \frac{1}{m}\sum_{r=1}^{m}\overline{\rho}_{y_{i,2},r,i,2}^{D,(\widehat{t}-1)} + \frac{\eta}{N_2 m}\cdot\frac{|S_i^{D,(0)}|}{m}\cdot\frac{1}{1+e^{\frac{1}{m}\sum_{r=1}^{m}\rho_{y_{i,2},r,i,2}^{D,(\widehat{t}-1)}+\kappa_D/2}}\cdot\|\boldsymbol{\xi}_{i,2}\|_2^2$$

$$\geq \frac{1}{m}\sum_{r=1}^{m}\overline{\rho}_{y_{i,2},r,i,2}^{D,(\widehat{t}-1)} + \frac{\eta\sigma_p^2 d}{5N_2 m}\cdot\frac{1}{1+e^{\frac{1}{m}\sum_{r=1}^{m}\rho_{y_{i,2},r,i,2}^{D,(\widehat{t}-1)}+\kappa_D/2}}.$$

So, the estimation of $\frac{1}{m}\sum_{r=1}^{m}\overline{\rho}_{y_{i,2},r,i,2}^{D,(\widehat{t})}$ can be approximated by solving the continuous-time iterative equation

$$\frac{dx_t^D}{dt} = \frac{a}{1+be^{x_t^D}} \quad and \quad x_0 = 0.$$

The result is shown in Lemma C.4. For the gradient counterparts, with Lemma D.3, the gradient $\ell_i'^{(\widehat{t}-1)}$ can be bounded as

$$\frac{1}{1+e^{\frac{1}{m}\sum_{r=1}^{m}\rho_{y_{i,2},r,i,2}^{D,(\widehat{t}-1)}+\kappa_D/2}} \leq -\ell_i'^{(\widehat{t}-1)} = \frac{1}{1+e^{y_{i,2}f(\mathbf{W}^{D,(\widehat{t}-1)},\mathbf{x}_{i,2})}} \leq \frac{1}{1+e^{\frac{1}{m}\sum_{r=1}^{m}\rho_{y_{i,2},r,i,2}^{D,(\widehat{t}-1)}-\kappa_D/2}}.$$

The result is obvious since that $1/m\sum_{r=1}^{m}\rho_{y_{i,2},r,i,2}^{D,(\widehat{t}-1)}$ is bounded. Since then we complete the proof. $\qquad\square$

*Proof of Proposition E.1.* We prove it by induction. When $t=0$, all results hold obviously. Now, we suppose there exists $\widehat{t}$ and all the results hold for $t \leq \widehat{t}-1$. Next, we prove these results hold at $t=\widehat{t}$.

First, for the first result, when $j \neq y_{i,2}$, we have $\overline{\rho}_{j,r,i,2}^{D,(\widehat{t})}=0$. When $j=y_{i,2}$, by the update rule, it holds that

$$\overline{\rho}_{j,r,i,2}^{D,(\widehat{t})} = \overline{\rho}_{j,r,i,2}^{D,(\widehat{t}-1)} - \frac{\eta}{N_2 m}\ell_i'^{(\widehat{t}-1)}\cdot\sigma'\big(\langle\mathbf{w}_{j,r}^{D,(\widehat{t}-1)},\boldsymbol{\xi}_{i,2}\rangle\big)\cdot\|\boldsymbol{\xi}_{i,2}\|_2^2. \tag{E.13}$$

If $\overline{\rho}_{j,r,i,2}^{D,(\widehat{t}-1)} \leq 2\log(T^{**})$, we have

$$\overline{\rho}_{j,r,i,2}^{D,(\widehat{t})} \leq \overline{\rho}_{j,r,i,2}^{D,(\widehat{t}-1)} + \frac{\eta}{N_2 m}\frac{3\sigma_{p,2}^2 d}{2}$$
$$\leq 2\log(T^{**}) + \log(T^{**}) \leq 4\log(T^{**}),$$

where the first inequality uses the fact $-1 \leq \ell_i'^{(\widehat{t}-1)} \leq 0$ and Lemma C.1, and the second inequality is by the condition of $\eta$ in Condition 4.1. If $\overline{\rho}_{j,r,i,2}^{D,(\widehat{t}-1)} \geq 2\log(T^{**})$, from equation E.13 we know that $\overline{\rho}_{j,r,i,2}^{D,(t)}$ increases with $t$. Therefore, suppose that $t_{j,r,i,2}$ is the last time satisfying $\overline{\rho}_{j,r,i,2}^{D,(t_{j,r,i,2})} \leq 2\log(T^{**})$. Now, we want to show that the increment of $\overline{\rho}_{j,r,i,2}^{D}$ from $t_{j,r,i,2}$ to $\widehat{t}$ does not exceed $2\log(T^{**})$.

$$\overline{\rho}_{j,r,i,2}^{D,(\widehat{t})} = \overline{\rho}_{j,r,i,2}^{D,(t_{j,r,i,2})} - \frac{\eta}{N_2 m}\ell_i'^{(t_{j,r,i,2})}\cdot\sigma'\big(\langle\mathbf{w}_{j,r}^{D,(t_{j,r,i,2})},\boldsymbol{\xi}_{i,2}\rangle\big)\cdot\|\boldsymbol{\xi}_{i,2}\|_2^2$$
$$- \sum_{t_{j,r,i,2}<t\leq\widehat{t}-1}\frac{\eta}{N_2 m}\ell_i'^{(t)}\cdot\sigma'\big(\langle\mathbf{w}_{j,r}^{D,(t)},\boldsymbol{\xi}_{i,2}\rangle\big)\cdot\|\boldsymbol{\xi}_{i,2}\|_2^2. \tag{E.14}$$

Here, the second term can be bounded as

$$\left|\frac{\eta}{N_2 m}\ell_i'^{(t_{j,r,i,2})}\cdot\sigma'\big(\langle\mathbf{w}_{j,r}^{D,(t_{j,r,i,2})},\boldsymbol{\xi}_{i,2}\rangle\big)\cdot\|\boldsymbol{\xi}_{i,2}\|_2^2\right| \leq \frac{3\eta\sigma_{p,2}^2 d}{2N_2 m} \leq \log(T^{**}),$$

where the first inequality is by Lemma C.1 and the second inequality is by the condition of $\eta$ in Condition 4.1. For the third term, note that when $t > t_{j,r,i,2}$,

$$\langle \mathbf{w}_{y_{i,2},r}^{D,(t)}, \boldsymbol{\xi}_{i,2} \rangle \geq \langle \mathbf{w}_{y_{i,2},r}^{D,(T^*+1)}, \boldsymbol{\xi}_{i,2} \rangle + \overline{\rho}_{j,r,i,2}^{D,(\widehat{t})} - 16 N_2 \sqrt{\frac{\log(4(N_1^2 + N_2^2)/\delta)}{d}} \log(T^{**})$$

$$\geq -2\sqrt{\log(12 m N_2/\delta)} \cdot \sigma_0 \sigma_{p,2} \sqrt{d} + 2\log(T^{**}) - 16(N_1 \frac{\sigma_{p,2}}{\sigma_{p,1}} + N_2) \sqrt{\frac{\log(4(N_1^2 + N_2^2)/\delta)}{d}} \log(T^{**})$$

$$\geq 1.8 \log(T^{**}), \tag{E.15}$$

where the first inequality is by Lemma E.3, the second inequality is by Lemma E.2 and the third inequality is by $\sqrt{\log(12 m N_2/\delta)} \cdot \sigma_0 \sigma_{p,2} \sqrt{d} \leq 0.1 \log(T^{**})$, $16(N_1 \frac{\sigma_{p,2}}{\sigma_{p,1}} + N_2) \sqrt{\frac{\log(4(N_1^2 + N_2^2)/\delta)}{d}} \log(T^{**}) \leq 0.1 \log(T^{**})$ from the Condition 4.1. Then, the gradient can be bounded as

$$|\ell_i^{(t)}| = \frac{1}{1 + e^{-y_{i,2}[F_{+1}(\mathbf{W}_{+1}^{D,(t)}, \mathbf{x}_{i,2}) - F_{-1}(\mathbf{W}_{-1}^{D,(t)}, \mathbf{x}_{i,2})]}}$$

$$\leq e^{-y_{i,2} F_{y_{i,2}}(\mathbf{W}_{+1}^{D,(t)}, \mathbf{x}_{i,2}) + 0.1}$$

$$= e^{-\frac{1}{m} \sum_{r=1}^{m} \sigma(\langle \mathbf{w}_{y_{i,2},r}^{D,(t)}, \boldsymbol{\xi}_{i,2} \rangle) + 0.1}$$

$$\leq e^{0.1} \cdot e^{-1.8 \log(T^{**})} \leq 2 e^{-1.8 \log(T^{**})},$$

where the first inequality is by Lemma D.3 that $\kappa_D \leq 0.2$, the second inequality is by equation E.15. Based on these results, we can bound the third term in equation E.14 as

$$\left| \sum_{t_{j,r,i,2} < t \leq \widehat{t}-1} \frac{\eta}{N_2 m} \ell_i^{\prime(t)} \cdot \sigma'(\langle \mathbf{w}_{j,r}^{D,(t)}, \boldsymbol{\xi}_{i,2} \rangle) \cdot \|\boldsymbol{\xi}_{i,2}\|_2^2 \right| \leq \frac{\eta T^{**}}{N_2 m} \cdot 2 e^{-1.8 \log(T^{**})} \cdot \frac{3 \sigma_{p,2}^2 d}{2}$$

$$\leq \frac{T^{**}}{(T^{**})^{1.8}} \cdot \frac{3 \eta \sigma_{p,2}^2 d}{N_2 m}$$

$$\leq 1 \leq \log(T^{**}),$$

where the first inequality is by the bound of $|\ell_i^{(t)}|$ and Lemma C.1, the second inequality is by the fact that $e^{-x} \leq 1/x, x > 0$ and the third inequality is by the selection of $\eta$ in Condition 4.1. Since then, we prove that $\overline{\rho}_{j,r,i,2}^{D,(\widehat{t})} \leq 4 \log(T^{**})$.

Next, we prove the second result. When $j = y_{i,2}$, we have $\underline{\rho}_{j,r,i,2}^{D,(\widehat{t})} = 0$. When $j \neq y_{i,2}$, If $\underline{\rho}_{j,r,i,2}^{D,(\widehat{t}-1)} \leq -2\sqrt{\log\left(\frac{12 m N_2}{\delta}\right)} \cdot \sigma_0 \sigma_{p,2} \sqrt{d} - (C_1 - 16)(N_1 \frac{\sigma_{p,2}}{\sigma_{p,1}} + N_2) \sqrt{\frac{\log(4(N_1^2 + N_2^2)/\delta)}{d}} \log(T^{**})$, by Lemma E.3, it holds that

$$\left| \langle \mathbf{w}_{j,r}^{D,(\widehat{t}-1)} - \mathbf{w}_{j,r}^{D,(T^*+1)}, \boldsymbol{\xi}_{i,2} \rangle - \underline{\rho}_{j,r,i,2}^{D,(\widehat{t}-1)} \right| \leq 16 N_2 \sqrt{\frac{\log(4(N_1^2 + N_2^2)/\delta)}{d}} \log(T^{**}).$$

Rearrange the inequality, we get

$$\langle \mathbf{w}_{j,r}^{D,(\widehat{t}-1)}, \boldsymbol{\xi}_{i,2} \rangle \leq \langle \mathbf{w}_{j,r}^{D,(0)}, \boldsymbol{\xi}_{i,2} \rangle + \underline{\rho}_{j,r,i,2}^{D,(\widehat{t}-1)} + 16 N_2 \sqrt{\frac{\log(4(N_1^2 + N_2^2)/\delta)}{d}} \log(T^{**})$$

$$\leq 0,$$

where the second inequality is by Lemma E.2 and $\underline{\rho}_{j,r,i,2}^{D,(\widehat{t}-1)} \leq -2\sqrt{\log\left(\frac{12 m N_2}{\delta}\right)} \cdot \sigma_0 \sigma_{p,2} \sqrt{d} - (C_1 - 16)(N_1 \frac{\sigma_{p,2}}{\sigma_{p,1}} + N_2) \sqrt{\frac{\log(4(N_1^2 + N_2^2)/\delta)}{d}} \log(T^{**})$. Then, by the update rule, it holds that

$$\underline{\rho}_{j,r,i,2}^{D,(\widehat{t})} = \underline{\rho}_{j,r,i,2}^{D,(\widehat{t}-1)} + \frac{\eta}{N_2 m} \ell_i^{\prime(\widehat{t}-1)} \cdot \sigma'(\langle \mathbf{w}_{j,r}^{D,(\widehat{t}-1)}, \boldsymbol{\xi}_{i,2} \rangle) \cdot \|\boldsymbol{\xi}_{i,2}\|_2^2$$

$$= \underline{\rho}_{j,r,i,2}^{D,(\widehat{t}-1)} \geq -2\sqrt{\log\left(\frac{12 m N_2}{\delta}\right)} \cdot \sigma_0 \sigma_{p,2} \sqrt{d} - C_1 (N_1 \frac{\sigma_{p,2}}{\sigma_{p,1}} + N_2) \sqrt{\frac{\log(4(N_1^2 + N_2^2)/\delta)}{d}} \log(T^{**}),$$

If $\underline{\rho}_{j,r,i,2}^{D,(\widehat{t}-1)} \geq -2\sqrt{\log\left(\frac{12mN_2}{\delta}\right)} \cdot \sigma_0\sigma_{p,2}\sqrt{d} - (C_1 - 16)(N_1\frac{\sigma_{p,2}}{\sigma_{p,1}} + N_2)\sqrt{\frac{\log\left(4(N_1^2+N_2^2)/\delta\right)}{d}}\log(T^{**})$, by the update rule, it holds that

$$\underline{\rho}_{j,r,i,2}^{D,(\widehat{t})} = \underline{\rho}_{j,r,i,2}^{D,(\widehat{t}-1)} + \frac{\eta}{N_2 m}\ell_i'^{(\widehat{t}-1)} \cdot \sigma'\left(\langle \mathbf{w}_{j,r}^{D,(\widehat{t}-1)}, \boldsymbol{\xi}_{i,2}\rangle\right) \cdot \|\boldsymbol{\xi}_{i,2}\|_2^2$$

$$\geq \underline{\rho}_{j,r,i,2}^{D,(\widehat{t}-1)} - \frac{3\eta\sigma_{p,2}^2 d}{2N_2 m}$$

$$\geq -2\sqrt{\log\left(\frac{12mN_2}{\delta}\right)} \cdot \sigma_0\sigma_{p,2}\sqrt{d} - C_1(N_1\frac{\sigma_{p,2}}{\sigma_{p,1}} + N_2)\sqrt{\frac{\log\left(4(N_1^2+N_2^2)/\delta\right)}{d}}\log(T^{**}),$$

where the first inequality uses the fact $-1 \leq \ell_i'^{(\widehat{t}-1)} \leq 0$ and Lemma C.1, and the second inequality is by the condition of $\eta$ in Condition 4.1.

Next, we prove the third result. We prove a stronger conclusion that for any $i^* \in S_{j,r}^{D,(0)}$, it holds that

$$\frac{\gamma_{j,r}^{D,(t)} - \gamma_{j,r}^{D,(T^*+1)}}{\overline{\rho}_{j,r,i^*,2}^t} \leq \frac{26N_2\|\mathbf{u}\|_2^2}{\sigma_{p,2}^2 d}.$$

Recall the update rule that

$$\gamma_{j,r}^{D,(\widehat{t})} = \gamma_{j,r}^{D,(\widehat{t}-1)} - \frac{\eta}{N_2 m}\sum_{i \in [N_2]}\ell_i'^{(\widehat{t}-1)} \cdot \sigma'\left(\langle \mathbf{w}_{j,r}^{D,(\widehat{t}-1)}, y_{i,2} \cdot (\mathbf{u} + \mathbf{v}_2)\rangle\right) \cdot \|\mathbf{u}\|_2^2$$

$$\leq \gamma_{j,r}^{D,(\widehat{t}-1)} - \frac{\eta}{N_2 m} \cdot 13N_2 \cdot \ell_{i^*}'^{(\widehat{t}-1)} \cdot \sigma'\left(\langle \mathbf{w}_{j,r}^{D,(\widehat{t}-1)}, y_{i^*,2} \cdot (\mathbf{u} + \mathbf{v}_2)\rangle\right) \cdot \|\mathbf{u}\|_2^2$$

where the inequality follows by $\ell_i'^{(t)}/\ell_k'^{(t)} \leq 13$ in Lemma D.4, and

$$\overline{\rho}_{j,r,i^*,2}^{D,(\widehat{t})} = \overline{\rho}_{j,r,i^*,2}^{D,(\widehat{t}-1)} - \frac{\eta}{N_2 m}\ell_{i^*}'^{(\widehat{t}-1)} \cdot \sigma'\left(\langle \mathbf{w}_{j,r}^{D,(\widehat{t}-1)}, \boldsymbol{\xi}_{i^*,2}\rangle\right) \cdot \|\boldsymbol{\xi}_{i^*,2}\|_2^2 \cdot \mathbf{1}\{y_{i^*,2} = j\}.$$

Compare the gradient, we have

$$\frac{\gamma_{j,r}^{D,(\widehat{t})} - \gamma_{j,r}^{D,(T^*+1)}}{\overline{\rho}_{j,r,i^*,2}^{D,(\widehat{t})}} \leq \max\left\{\frac{\gamma_{j,r}^{D,(\widehat{t}-1)} - \gamma_{j,r}^{D,(T^*+1)}}{\overline{\rho}_{j,r,i^*,2}^{D,(\widehat{t}-1)}}, \frac{13N_2 \cdot \ell_{i^*}'^{(\widehat{t}-1)} \cdot \sigma'\left(\langle \mathbf{w}_{j,r}^{D,(\widehat{t}-1)}, y_{i^*} \cdot (\mathbf{u} + \mathbf{v}_1)\rangle\right) \cdot \|\mathbf{u}\|_2^2}{\ell_{i^*}'^{(\widehat{t}-1)} \cdot \sigma'\left(\langle \mathbf{w}_{j,r}^{D,(\widehat{t}-1)}, \boldsymbol{\xi}_{i^*,2}\rangle\right) \cdot \|\boldsymbol{\xi}_{i^*,2}\|_2^2}\right\}$$

$$\leq \max\left\{\frac{\gamma_{j,r}^{D,(\widehat{t}-1)} - \gamma_{j,r}^{D,(T^*+1)}}{\overline{\rho}_{j,r,i^*,2}^{D,(\widehat{t}-1)}}, \frac{13N_2\|\mathbf{u}\|_2^2}{\|\boldsymbol{\xi}_{i^*,2}\|_2^2}\right\}$$

$$\leq \max\left\{\frac{\gamma_{j,r}^{D,(\widehat{t}-1)} - \gamma_{j,r}^{D,(T^*+1)}}{\overline{\rho}_{j,r,i^*,2}^{D,(\widehat{t}-1)}}, \frac{26N_2\|\mathbf{u}\|_2^2}{\sigma_{p,2}^2 d}\right\}$$

$$\leq \frac{26N_2\|\mathbf{u}\|_2^2}{\sigma_{p,2}^2 d},$$

where the first inequality is from two update rules, the second inequality is by $i^* \in S_{j,r}^{D,(0)}$, the third inequality is by Lemma C.1 and the last inequality use the induction $\frac{\gamma_{j,r}^{D,(\widehat{t}-1)}}{\overline{\rho}_{j,r,i^*,2}^{D,(\widehat{t}-1)}} \leq \frac{26N_2\|\mathbf{u}\|_2^2}{\sigma_{p,2}^2 d}$. Similarly, it holds that $\frac{\gamma_{j,r,2}^{D,(\widehat{t})}}{\overline{\rho}_{j,r,i^*,2}^{D,(\widehat{t})}} \leq \frac{26N_2\|\mathbf{v}_1\|_2^2}{\sigma_{p,2}^2 d}$. $\qquad\square$

**Proposition E.6.** *Under Condition 4.1, for $T^* + 1 \le t \le T^{**}$, it holds that*

$$0 \le \bar{\rho}_{j,r,i,2}^{D,(t)} \le 4\log(T^{**}), \tag{E.16}$$

$$0 \ge \underline{\rho}_{j,r,i,2}^{D,(t)} \ge -2\sqrt{\log(12mN_2/\delta)} \cdot \sigma_0 \sigma_{p,2} \sqrt{d} - C_1(N_1 \frac{\sigma_{p,2}}{\sigma_{p,1}} + N_2)\sqrt{\frac{\log(4(N_1^2 + N_2^2)/\delta)}{d}} \log(T^{**}) \ge -4\log(T^{**}), \tag{E.17}$$

$$0 \le \gamma_{j,r}^{D,(t)} - \widetilde{\gamma}_{j,r}^{D,(T^*+1)} \le \frac{C_2 N_2 \|\mathbf{u}\|_2^2}{\sigma_{p,2}^2 d} \log(T^{**}), \tag{E.18}$$

$$0 \le \gamma_{j,r,2}^{D,(t)} \le \frac{C_2 N_2 \|\mathbf{v}_2\|_2^2}{\sigma_{p,2}^2 d} \log(T^{**}), \tag{E.19}$$

*for all $r \in [m], j \in \{\pm 1\}, i \in [N_2]$, where $C_1$ and $C_2$ are two absolute constant. Besides, we also have the following results:*

1. $\frac{1}{m}\sum_{r=1}^m \left[\rho_{y_{i,2},r,i,2}^{D,(t)} - \bar{\rho}_{r,k,i,2}^{D,(t)}\right] \le \log(12) + \kappa_D + \sqrt{\log(2N_2/\delta)/m}$ *for all $i, k \in [n]$.*

2. $S_i^{D,(0)} \subseteq S_i^{D,(t)}$, *where* $S_i^{D,(t)} = \{r \in [m] : \langle \mathbf{w}_{y_i,r}^{D,(t)}, \boldsymbol{\xi}_{i,2}\rangle > 0\}$.

3. $S_{j,r}^{D,(0)} \subseteq S_{j,r}^{D,(t)}$, *where* $S_{j,r}^{D,(t)} = \{i \in [N_2] : y_{i,2} = j, \langle \mathbf{w}_{j,r}^{D,(t)}, \boldsymbol{\xi}_{i,2}\rangle > 0\}$.

4. $\ell_i'^{(t)}/\ell_k'^{(t)} \le 13$.

5. *A refined estimation of $\frac{1}{m}\sum_{r=1}^m \rho_{y_{i,2},r,i,2}^{D,(t)}$ and $\ell_i'^{(t)}$. It holds that*

$$\underline{x}_t^D \le \frac{1}{m}\sum_{r=1}^m \bar{\rho}_{y_{i,2},r,i,2}^{D,(t)} \le \overline{x}_t^D + \bar{c}^D/(1 + \bar{b}^D),$$

$$\frac{1}{1 + \underline{b}^D e^{\underline{x}_t^D}} \le -\ell_i'^{(t)} \le \frac{1}{1 + \bar{b}^D e^{\overline{x}_t^D}},$$

*where $\overline{x}_t^D, \underline{x}_t^D$ are the the unique solution of*

$$\overline{x}_t^D + \bar{b}^D e^{\overline{x}_t^D} = \bar{c}^D t + \bar{b}^D,$$

$$\underline{x}_t^D + \underline{b}^D e^{\underline{x}_t^D} = \underline{c}^D t + \underline{b}^D,$$

*and $\bar{b}^D = e^{-\kappa_D/2}, \bar{c}^D = \frac{3\eta\sigma_{p,2}^2 d}{2N_2 m}, \underline{b}^D = e^{\kappa_D/2}$ and $\underline{c}^D = \frac{\eta\sigma_{p,2}^2 d}{5N_2 m}$.*

**Lemma E.7** (Meng et al. (2024)). *It holds that*

$$\log\left(\frac{\eta\sigma_{p,2}^2 d}{8N_2 m}t + \frac{2}{3}\right) \le \overline{x}_t^D \le \log\left(\frac{2\eta\sigma_{p,2}^2 d}{N_2 m}t + 1\right),$$

$$\log\left(\frac{\eta\sigma_{p,2}^2 d}{8N_2 m}t + \frac{2}{3}\right) \le \underline{x}_t^D \le \log\left(\frac{2\eta\sigma_{p,2}^2 d}{N_2 m}t + 1\right),$$

*for the defined $\bar{b}^D, \bar{c}^D, \underline{b}^D, \underline{c}^D$.*

### E.2 SIGNAL LEARNING AND NOISE MEMORIZATION

In this part, we will give detailed analysis of signal learning and noise memorization of the second system.

**Lemma E.8.** *Under Condition 4.1, for $T^* + 1 \le t \le T^{**}$, $\langle \mathbf{w}_{j,r}^{D,(t)}, j(\mathbf{u} + \mathbf{v}_2)\rangle$ increases with t.*

*Proof.* By Definition B.1, it holds that

$$\langle \mathbf{w}_{j,r}^{D,(t)}, j(\mathbf{u} + \mathbf{v}_2)\rangle = \gamma_{j,r}^{D,(t)} + \gamma_{j,r,2}^{D,(t)}.$$

By the update rule in Task 2 in Lemma B.2, we know that $\gamma_{j,r}^{D,(t)}$ and $\gamma_{j,r,2}^{D,(t)}$ increase with $t$. So $\langle \mathbf{w}_{j,r}^{D,(t)}, j(\mathbf{u}+\mathbf{v}) \rangle$ increases with $t$. $\square$

**Lemma E.9.** *Under Condition 4.1, for $T^* + 1 \leq t \leq T^{**}$, it holds that*

$$\frac{\eta \|\mathbf{u}\|_2^2}{\underline{c}^D m} \underline{x}_{t-2}^D - \frac{2\eta \|\mathbf{u}\|_2^2}{m} \leq \gamma_{j,r}^{D,(t)} - \gamma_{j,r}^{D,(T^*+1)} \leq \frac{\eta \|\mathbf{u}\|_2^2}{\bar{c}^D m} \bar{x}_{t-1}^D - \frac{2\eta \|\mathbf{u}\|_2^2}{m},$$

$$\frac{\eta \|\mathbf{v}\|_2^2}{\underline{c}^D m} \underline{x}_{t-2}^D - \frac{2\eta \|\mathbf{v}_2\|_2^2}{m} \leq \gamma_{j,r,2}^{D,(t)} \leq \frac{\eta \|\mathbf{v}\|_2^2}{\bar{c}^D m} \bar{x}_{t-1}^D - \frac{2\eta \|\mathbf{v}_2\|_2^2}{m}.$$

*Proof.* By the update rule, it holds that

$$\gamma_{j,r}^{D,(t+1)} + \gamma_{j,r,2}^{D,(t+1)} = \gamma_{j,r}^{D,(t)} + \gamma_{j,r,2}^{D,(t)} - \frac{\eta}{N_2 m} \sum_{i'=1}^{N_2} \ell_{i'}'^{(t)} \cdot \sigma'\left(\langle \mathbf{w}_{j,r}^{D,(t)}, y_i(\mathbf{u}+\mathbf{v}_2) \rangle\right) \|\mathbf{u}+\mathbf{v}_2\|_2^2$$

$$\leq \gamma_{j,r}^{D,(t)} + \gamma_{j,r,2}^{D,(t)} + \frac{\eta \|\mathbf{u}+\mathbf{v}_2\|_2^2}{m} \frac{1}{1 + \bar{b}^D e^{\bar{x}_t^D}}$$

$$\leq \gamma_{j,r}^{D,(0)} + \gamma_{j,r,2}^{D,(0)} + \frac{\eta \|\mathbf{u}+\mathbf{v}_2\|_2^2}{m} \sum_{s=0}^{t} \frac{1}{1 + \bar{b}^D e^{\bar{x}_s^D}}$$

$$\leq \gamma_{j,r}^{D,(0)} + \gamma_{j,r,2}^{D,(0)} + \frac{\eta \|\mathbf{u}+\mathbf{v}_2\|_2^2}{m} \int_{s=0}^{t} \frac{1}{1 + \bar{b}^D e^{\bar{x}_s^D}} ds$$

$$\leq \gamma_{j,r}^{D,(0)} + \gamma_{j,r,2}^{D,(0)} + \frac{\eta \|\mathbf{u}+\mathbf{v}_2\|_2^2}{m} \int_{s=0}^{t} \frac{1}{\bar{c}^D} d\bar{x}_s^D$$

$$\leq \gamma_{j,r}^{D,(0)} + \gamma_{j,r,2}^{D,(0)} + \frac{\eta \|\mathbf{u}+\mathbf{v}_2\|_2^2}{\bar{c}^D m} \bar{x}_t^D - \frac{2\eta \|\mathbf{u}+\mathbf{v}_2\|_2^2}{m}$$

$$\leq \frac{\eta \|\mathbf{u}+\mathbf{v}_2\|_2^2}{\bar{c}^D m} \bar{x}_t^D - \frac{2\eta \|\mathbf{u}+\mathbf{v}_2\|_2^2}{m},$$

where the first inequality is by the fifth result in Lemma E.6, the second inequality is by summation and the forth inequality is by the definition of $\bar{x}_s^D$. On the other side, we have

$$\gamma_{j,r}^{D,(t+1)} + \gamma_{j,r,2}^{D,(t+1)} = \gamma_{j,r}^{D,(t)} + \gamma_{j,r,2}^{D,(t)} - \frac{\eta}{N_2 m} \sum_{i'=1}^{n} \ell_{i'}'^{(t)} \cdot \sigma'\left(\langle \mathbf{w}_{j,r}^{D,(t)}, y_i(\mathbf{u}+\mathbf{v}_2) \rangle\right) \|\mathbf{u}+\mathbf{v}_2\|_2^2$$

$$\geq \gamma_{j,r}^{D,(t)} + \gamma_{j,r,2}^{D,(t)} + \frac{\eta \|\mathbf{u}+\mathbf{v}_2\|_2^2}{m} \frac{1}{1 + \underline{b}^D e^{\underline{x}_t^D}}$$

$$\geq \gamma_{j,r}^{D,(0)} + \gamma_{j,r,2}^{D,(0)} + \frac{\eta \|\mathbf{u}+\mathbf{v}_2\|_2^2}{m} \sum_{s=0}^{t} \frac{1}{1 + \underline{b}^D e^{\underline{x}_s^D}}$$

$$\geq \gamma_{j,r}^{D,(0)} + \gamma_{j,r,2}^{D,(0)} + \frac{\eta \|\mathbf{u}+\mathbf{v}_2\|_2^2}{m} \int_{s=0}^{t-1} \frac{1}{1 + \underline{b}^D e^{\underline{x}_s^D}} ds$$

$$\geq \gamma_{j,r}^{D,(0)} + \gamma_{j,r,2}^{D,(0)} + \frac{\eta \|\mathbf{u}+\mathbf{v}_2\|_2^2}{m} \int_{s=0}^{t-1} \frac{1}{\underline{c}^D} d\underline{x}_s^D$$

$$\geq \gamma_{j,r}^{D,(0)} + \gamma_{j,r,2}^{D,(0)} + \frac{\eta \|\mathbf{u}+\mathbf{v}_2\|_2^2}{\underline{c}^D m} \underline{x}_{t-1}^D - \frac{2\eta \|\mathbf{u}+\mathbf{v}_2\|_2^2}{m}$$

$$\geq \frac{\eta \|\mathbf{u}+\mathbf{v}_2\|_2^2}{\underline{c}^D m} \underline{x}_{t-1}^D - \frac{2\eta \|\mathbf{u}+\mathbf{v}_2\|_2^2}{m},$$

where the first inequality is by the fifth result in Lemma D.4, the second inequality is by summation and the forth inequality is by the definition of $\underline{x}_s^D$. Since that $\mathbf{u} \perp \mathbf{v}_2$, we have

$$\gamma_{j,r}^{D,(t)} = \frac{\|\mathbf{u}\|_2^2}{\|\mathbf{u}+\mathbf{v}_2\|_2^2} (\gamma_{j,r}^{D,(t)} + \gamma_{j,r,2}^{D,(t)}),$$

$$\gamma_{j,r,2}^{D,(t)} = \frac{\|\mathbf{v}_2\|_2^2}{\|\mathbf{u}+\mathbf{v}_2\|_2^2} (\gamma_{j,r}^{D,(t)} + \gamma_{j,r,2}^{D,(t)}).$$

Then, we complete the proof. □

**Lemma E.10.** *Under Condition 4.1, for $T^* + 1 \le t \le T^{**}$, it holds that*

$$\frac{N_2}{12}(\underline{x}_{t-2}^D - \underline{x}_1^D) \le \sum_{i \in [N_2]} \bar{\rho}_{j,r,i,2}^{D,(t)} \le 5N_2 \bar{x}_{t-1}^D.$$

*Proof.* For $j = y_i$, it holds that

$$\begin{aligned}
\sum_{i \in [N_2]} \underline{\rho}_{j,r,i,2}^{D,(t+1)} &= \sum_{i \in [N_2]} \underline{\rho}_{j,r,i,2}^{D,(t)} - \sum_{i \in [N_2]} \frac{\eta}{N_2 m} \ell_i'^{(t)} \cdot \sigma'\big(\langle \mathbf{w}_{j,r}^{D,(t)}, \boldsymbol{\xi}_{i,2} \rangle\big) \cdot \|\boldsymbol{\xi}_{i,2}\|_2^2 \\
&= \sum_{i \in [N_2]} \underline{\rho}_{j,r,i,2}^{D,(t)} - \sum_{i \in S_{j,r}^{D,(t)}} \frac{\eta}{N_2 m} \ell_i'^{(t)} \cdot \|\boldsymbol{\xi}_{i,2}\|_2^2 \\
&\ge \sum_{i \in [N_2]} \underline{\rho}_{j,r,i,2}^{D,(t)} + |S_{j,r}^{D,(0)}| \frac{\eta}{N_2 m} \frac{1}{1 + \underline{b}^D \underline{x}_t^D} \cdot \|\boldsymbol{\xi}_{i,2}\|_2^2 \\
&\ge \sum_{s=1}^{t} |S_{j,r}^{D,(0)}| \frac{\eta}{N_2 m} \frac{1}{1 + \underline{b}^D \underline{x}_s^D} \cdot \|\boldsymbol{\xi}_{i,2}\|_2^2 \\
&\ge \int_{s=1}^{t-1} |S_{j,r}^{D,(0)}| \frac{\eta}{N_2 m} \frac{1}{1 + \underline{b}^D \underline{x}_s^D} \cdot \|\boldsymbol{\xi}_{i,2}\|_2^2 ds \\
&\ge \frac{N_2}{12}(\underline{x}_{t-1}^D - \underline{x}_1^D),
\end{aligned}$$

where the first inequality is by $|S_{j,r}^{D,(t)}| \ge |S_{j,r}^{D,(0)}|$ and the fifth result in Lemma E.6, the second inequality is by summation and the forth inequality is by the definition of $\underline{x}_s^D$. On the other side, it holds that

$$\begin{aligned}
\sum_{i \in [N_2]} \underline{\rho}_{j,r,i,2}^{D,(t+1)} &\le \sum_{i \in [N_2]} \underline{\rho}_{j,r,i,2}^{D,(t)} + N_2 \frac{\eta}{N_2 m} \frac{1}{1 + \bar{b}^D \bar{x}_t^D} \cdot \|\boldsymbol{\xi}_{i,2}\|_2^2 \\
&\le \sum_{s=1}^{t} N_2 \frac{\eta}{N_2 m} \frac{1}{1 + \bar{b}^D \bar{x}_s^D} \cdot \|\boldsymbol{\xi}_{i,2}\|_2^2 \\
&\le \int_{s=1}^{t} N_2 \frac{\eta}{N_2 m} \frac{1}{1 + \bar{b}^D \bar{x}_s^D} \cdot \|\boldsymbol{\xi}_{i,2}\|_2^2 ds \\
&\le 5N_2(\bar{x}_t^D - \bar{x}_1^D) \\
&\le 5N_2 \bar{x}_t^D,
\end{aligned}$$

where the first inequality is by the fifth result in Lemma E.6, the second inequality is by summation and the forth inequality is by the definition of $\bar{x}_s^D$. Then, we complete the proof. □

## E.3 TEST ERROR ANALYSIS

**Lemma E.11** (Devroye et al. (2018)). *The TV distance between $\mathcal{N}(0, \sigma_{p,2}^2 \mathbf{I}_d)$ and $\mathcal{N}(\mathbf{v}, \sigma_{p,2}^2 \mathbf{I}_d)$ satisfies*

$$\mathrm{TV}\big(\mathcal{N}(0, \sigma_{p,2}^2 \mathbf{I}_d), \mathcal{N}(\mathbf{v}, \sigma_{p,2}^2 \mathbf{I}_d)\big) \le \frac{\|\mathbf{v_2}\|_2}{2\sigma_{p,2}}.$$

**Theorem E.12.** *For task1 and task2 with*

$$T_1 = \frac{C_1^* N_1 m}{\eta \sigma_{p,1}^2 d}, T_2 = \frac{C_2^* N_2 m}{\eta \sigma_{p,2}^2 d},$$

*where $C_1^*, C_2^*$ are two absolute constants. Then, it holds that:*

    *1. The training loss is below $\varepsilon$: $L_S(\mathbf{W}^{D,(t)}) \le \varepsilon$.*

2. *If*

$$d \leq C' \frac{\frac{\alpha^2 N_1^2 \|\mathbf{u}\|_2^4}{\sigma_{p,1}^4} + \frac{N_2^2 \|\mathbf{u}+\mathbf{v}_2\|_2^4}{\sigma_{p,2}^4}}{\frac{\alpha^2 \sigma_{p,2}^2 N_1}{\sigma_{p,1}^2} + N_2},$$

*the test error converges to 0: For any new data $(x, y)$,*

$$\mathbb{P}(yf(\mathbf{W}^{D,(t)}, x) < 0) \leq \exp\{-C_2 \frac{1}{d} \frac{\frac{\alpha^2 N_1^2 \|\mathbf{u}\|_2^4}{\sigma_{p,1}^4} + \frac{N_2^2 \|\mathbf{u}+\mathbf{v}_2\|_2^4}{\sigma_{p,2}^4}}{\frac{\alpha^2 \sigma_{p,2}^2 N_1}{\sigma_{p,1}^2} + N_2}\}.$$

3. *If*

$$d \geq C'' \frac{\frac{\alpha^2 N_1^2 \|\mathbf{u}\|_2^4}{\sigma_{p,1}^4} + \frac{N_2^2 \|\mathbf{u}+\mathbf{v}_2\|_2^4}{\sigma_{p,2}^4}}{\frac{\alpha^2 \sigma_{p,2}^2 N_1}{\sigma_{p,1}^2} + N_2},$$

*the test error only achieves a sub-optimal error rate: For any new data $(\mathbf{x}, y)$, $\mathbb{P}(yf(\mathbf{W}^{D,(t)}, x) < 0) \geq 0.1$.*

*Proof.* For the first result, by Lemma E.4, we have

$$
\begin{aligned}
y_{i,2} f(\mathbf{W}^{D,(t)}, \mathbf{x}_{i,2}) &\geq -\kappa_D/2 + \frac{1}{m} \sum_{r=1}^m \bar{\rho}_{j,r,i,2}^{D,(t)} \\
&\geq -\kappa_D/2 + \underline{x}_t^D \\
&\geq -\kappa_D/2 + \log\left(\frac{\eta \sigma_{p,1}^2 d}{8 N_1 m} t + \frac{2}{3}\right),
\end{aligned}
$$

where the first inequality is by Lemma E.4, the second inequality is by Proposition E.6 and the third inequality is by Lemma E.7. Then, we can calculate the training loss as

$$
\begin{aligned}
L(\mathbf{W}^{D,(t)}) &\leq \log\left(1 + e^{\kappa_D/2 - \log\left(\frac{\eta \sigma_{p,1}^2 d}{8 N_1 m} t + \frac{2}{3}\right)}\right) \\
&\leq \frac{e^{\kappa_D/2}}{\frac{\eta \sigma_{p,1}^2 d}{8 N_1 m} t + \frac{2}{3}} \\
&\leq \frac{e^{\kappa_D/2}}{2/\varepsilon + 1.5} \\
&\leq \varepsilon,
\end{aligned}
$$

where the second inequality uses the fact that $\log(1+x) \leq x, x \geq 0$, the third inequality is by $T_2 \geq \Omega\left(\frac{N_2 m}{\eta \sigma_{p,2}^2 d}\right)$ and the last inequality is by $\kappa_D \leq 0.1$. Then we complete the proof of the first result.

Next, for the second result, for data $(\mathbf{x}, y) \sim \mathcal{D}$, we have

$$
\begin{aligned}
yf(\mathbf{W}^{D,(t)}, \mathbf{x}) &\geq \frac{1}{m} \sum_{r=1}^m \sigma(\langle \mathbf{w}_{+1,r}^{D,(t)}, \mathbf{u}+\mathbf{v}_2\rangle) - \frac{1}{m} \sum_{r=1}^m \sigma(\langle \mathbf{w}_{-1,r}^{D,(t)}, \boldsymbol{\xi}\rangle) \\
&\quad - \frac{1}{m} \sum_{r=1}^m \sigma(\langle \mathbf{w}_{-1,r}^{D,(t)}, \mathbf{u}+\mathbf{v}_2\rangle) \\
&\geq -2\sqrt{\log(12m/\delta)} \cdot \sigma_0 \|\mathbf{u}+\mathbf{v}_2\|_2 + c\left(\frac{\alpha N_1 \|\mathbf{u}\|_2^2}{\sigma_{p,1}^2 d} + \frac{N_2 \|\mathbf{u}+\mathbf{v}_2\|_2^2}{\sigma_{p,2}^2 d}\right) \cdot \bar{x}_{t-1} - \frac{2\eta \|\mathbf{u}+\mathbf{v}_2\|_2^2}{m} \\
&\quad - \frac{1}{m} \sum_{r=1}^m \sigma(\langle \mathbf{w}_{-1,r}^{D,(t)}, \boldsymbol{\xi}\rangle) - \frac{1}{m} \sum_{r=1}^m \sigma(\langle \mathbf{w}_{-1,r}^{D,(t)}, \mathbf{u}+\mathbf{v}_2\rangle),
\end{aligned}
$$

where the first inequality is by $F_{y,r}(\mathbf{W}^{D,(t)}, \boldsymbol{\xi}) \geq 0$, and the second inequality is by the growth of the signal and $|\{r \in [m], \langle \mathbf{w}_{+1,r}^{D,(0)}, \mathbf{u} + \mathbf{v}_2 \rangle > 0\}|/m \geq 1/3$. Then for $\overline{x}_t \geq \underline{x}_t \geq C > 0$, it holds that

$$yf(\mathbf{W}^{D,(t)}, \mathbf{x}) \geq c\left(\frac{\alpha N_1 \|\mathbf{u}\|_2^2}{\sigma_{p,1}^2 d} + \frac{N_2 \|\mathbf{u} + \mathbf{v}_2\|_2^2}{\sigma_{p,2}^2 d}\right) \cdot \overline{x}_{t-1} - \frac{1}{m}\sum_{r=1}^m \sigma(\langle \mathbf{w}_{-1,r}^{D,(t)}, \boldsymbol{\xi} \rangle)$$

$$- \frac{1}{m}\sum_{r=1}^m \sigma(\langle \mathbf{w}_{-1,r}^{D,(t)}, \mathbf{u} + \mathbf{v}_2 \rangle)$$

$$- 2\sqrt{\log(12mn/\delta)} \cdot \sigma_0 \|\mathbf{u} + \mathbf{v}_2\|_2$$

$$\geq c\left(\frac{\alpha N_1 \|\mathbf{u}\|_2^2}{\sigma_{p,1}^2 d} + \frac{N_2 \|\mathbf{u} + \mathbf{v}_2\|_2^2}{\sigma_{p,2}^2 d}\right) \cdot \overline{x}_{t-1} - \frac{1}{m}\sum_{r=1}^m \sigma(\langle \mathbf{w}_{-1,r}^{D,(t)}, \boldsymbol{\xi} \rangle)$$

$$- 4\sqrt{\log(12mn/\delta)} \cdot \sigma_0 \|\mathbf{u} + \mathbf{v}_2\|_2 - 2\eta\|\mathbf{u} + \mathbf{v}_2\|_2^2/m$$

$$\geq \frac{c}{2}\left(\frac{\alpha N_1 \|\mathbf{u}\|_2^2}{\sigma_{p,1}^2 d} + \frac{N_2 \|\mathbf{u} + \mathbf{v}_2\|_2^2}{\sigma_{p,2}^2 d}\right) \cdot \overline{x}_{t-1} - \frac{1}{m}\sum_{r=1}^m \sigma(\langle \mathbf{w}_{-1,r}^{D,(t)}, \boldsymbol{\xi} \rangle).$$

Here, the first inequality is by the condition of $\sigma_0, \eta$ in Condition 3.1, and $\overline{x}_{t-1} \geq C$, the third inequality is still by the condition of $\sigma_0, \eta$ in Condition 4.1 which indicates that $\frac{c}{2}\left(\frac{\alpha N_1 \|\mathbf{u}\|_2^2}{\sigma_{p,1}^2 d} + \frac{N_2 \|\mathbf{u}+\mathbf{v}_2\|_2^2}{\sigma_{p,2}^2 d}\right) \cdot \overline{x}_{t-1} - 4\sqrt{\log(12mn/\delta)} \cdot \sigma_0 \|\mathbf{u} + \mathbf{v}_2\|_2 - 2\eta\|\mathbf{u} + \mathbf{v}_2\|_2^2/m \geq 0$. We denote by $h(\boldsymbol{\xi}) = \frac{1}{m}\sum_{r=1}^m \sigma(\langle \mathbf{w}_{-1,r}^{D,(t)}, \boldsymbol{\xi} \rangle)$. By Theorem 5.2.2 in Vershynin (2018), we have

$$P(h(\boldsymbol{\xi}) - Eh(\boldsymbol{\xi}) \geq x) \leq \exp\left(-\frac{c'x^2}{\sigma_{p,2}^2\|h\|_{\text{Lip}}^2}\right).$$

Here $c'$ is some constant. By

$$d \leq C_1 \frac{\frac{\alpha^2 N_1^2\|\mathbf{u}\|_2^4}{\sigma_{p,1}^4} + \frac{N_2^2\|\mathbf{u}+\mathbf{v}_2\|_2^4}{\sigma_{p,2}^4}}{\frac{\alpha^2\sigma_{p,2}^2 N_1}{\sigma_{p,1}^2} + N_2},$$

for some sufficiently large $C_1$ and Proposition E.5, we directly have

$$C\left(\frac{\alpha N_1 \|\mathbf{u}\|_2^2}{\sigma_{p,1}^2 d} + \frac{N_2 \|\mathbf{u} + \mathbf{v}_2\|_2^2}{\sigma_{p,2}^2 d}\right) \cdot \overline{x}_{t-1} \geq Eh(\boldsymbol{\xi}) = \frac{\sigma_{p,2}}{\sqrt{2\pi}m}\sum_{r=1}^m \|\mathbf{w}_{-1,r}^{D,(t)}\|_2,$$

where $\|\mathbf{w}_{-1,r}^{D,(t)}\|_2 \leq \Theta\left(\frac{\alpha}{\sigma_{p,1}d^{1/2}N_1^{1/2}}\sum_{i\in[N_1]}\overline{\rho}_{j,r,i,1}^{D,(t)} + \frac{1}{\sigma_{p,2}d^{1/2}N_2^{1/2}}\sum_{i\in[N_2]}\overline{\rho}_{j,r,i,2}^{D,(t)}\right).$

Now using methods in equation F.3 we get that

$$P(yf(\mathbf{W}^{D,(t)}, \mathbf{x}) < 0)$$

$$\leq P\left(h(\boldsymbol{\xi}) - Eh(\boldsymbol{\xi}) > \sum_r \sigma(\langle \mathbf{w}_{y,r}^{D,(t)}, y(\mathbf{u} + \mathbf{v}_2)\rangle) - \frac{\sigma_{p,2}}{\sqrt{2\pi}m}\sum_{r=1}^m \|\mathbf{w}_{-1,r}^{D,(t)}\|_2\right)$$

$$\leq \exp\left[-\frac{c''(\sum_r \sigma(\langle \mathbf{w}_{y,r}^{D,(t)}, y(\mathbf{u} + \mathbf{v}_2)\rangle) - \frac{\sigma_{p,2}}{\sqrt{2\pi}m}\sum_{r=1}^m \|\mathbf{w}_{-1,r}^{D,(t)}\|_2)^2}{\sigma_{p,2}^2\left(\sum_{r=1}^m \|\mathbf{w}_{-1,r}^{D,(t)}\|_2\right)^2}\right]$$

$$\leq \exp(c''/(2\pi))\exp\left[-\frac{c''\left(\sum_r \sigma(\langle \mathbf{w}_{y,r}^{D,(t)}, y(\mathbf{u} + \mathbf{v}_2)\rangle)\right)^2}{\sigma_{p,2}^2\left(\sum_{r=1}^m \|\mathbf{w}_{-1,r}^{D,(t)}\|_2\right)^2}\right]$$

$$\leq \exp\left[-C_2\frac{1}{d}\frac{\frac{\alpha^2 N_1^2\|\mathbf{u}\|_2^4}{\sigma_{p,1}^4} + \frac{N_2^2\|\mathbf{u}+\mathbf{v}_2\|_2^4}{\sigma_{p,2}^4}}{\frac{\alpha^2\sigma_{p,2}^2 N_1}{\sigma_{p,1}^2} + N_2}\right].$$

Here, $C_2 = O(1)$ is some constant. The first inequality is directly by equation F.2, the second inequality is by equation F.3 and the last inequality is by Proposition E.2 which directly gives the lower bound of signal learning and Proposition E.5 which directly gives the scale of $\|\mathbf{w}_{-1,r}^{D,(t)}\|_2$. Combined the results with equation F.1, we have

$$P(yf(\mathbf{W}^{D,(t)}, \mathbf{x}) < 0) \leq \exp\left[ -C_2 \frac{1}{d} \frac{\frac{\alpha^2 N_1^2 \|\mathbf{u}\|_2^4}{\sigma_{p,1}^4} + \frac{N_2^2 \|\mathbf{u}+\mathbf{v}_2\|_2^4}{\sigma_{p,2}^4}}{\frac{\alpha^2 \sigma_{p,2}^2 N_1}{\sigma_{p,1}^2} + N_2} \right].$$

Next, for the third result, we have

$$\mathbb{P}_{(\mathbf{x},y)\sim\mathcal{D}}\big(y \neq \text{sign}(f(\mathbf{W}^{D,(t)}, \mathbf{x}))\big) = \mathbb{P}_{(\mathbf{x},y)\sim\mathcal{D}}\big(yf(\mathbf{W}^{D,(t)}, \mathbf{x}) \leq 0\big).$$

$$= \mathbb{P}_{(\mathbf{x},y)\sim\mathcal{D}}\bigg( \sum_r \sigma(\langle \mathbf{w}_{-y,r}^{D,(t)}, \boldsymbol{\xi}\rangle) - \sum_r \sigma(\langle \mathbf{w}_{y,r}^{D,(t)}, \boldsymbol{\xi}\rangle)$$

$$\geq \sum_r \sigma(\langle \mathbf{w}_{y,r}^{D,(t)}, y(\mathbf{u}+\mathbf{v}_2)\rangle) - \sum_r \sigma(\langle \mathbf{w}_{-y,r}^{D,(t)}, y(\mathbf{u}+\mathbf{v}_2)\rangle) \bigg)$$

$$\geq 0.5\mathbb{P}_{(\mathbf{x},y)\sim\mathcal{D}}\bigg( \bigg| \sum_r \sigma(\langle \mathbf{w}_{+1,r}^{D,(t)}, \boldsymbol{\xi}\rangle) - \sum_r \sigma(\langle \mathbf{w}_{-1,r}^{D,(t)}, \boldsymbol{\xi}\rangle) \bigg|$$

$$\geq \max\bigg\{ \sum_r \sigma(\langle \mathbf{w}_{+1,r}^{D,(t)}, (\mathbf{u}+\mathbf{v}_2)\rangle), \sum_r \sigma(\langle \mathbf{w}_{-1,r}^{D,(t)}, (\mathbf{u}+\mathbf{v}_2)\rangle) \bigg\} \bigg),$$

where $C_6$ is a constant, the inequality holds since if $\left| \sum_r \sigma(\langle \mathbf{w}_{+1,r}^{D,(t)}, \boldsymbol{\xi}\rangle) - \sum_r \sigma(\langle \mathbf{w}_{-1,r}^{D,(t)}, \boldsymbol{\xi}\rangle) \right|$ is too large that we can always pick a corresponding $y$ given $\boldsymbol{\xi}$ to make a wrong prediction. Let $g(\boldsymbol{\xi}) = \sum_r \sigma(\langle \mathbf{w}_{1,r}^{D,(t)}, \boldsymbol{\xi}\rangle) - \sum_r \sigma(\langle \mathbf{w}_{-1,r}^{D,(t)}, \boldsymbol{\xi}\rangle)$. Denote the set

$$\Omega := \bigg\{ \boldsymbol{\xi} \bigg| |g(\boldsymbol{\xi})| \geq \max\bigg\{ \sum_r \sigma(\langle \mathbf{w}_{+1,r}^{D,(t)}, (\mathbf{u}+\mathbf{v}_2)\rangle), \sum_r \sigma(\langle \mathbf{w}_{-1,r}^{D,(t)}, (\mathbf{u}+\mathbf{v}_2)\rangle) \bigg\} \bigg\}.$$

By plugging the definition of $\Omega$, we have

$$\mathbb{P}_{(\mathbf{x},y)\sim\mathcal{D}}\big(yf(\mathbf{W}^{D,(t)}, \mathbf{x}) \leq 0\big) \geq 0.5\mathbb{P}(\Omega)$$

Next, we will give a lower bound of $\mathbb{P}(\Omega)$. We will prove that for a vector $\boldsymbol{\xi}'$ with $\|\boldsymbol{\xi}'\|_2 \leq 0.02\sigma_{p,2}$

$$\sum_j [g(j\boldsymbol{\xi} + \boldsymbol{\xi}') - g(j\boldsymbol{\xi})] \geq 4\max\bigg\{ \sum_r \sigma(\langle \mathbf{w}_{+1,r}^{D,(t)}, (\mathbf{u}+\mathbf{v}_2)\rangle), \sum_r \sigma(\langle \mathbf{w}_{-1,r}^{D,(t)}, (\mathbf{u}+\mathbf{v}_2)\rangle) \bigg\}$$

Therefore, by pigeon's hole principle, there must exist one of the $\boldsymbol{\xi}, \boldsymbol{\xi}+\boldsymbol{\xi}', -\boldsymbol{\xi}, -\boldsymbol{\xi}+\boldsymbol{\xi}'$ belongs $\Omega$.

$$|\mathbb{P}(\Omega) - \mathbb{P}(\Omega - \boldsymbol{\xi}')| = |\mathbb{P}_{\boldsymbol{\xi}\sim\mathcal{N}(0,\sigma_{p,2}^2\mathbf{I}_d)}(\boldsymbol{\xi} \in \Omega) - \mathbb{P}_{\boldsymbol{\xi}\sim\mathcal{N}(\boldsymbol{\xi}',\sigma_{p,2}^2\mathbf{I}_d)}(\boldsymbol{\xi} \in \Omega)|$$

$$\leq \text{TV}(\mathcal{N}(0,\sigma_{p,2}^2\mathbf{I}_d), \mathcal{N}(\boldsymbol{\xi}', \sigma_{p,2}^2\mathbf{I}_d))$$

$$\leq \frac{\|\boldsymbol{\xi}'\|_2}{2\sigma_{p,2}}$$

$$\leq 0.01,$$

where the first inequality is by the definition of Total variation (TV) distance, the second inequality is by Lemma E.11. Therefore, $\mathbb{P}(\Omega) \geq 0.24$ and then, it holds that

$$\mathbb{P}_{(\mathbf{x},y)\sim\mathcal{D}}\big(yf(\mathbf{W}^{D,(t)}, \mathbf{x}) \leq 0\big) \geq 0.1.$$

Now, all that's left is to prove the existence of $\boldsymbol{\xi}'$. Define $\lambda = C\frac{\alpha N_1 \|\mathbf{u}\|_2^2/(\sigma_{p,1}^2 d) + N_2 \|\mathbf{u}+\mathbf{v}_2\|_2^2/(\sigma_{p,2}^2 d)}{\alpha\sigma_{p,2} N_1/\sigma_{p,1} + N_2}$ and $\boldsymbol{\xi}'$ as

$$\boldsymbol{\xi}' = \lambda \cdot \sum_{i\in[N_2]} \mathbf{1}(y_i = 1)\boldsymbol{\xi}_i.$$

Then, we have

$$\|\boldsymbol{\xi}'\|_2 = \Theta\left(\frac{\alpha N_1 \|\mathbf{u}\|_2^2/(\sigma_{p,1}^2 d) + N_2 \|\mathbf{u}+\mathbf{v}_2\|_2^2/(\sigma_{p,2}^2 d)}{\alpha \sigma_{p,2} N_1/\sigma_{p,1} + N_2} \cdot \sqrt{N_2 \cdot \sigma_{p,2}^2 d}\right) \le 0.02\sigma_{p,2},$$

where the last inequality is by the condition

$$d \ge C \frac{\frac{\alpha^2 N_1^2 \|\mathbf{u}\|_2^4}{\sigma_{p,1}^4} + \frac{N_2^2 \|\mathbf{u}+\mathbf{v}_2\|_2^4}{\sigma_{p,2}^4}}{\frac{\alpha^2 \sigma_{p,2}^2 N_1}{\sigma_{p,1}^2} + N_2}.$$

Here, we use the fact that $a^2 + b^2 \le (a+b)^2 \le 2a^2 + 2b^2$ for positive $a, b > 0$, and we have for any sequences $a_n, b_n, c_n, d_n > 0$, $(a_n + b_n)^2/(c_n + d_n)^2 = \Theta((a_n^2 + b_n^2)/(c_n^2 + d_n^2))$. By the construction of $\boldsymbol{\xi}'$, we have almost surely that

$$\sigma(\langle \mathbf{w}_{+1,r}^{D,(t)}, \boldsymbol{\xi} + \boldsymbol{\xi}'\rangle) - \sigma(\langle \mathbf{w}_{+1,r}^{D,(t)}, \boldsymbol{\xi}\rangle)$$
$$+ \sigma(\langle \mathbf{w}_{+1,r}^{D,(t)}, -\boldsymbol{\xi} + \boldsymbol{\xi}'\rangle) - \sigma(\langle \mathbf{w}_{+1,r}^{D,(t)}, -\boldsymbol{\xi}\rangle)$$
$$\ge \langle \mathbf{w}_{+1,r}^{D,(t)}, \boldsymbol{\xi}'\rangle$$
$$\ge \lambda\left[\sum_{y_i=1} \overline{\rho}_{+1,r,i,1}^{D,(t)} + \sum_{y_i=1} \overline{\rho}_{+1,r,i,2}^{D,(t)} - o(1)\right], \tag{E.20}$$

where the first inequality is by the convexity of ReLU, and the second inequality is by Lemma C.2. Similarly, for $j = -1$, we have

$$\sigma(\langle \mathbf{w}_{-1,r}^{D,(t)}, \boldsymbol{\xi} + \boldsymbol{\xi}'\rangle) - \sigma(\langle \mathbf{w}_{-1,r}^{D,(t)}, \boldsymbol{\xi}\rangle)$$
$$+ \sigma(\langle \mathbf{w}_{-1,r}^{D,(t)}, -\boldsymbol{\xi} + \boldsymbol{\xi}'\rangle) - \sigma(\langle \mathbf{w}_{-1,r}^{D,(t)}, -\boldsymbol{\xi}\rangle)$$
$$\le 2|\langle \mathbf{w}_{-1,r}^{D,(t)}, \boldsymbol{\xi}'\rangle|$$
$$\le 2\lambda\left[\sum_{y_i=1} \underline{\rho}_{-1,r,i,1}^{D,(t)} + \sum_{y_i=1} \underline{\rho}_{-1,r,i,2}^{D,(t)} - o(1)\right], \tag{E.21}$$

where the first inequality is by the Lipschitz continuity of ReLU, and the second inequality is by Lemma C.2. Combining equation E.20 and equation E.21, we have

$$g(\boldsymbol{\xi} + \boldsymbol{\xi}') - g(\boldsymbol{\xi}) + g(-\boldsymbol{\xi} + \boldsymbol{\xi}') - g(-\boldsymbol{\xi}) \ge \lambda\left[\sum_r \sum_{y_i=1} \overline{\rho}_{1,r,i,1}^{D,(t)}/m + \sum_r \sum_{y_i=1} \overline{\rho}_{1,r,i,2}^{D,(t)}/m - o(1)\right] \tag{E.22}$$
$$\ge (\lambda/2) \cdot \sum_r \sum_{y_i=1} (\overline{\rho}_{1,r,i,1}^{D,(t)}/m + \overline{\rho}_{1,r,i,2}^{D,(t)}/m). \tag{E.23}$$

On the other side, we know that

$$\sum_r \sigma(\langle \mathbf{w}_{+1,r}^{D,(t)}, \mathbf{u}+\mathbf{v}_2\rangle)/m = \sum_{1 \le r \le \alpha m} \sigma(\langle \mathbf{w}_{+1,r}^{D,(t)}, \mathbf{u}+\mathbf{v}_2\rangle)/m + \sum_{\alpha m < r \le m} \sigma(\langle \mathbf{w}_{+1,r}^{D,(t)}, \mathbf{u}+\mathbf{v}_2\rangle)/m \tag{E.24}$$
$$\le \alpha\left(\frac{N_1 \|\mathbf{u}\|_2^2}{\sigma_{p,1}^2 d} \log(T^*) + \frac{N_2 \|\mathbf{u}+\mathbf{v}_2\|_2^2}{\sigma_{p,2}^2 d}\right) + (1-\alpha)\frac{N_2 \|\mathbf{u}+\mathbf{v}_2\|_2^2}{\sigma_{p,2}^2 d} \log(T^*) \tag{E.25}$$
$$= \left(\alpha \frac{N_1 \|\mathbf{u}\|_2^2}{\sigma_{p,1}^2 d} + \frac{N_2 \|\mathbf{u}+\mathbf{v}_2\|_2^2}{\sigma_{p,2}^2 d}\right) \log(T^*). \tag{E.26}$$

Comparing equation E.23 and equation E.26, by selecting $\lambda = C\frac{\alpha N_1 \|\mathbf{u}\|_2^2/(\sigma_{p,1}^2 d) + N_2 \|\mathbf{u}+\mathbf{v}_2\|_2^2/(\sigma_{p,2}^2 d)}{\alpha \sigma_{p,2} N_1/\sigma_{p,1} + N_2}$, we have

$$g(\boldsymbol{\xi} + \boldsymbol{\xi}') - g(\boldsymbol{\xi}) + g(-\boldsymbol{\xi} + \boldsymbol{\xi}') - g(-\boldsymbol{\xi}) \ge 4\sum_r \sigma(\langle \mathbf{w}_{+1,r}^{D,(t)}, \mathbf{u}+\mathbf{v}_2\rangle)/m$$

Since then, we complete the proof. □

## F OTHER EXPERIMENTS

We give additional experimental results about inherited parameters extracted from ViT models, which is shown in Figure 4. We also conduct experiments in Table 2, where the upstream task is image segmentation and the downstream task is image classification. For the upstream segmentation task, we use two models, deeplabv3_resnet50 and deeplabv3_mobilenet_v3_large, whose backbones are resnet50 and mobilenet_v3_large, respectively. For the downstream classification task, we use resnet50, resnet34, and mobilenet_v3_large. The results show that cross-task parameter transfer can also be beneficial, indicating the presence of shared knowledge across different tasks.

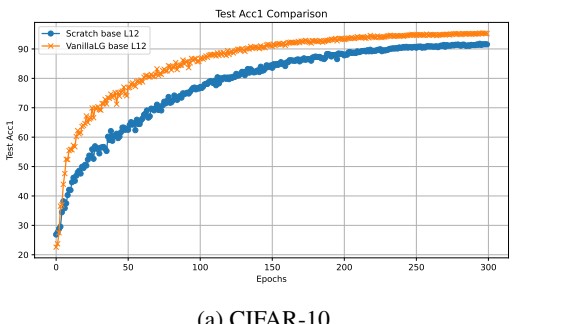
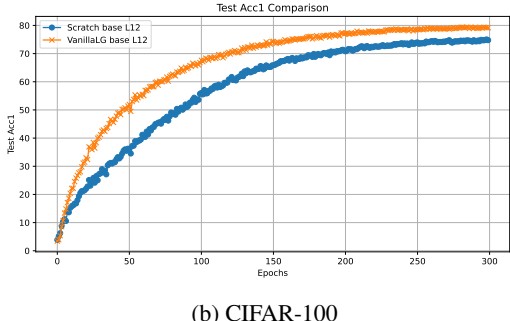

(a) CIFAR-10          (b) CIFAR-100

Figure 4: We adapt ViT models as the upstream model and downstream models. The upstream model is pre-trained on ImageNet-1K and the downstream models are trained on CIFAR-10 and CIFAR-100, separately.

Table 2: Transferring segmentation-pretrained backbones to CIFAR image classification. Upstream models (DeepLabV3 series) are pretrained on a COCO subset; downstream models are trained on CIFAR-10/100 with (w/ PL) and without (w/o PL) parameter transfer. Accuracy (%) is reported for each upstream–downstream pair.

| Dataset | Upstream Model | Downstream Model | Acc (w/o PL) | Acc (w/ PL) |
|---------|----------------|------------------|--------------|-------------|
| CIFAR10 | deeplabv3_resnet50 | resnet50 | 89.25 | 94.67 |
| | deeplabv3_resnet50 | resnet34 | 90.80 | 93.39 |
| | deeplabv3_mobilenet_v3_large | mobilenet_v3_large | 83.06 | 89.45 |
| CIFAR100 | deeplabv3_resnet50 | resnet50 | 70.45 | 75.31 |
| | deeplabv3_resnet50 | resnet34 | 68.35 | 73.96 |
| | deeplabv3_mobilenet_v3_large | mobilenet_v3_large | 66.64 | 72.24 |

## G DISCUSSION ABOUT TRANSFER LEARNING APPLICATIONS

**Transfer Learning Applications:** Transfer learning (TL) has emerged as a powerful paradigm in machine learning, aiming to leverage knowledge from a source domain to improve learning performance in a related but different target domain. Tan et al. (2015) introduces an intermediate domain to bridge source and target domains using non-negative matrix tri-factorization, enabling label propagation across heterogeneous spaces. Li et al. (2013) augments source and target features by projecting them into a common subspace while preserving domain-specific information, enabling knowledge transfer across different dimensions. Tsai et al. (2016) learns a transformation matrix to project source data into a PCA-based target subspace, aligning both marginal and conditional distributions for heterogeneous domain adaptation. Ye et al. (2021) rectifies heterogeneous model parameters by learning a semantic mapping function, enabling transfer of prior knowledge from source to target even with differing label spaces. In recent years, transfer Learning has found widespread applications across domains. Gardner et al. (2024) demonstrates how large-scale pretraining on a diverse tabular corpus enables strong zero-shot and few-shot generalization to unseen tabular tasks, effectively transferring knowledge across domains without task-specific fine-tuning. Wang et al. (2025) proposes a minimax-optimal transfer learning algorithm for nonparametric contextual dynamic pricing under covariate shift, leveraging source data to improve target-domain pricing decisions. Garau-Luis et al. (2024) presents a multi-modal

transfer learning framework that effectively bridges pre-trained DNA, RNA, and protein encoders to predict RNA isoform expression. (Wang et al., 2022) selects certain layers as learngene based on gradient information observed in the upstream model, and subsequently stacks these learngene layers with some randomly initialized layers to initialize downstream models. Li et al. (2023) proposes a scalable surrogate-model framework that learns linear relevance scores to predict which source tasks will cause negative transfer, enabling efficient subset selection that outperforms existing multi-task learning methods across weak supervision, NLP and fairness benchmarks. Imani et al. (2021) introduces the notion of the degree of alignment and investigates its relationship with transfer learning performance. It argues that neural networks automatically adjust their representations during training so that the top singular vectors align with the task labels, which is validated by the experiments. Instead, our work provides a theoretical explanation for the underlying dynamics. We obtain some similar findings with proof: a neural network memorizes both signal and noise during training, as shown in Lemma A.1 and Lemma A.2. The transferred parameters therefore retain the shared signal between the two tasks. When the norm of the shared signal becomes too small, negative transfer emerges. Our work theoretically characterizes and explains this dynamical process, while Imani et al. (2021) proposes the conjectures and then verifies it from an empirical perspective. The core viewpoints are different but share conceptual similarities.

## H    DISCUSSION ABOUT CONNECTIONS BETWEEN THEORY AND PRACTICE

In this section, we would like to give more discussion on the connection between our theory and the practice. Our intention is not to position theory as dominating practice, but to highlight how the two can develop hand in hand. For instance: the nagative transfer was first observed in practical applications, and Proposition 4.4 now provides a theoretical characterization that confirms its existence and explains when it arises. This illustrates how empirical phenomena can motivate theoretical inquiry, and how theory can, in turn, contextualize those empirical observations. By making this connection clearer, we hope to support a more integrated and mutually informative relationship between theory and practice. Moreover, our theoretical analysis could indicate several concrete inspirations for algorithm design.

- **Selecting task-specific neurons for transfer.** The dynamics reveal that neurons activated by the task-specific signal dominate the effective transfer. This suggests a more targeted strategy in practice: instead of transferring all parameters, one can identify neurons that are strongly activated by the downstream task (for example, via activation statistics or gradient-based criteria) and preferentially transfer or fine-tune only these neurons. Such neuron-level selection aims to retain parameters that encode task-relevant structure while mitigating the influence of noise, thereby improving the robustness of transfer.

- **Estimating transferability.** Our results indicate that transfer performance is primarily governed by the correlation between the structural components of the upstream and downstream tasks. In practice, this correlation can be estimated using a small downstream validation set by monitoring the early-stage learning curves under different initialization scales or different subsets of transferred parameters. Consistently slower or noisier initial improvements signal weak structural correlation and thus a higher risk of negative transfer, in which case one may reduce the amount of transferred parameters or fall back to training from scratch.

Regarding more realistic constraints such as layer-wise exposure or fixed adapter interfaces: fully analyzing multi-layer networks is mathematically challenging because interaction in different layers leads to unstable and complicated dynamics. However, the mechanisms revealed in the two layer case could also offer valuable insight. When the upstream task has high quality data with strong signal and large sample size, deeper models are expected to preserve and propagate this useful structure across layers. Our real data experiments results also support this conclusion aligning with our theoretical analysis. From this example, we would like to point out the core spirits of feature learning theory. Theoretical models in feature learning intentionally use simplified architectures to capture universal phenomenon in representation learning, rather than to replicate every detail of practical systems. Despite their simplicity, these models have repeatedly shown that early learned features or noise can strongly influence downstream performance, even in deep networks with high capacity and nonlinear expressiveness. Such results explain why shallow analyses remain valuable. They isolate core principles of signal learning, noise memorization, and transfer quality that extend to more complex architectures, though in more intricate forms which are harder to characterize with full mathematical rigor.

## I    THE USE OF LARGE LANGUAGE MODELS (LLMS)

We employed an LLM to refine the writing of the entire manuscript and to ensure its grammatical correctness.

