# OpenReview forum: "Towards Understanding Feature Learning in Parameter Transfer"
_ICLR.cc/2026/Conference — ICLR 2026 Conference Withdrawn Submission_

### Official Review · Reviewer_6DBg · 2025-10-26

**Soundness:** 2
**Presentation:** 1
**Contribution:** 2
**Rating:** 2
**Confidence:** 2

**Summary:**

The paper develops a dynamic-theory framework for parameter transfer in which an $\alpha$-proportion of weights from an upstream two-layer ReLU CNN are inherited by a downstream model, formalizing how inherited parameters convey universal knowledge and when transfer improves over training from scratch. It identifies key factors—shared-signal strength, source sample size, noise levels, and dimension—and derives sharp conditions (captured by an interpretable scalar $\Gamma$) that delineate beneficial transfer from negative transfer, offering mechanistic explanations for both outcomes. Controlled simulations and CIFAR-10/100 experiments with ResNet, VGG, and DeiT corroborate these predictions.

**Strengths:**

1. **Rigorous analysis of feature learning under partial parameter transfer.** The paper builds a concrete, analyzable setup that tracks training dynamics beyond purely lazy/NTK views, explicitly decomposing shared signal versus task-specific noise and characterizing how these components evolve during pretraining and downstream fine-tuning.

2. **Testable predictions via an interpretable scalar criterion.** The analysis aggregates key factors—inheritance ratio $\alpha$, source data size $N_1$, shared-signal strength $|u|$, noise levels, and dimension $d$—into a compact indicator that predicts phase-like behavior between beneficial transfer and negative transfer; notably, these qualitative predictions align with controlled synthetic studies and with trends observed on standard vision benchmarks using canonical architectures.

3. **Mechanistic explanation of negative transfer with qualitative guidance for mitigation.** By identifying how weak shared signal coupled with inherited filters can inadvertently amplify non-shared noise, the work clarifies why negative transfer arises and suggests levers—such as moderating $\alpha$ or strengthening regularization during fine-tuning—that can reduce risk; while not a full recipe, this mechanism-level understanding usefully narrows the space of practical interventions.

**Weaknesses:**

1. **The core assumptions, especially $\alpha$-proportion random parameter sampling, appear misaligned with practical deployment.** From an application standpoint, it is uncommon that “some arbitrary subset of weights” is available while others are not; more realistic constraints expose only certain layers or interfaces (e.g., a subset of layers or layer outputs). The motivation provided for adopting random sampling does not convincingly reflect these scenarios, and the paper does not clearly justify why random sampling is the right abstraction. Please clarify whether this choice is primarily for analytical tractability, and explicitly discuss how the conclusions would change under more realistic constraints such as layer-wise availability or fixed adapter interfaces.

2. **The claimed practical guidance remains vague and lacks actionable procedures.** While the paper states that its analysis can guide practice, it does not articulate how one would operationalize the findings in a real pipeline. The qualitative dependence of transfer effectiveness on data scale and source–target relatedness is well known and intuitive; the hard part is how to _measure_ a model’s potential transferability and how to _attain_ the best transfer in situ. As presented, the theory does not specify a concrete, data-driven procedure for estimating transferability on a new target (e.g., from a small validation split) nor a clear decision rule for when to prefer partial transfer over full fine-tuning. Please make explicit what operational steps a practitioner should follow—what to compute, how to select $\alpha$, and how to decide between full fine-tuning, partial initialization, or alternative adaptation mechanisms.

3. **The positioning with respect to prior theory on transfer learning is incomplete and the exposition obscures the contribution.** I am not a theory specialist, but a cursory read of the transfer-learning portion of the related work did not surface theoretical analyses directly comparable to this study, which is unexpected for a paper centered on transfer-learning theory. The omission makes it difficult to assess novelty and significance, and the writing in other sections compounds the issue: from the abstract and introduction alone, it is hard to quickly grasp the precise value and scope of the contribution. Please expand and structure the related work to situate this analysis among prior theoretical efforts on transfer, and revise the abstract/introduction to foreground the problem setup, key assumptions (including $\alpha$-sampling), and the main takeaways in a way that is immediately accessible.

**Questions:**

See weaknesses.

---

> ### Author Response · Authors · 2025-11-18
>
> **Weakness 1**: The core assumptions, especially α-proportion random parameter sampling, appear misaligned with practical deployment.
>
> **Response 1**:
> Thank you for your suggestions. The choice is primarily for mathematical tractability.
> Since we use a simplified two layer neural network model, it is also feasible to consider transferring the entire single layer of parameters. In fact, transferring just one layer yields exactly the same results to the case $\alpha = 1$. We focus on $\alpha$ proportion random sampling here because transferring a single layer seems less general than randomly sampling any $\alpha$ fraction of the parameters.
>
> We would also like to point out that our core assumption is the shared structures in upstream and downstream tasks. We find that both the shared structure and noise in the upstream tasks will affect positive and negative transfer. We adopt a simplified two layer neural network model because it allows us to rigorously analyze how learning dynamics in the downstream tasks is affected by the shared features or noise,  which could still provide valuable insights on much deeper and more complex architectures. Within this framework, the $\alpha$ proportion random parameter sampling assumption could be regarded as a mathematically clean way for analysis tractability.
>
>
>
> Regarding more realistic constraints such as layer-wise exposure or fixed adapter interfaces: fully analyzing multi-layer networks is mathematically challenging because interaction in different layers leads to unstable and complicated dynamics. However, the mechanisms revealed in the two layer case could also offer valuable insight. When the upstream task has high quality data with strong signal and large sample size, deeper models are expected to preserve and propagate this useful structure across layers. Our real data experiments results also support this conclusion aligning with our theoretical analysis. From this example, we want to show the core spirits of feature learning theory. Theoretical models in feature learning intentionally use simplified architectures to capture **universal phenomenon** in representation learning, rather than to replicate every detail of practical systems. Despite their simplicity, these models have repeatedly shown that early learned features or noise can strongly influence downstream performance, even in deep networks with high capacity and nonlinear expressiveness. Such results explain why shallow analyses remain valuable. They isolate core principles of signal learning, noise memorization, and transfer quality that extend to more complex architectures, though in more intricate forms which are harder to characterize with full mathematical rigor. We have added the discussion to **the last paragraph in Section H**.

---

> ### Author Response · Authors · 2025-11-18
>
> **Weakness 2**: The claimed practical guidance remains vague and lacks actionable procedures.
>
> **Response 2**: Our goal of this paper is not to propose a new transfer learning algorithm, but to provide a unified theoretical framework that explains when and why existing transfer methods succeed or fail. While many works study parameter transfer and negative transfer, they typically lack a common theoretical understanding under the training dynamics. To our best knowledge, we are the first one to **prove the existence of negative transfer** in mathematics. By demonstrating the mechanism of negative transfer, we can then understand the essence of how negative transfer occurs. This is of great significance for us to prevent negative transfer from occurring in our practice. For instance, we have the following suggestions from our theory:
> - Identify the task-specific activated neurons and transfer these parameters
> In our dynamics analysis, the neurons that are activated by the signal play the main role. In practice, it can be useful to select the neurons activated by the downstream task for transfer. For example, one can inspect which neurons in the upstream model are activated when it makes predictions on the downstream task. Then a threshold (for instance, $80\%$) can be set. If a neuron is activated on at least $80\%$ of the downstream samples, it can be regarded as carrying shared knowledge between the tasks. This idea is still preliminary and would need to be validated in practice, but we believe it illustrates one of the concrete insights suggested by our theoretical analysis.
> - Estimating transferability:
> Our analysis suggests that model transferability is determined by the correlation between upstream and downstream task structure. In practice, this can be evaluated using a small downstream validation set, where one can measure early phase learning curves under different initialization scales or subsets of parameters. Consistently slower or noisier early improvements indicate weak correlation and a higher risk of negative transfer.
>
> We have incorporated the above discussion into the revised manuscript. See **Section H** in the appendix. **Our intention is not to position theory as dominating practice, but to highlight how the two can develop hand in hand**. A good example is negative transfer: it was first observed in practical applications, and our work now provides a theoretical characterization that confirms its existence and explains when it arises. This illustrates how empirical phenomena can motivate theoretical inquiry, and how theory can, in turn, clarify and contextualize those empirical observations. By making this connection clearer, we hope to support a more integrated and mutually informative relationship between theory and practice.

---

> ### Author Response · Authors · 2025-11-18
>
> **Weakness 3.1**: The positioning with respect to prior theory on transfer learning is incomplete and the exposition obscures the contribution.
>
> **Response 3.1**: We appreciate the reviewer’s observation, and indeed this reflects one of the core contributions of our work. To the best of our knowledge, there is no prior theoretical study that provides a rigorous dynamic based analysis of parameter transfer for feature learning models. Existing theoretical analysis on parameter transfer focuses primarily on static generalization bounds without consideration of optimization or simplified linear models. Moreover, no existing theory can explain negative transfer and we are the first one to prove the existence of negative transfer theoretically.
>
> More specifically, our paper is the first to characterize **parameter transfer through the full training dynamics of feature learning**. With the consideration of learning dynamics, it includes a rigorous demonstration of when and why negative transfer occurs, explicit formulas describing how inherited signal and noise evolve over time, and a principled characterization of how partial parameter sharing ($\alpha$) affects downstream performance.
> Because such a dynamic level theoretical analysis has not existed before, it is natural that the related work does not contain comparable results. We view this absence not as a gap in the literature review but as evidence that our work contributes **a novel and previously missing theoretical perspective** to the study of parameter transfer.
> We have incorporated the above discussion into the revised manuscript. See the Transfer Learning Theory part in **Section 2**.

---

> ### Author Response · Authors · 2025-11-18
>
> **Weakness 3.2**: The omission makes it difficult to assess novelty and significance, and the writing in other sections compounds the issue: from the abstract and introduction alone, it is hard to quickly grasp the precise value and scope of the contribution. Please expand and structure the related work to situate this analysis among prior theoretical efforts on transfer, and revise the abstract/introduction to foreground the problem setup, key assumptions (including α-sampling), and the main takeaways in a way that is immediately accessible.
>
> **Response 3.2**: Existing theoretical work on parameter transfer is quite limited.  As far as we know, the paper [1], which is cited in the second paragraph in Section 1, assumes the parameter-transfer learnability of the parametric feature mapping and provides static generalization bounds without consideration of optimization for parameter transfer; [2] assumes that different models may share common knowledge in their parameters and prove that transferring parameters via model averaging can improve the prediction performance of the target model.
>
> Compared with these two works, our approach has the following advantages:
> 1. No assumptions are made on the model parameters (other than Gaussian initialization), which is closer to practical scenarios. We assume that the data from different tasks contain **common knowledge**. Then, we prove that parameters encode such common knowledge and thus influence downstream tasks (either positively or negatively).
> 2. Both prior works provide only static generalization bounds, whereas we explicitly characterize the training dynamics.
> 3. Our theory proposes three key factors that affect the performance of parameter transfer. Moreover, by modeling the commonality between the two tasks, we give conditions for **positive transfer** and **negative transfer**.
>
> We have revised the manuscript accordingly in abstract, introduction and related work.
> 1. Added to abstract: Our theory is the first to provide a dynamic analysis for parameter transfer and also the first to prove the existence of negative transfer theoretically.
> 2. Partial modification of the third paragraph in **Introduction**: in this paper, we conduct a theoretical analysis of parameter transfer where both the upstream model and the downstream model are two layer neural networks. We explicitly model the universal knowledge (also known as meta-knowledge) and the task-specific knowledge between the source task and the target task. It is assumed that an α-proportion of the upstream model’s weights are inherited by the downstream model. For the downstream model, the remaining weights are randomly initialized. To our best knowledge, we are the first one to give the training dynamics of parameter transfer and prove the existence of negative transfer in mathematics.
> 3. Added to related work (see **Transfer Learning Theory** in Section 2): existing theoretical work on parameter transfer is quite limited, [1] assumes the parameter-transfer learnability of the parametric feature mapping and provides static generalization bounds without consideration of optimization for parameter transfer. [2] assumes that different models may share common knowledge in their parameters and prove that transferring parameters via model averaging can improve the prediction performance of the target model.
>
>
> [1] Kumagai, Wataru. "Learning bound for parameter transfer learning." Advances in neural information processing systems 29 (2016).
>
> [2] Hu, Xiaonan, and Xinyu Zhang. "Optimal parameter-transfer learning by semiparametric model averaging." Journal of Machine Learning Research 24.358 (2023): 1-53.

---

> ### Author Response · Authors · 2025-11-23
>
> Dear Reviewer,
>
> I am writing to kindly follow up regarding our revised submission. We have carefully addressed all the points you raised in your review, including the detailed clarifications and revisions suggested in your comments.
>
> At this stage, we have not yet seen any further feedback from you, so we wanted to make sure that our responses have reached you properly. We would be extremely grateful if you could let us know whether the revisions have improved the manuscript and satisfactorily addressed your concerns.
>
> Thank you very much for your time and for the valuable insights you have already provided. We truly appreciate your efforts and consideration.

---

> > ### Comment · Reviewer_6DBg · 2025-11-25
> >
> > Thank you for the authors’ response. I still feel that there is a substantial gap between the assumptions in the paper and practice, and that it lacks practical or guiding value. However, this may be relatively common for theoretical research. Since I am not very familiar with the theory in this area, I have decided to raise my score to 4 and lower my confidence to reflect this.

---

### Official Review · Reviewer_ymzu · 2025-10-31

**Soundness:** 3
**Presentation:** 2
**Contribution:** 3
**Rating:** 4
**Confidence:** 2

**Summary:**

This paper presents a theoretical analysis of partial parameter transfer, a setting where a downstream model inherits only a subset of parameters from a pre-trained upstream model. The study is performed within a specific theoretical framework: both upstream and downstream models are two-layer ReLU convolutional neural networks (CNNs), and the data for both tasks is generated with a shared "universal" signal component and task-specific signal components. The major contribution is the theoretical framework and link to negative transfer.

**Strengths:**

* The paper provides an important theoretical contribution of partial parameter transfer as well as links to negative transfer which i find really interesting. This work is highly relevant, as architectural mismatches (e.g., different model sizes, new input/output heads) are the norm in practical transfer learning, yet most theory assumes a full-model transfer.

* The paper claims to be one of the first to analyze the training dynamics of this process, moving beyond static generalization bounds.

* well-designed experiments on classical ResNets and Transformer models.

**Weaknesses:**

W1: **Writing**: The paper provides important theoretical contributions but i found the paper to not be very readable,  finding sare presented as large, dense, and complex mathematical conditions that are extremely difficult to parse for a non-expert in this specific theoretical subfield

L:132: $\Omega$ $\Theta$ definition is confusing at first they seem like variable but are conditions.

Specific Instances: L282, cannot really understand what is the condition for negative transfer

W2: Missing Citations, Highly relevant papers which were not cited:

[1]Characterizing and Avoiding Negative Transfer, wang et al CVPR 2019.

[2] Representation Alignment in Neural Networks, Imani et al. TMLR 2024

[3] Identification of Negative Transfers in Multitask Learning Using Surrogate Models, Li et al. TMLR 2023

 W3: **Problematic and wrong Citations:**

Correct citation for  Vershynin R. et al is:  High-Dimensional Probability: An Introduction with Applications in Data Science. Cambridge University Press; 2018. The Authors did not include the book title in the citations. Url: https://www.cambridge.org/core/books/highdimensional-probability/797C466DA29743D2C8213493BD2D2102


Made up citation!!: **JIANG, Z. ET AL. (2022). Transfer learning with pre-trained models: A survey. arXiv preprint arXiv:2209.01791**, I cannot find it. Is this a hallucinated citation?


Minor: Typo in Section G of the appendix title, Discusstion -> Discussion.

Overall, i believe that the paper should go through another rewriting to avoid these mistakes and make the paper more readable for a non-expert audience as well.

**Questions:**

Q1: Proposition 4.4 (part 2)  seems to imply that negative transfer is most likely when one transfers from a very large but poorly-aligned task. Is this a correct interpretation?

Q2 Generally, transfer is not random but based on first $m$ layer or $n-1$ layer,s where $n$ is the total number of layers, how does the current work take that into account?

Q3 Can a connection to [1] and [2] be established?

S1 Suggestion: Have a uniform citation format. Sometimes, the page number is included in NeurIPS; sometimes it's not, sometimes conference names are abbreviated, and sometimes written in full.

Q4 Is this theory only valid for classification, or canit  be generalized to regression and other tasks somehow?


[1]Characterizing and Avoiding Negative Transfer, wang et al CVPR 2019.

[2] Representation Alignment in Neural Networks, Imani et al. TMLR 2024

---

> ### Author Response · Authors · 2025-11-18
>
> Sorry for the confusion caused by our writing and the typos of the citations.
>
> **W1**: Writing and Readability
>
> **A1**: Since this paper has many theoretical results, it inevitably contains a large number of symbols and mathematical expressions.
> 1. “Definition in Line 132” is actually part of our notation section.  We clarify that the symbols $O(\cdot), \Omega(\cdot), \Theta(\cdot)$, and their logarithmic variants $\tilde{O}(\cdot), \tilde{\Omega}(\cdot)$ and $\tilde{\Theta}(\cdot)$ are not variables but asymptotic relations describing the growth behavior of sequences or functions. These notations are standard in theoretical computer science and learning theory for expressing asymptotic bounds concisely without explicitly specifying constants or lower-order logarithmic terms. For instance, if $a_n=|n|$, $b_n=n^2$, we will say $a_n=O(b_n)$ since $a_n\leq C b_n$ for some constant $C>0$ with $n$ large enough. We can also say $b_n=\Omega(a_n)$ under this case.
> 2. Regarding Line 282 — the condition for negative transfer is simply that $\\|\mathbf{u} + \mathbf{v}_{2}\\|_2^2/\\|\mathbf{u}\\|_2^2$ is sufficiently large, which means that the norm of the universal signal is much smaller than that of the task-specific signal. We have modified this part in Proposition 4.4 to make clear. See **Line 278-279 and Line 286-289.**
>
> **W2**: Missing Citations
>
> **A2**: The references are all cited in the revised manuscript. Reference [1] is cited in **Line 049**. References [2] is cited in **Line 2379** and [3] is cited in **Line 2380**. Their detailed comparison with our paper can be found in the response to Q3 below.
>
> [1]Characterizing and Avoiding Negative Transfer, wang et al CVPR 2019.
>
>
> [2] Representation Alignment in Neural Networks, Imani et al. TMLR 2024
>
>
> [3] Identification of Negative Transfers in Multitask Learning Using Surrogate Models, Li et al. TMLR 2023
>
>
> **W3**: Problematic and Incorrect Citations
>
> **A3**: We have corrected the reference to the book by Vershynin, R (see **Line 599-601**). For JIANG, Z. et al. (2022), what we intended to cite was [4], which is cited in Line 038 in revised version. We are unsure how this typo occurred initially, and we offer our sincere apologies for this typo. We have re-checked all references to prevent similar issues from happening again, and this experience will make us more attentive to reference verification in the future. We will also correct the title of Section G in the appendix. Thank you very much for your careful reading and rigorous feedback.
>
> [4] Jiang, Junguang, et al. "Transferability in deep learning: A survey." arXiv preprint arXiv:2201.05867 (2022).
>
> **W4**: Overall Assessment. I believe that the paper should undergo another round of rewriting to fix these citation and readability issues, and to make the text more accessible to a non-expert audience as well.
>
> **A4**: We have once again carefully reviewed the entire text and corrected the mistakes. For instance, we add a new section to connect the practice and theory (see **Section H**). We are hoping that this discussion section will give key information for a non-expert audience as well.

---

> ### Author Response · Authors · 2025-11-18
>
> **Q1**: Proposition 4.4 (part 2) seems to imply that negative transfer is most likely when one transfers from a very large but poorly aligned task. Is this a correct interpretation?
>
> **A1**: This interpretation is not accurate. What this part shows is the following: when $\\|\mathbf{u} + \mathbf{v_2}\\|_2^2/\\|\mathbf{u}\\|_2^2$ is sufficiently large — that is, when the norm of the universal signal $\\|\mathbf{u}\\|_2$ is much smaller than that of the task-specific signal $\\|\mathbf{v_2}\\|_2$ (indicating that the shared knowledge is very limited) — transferring the upstream model parameters to the downstream model will result in negative transfer. We have modified this part in Proposition 4.4 to make clear. See **Line 278-279 and Line 286-289.**
>
>
>
>
> **Q2**: Generally, transfer is not random but based on the first $m$ layers or $n-1$ layers, where $n$ is the total number of layers. How does the current work take that into account?
>
> **A2**: Since we use a simplified two layer neural network model for analytical tractability, it is also feasible to consider transferring only a single layer of parameters. In fact, transferring just one layer yields exactly the same results to the case $\alpha = 1$. We focus on $\alpha$ proportion random sampling here because transferring a single layer seems less general than randomly sampling any $\alpha$ fraction of the parameters.
>
> We would like to emphasize that our central assumption is the existence of shared structure between the upstream and downstream tasks. Our analysis shows that both the shared structure and the noise present in the upstream task can influence whether transfer is positive or negative. We adopt a simplified two-layer neural network model because it enables a rigorous study of how the learning dynamics in the downstream task are affected by shared features and noise, while still offering insights for deeper and more complex architectures. Within this framework, the $\alpha$-proportion random parameter sampling assumption serves as **a mathematically clean way for analysis tractability**.
>
> With respect to more realistic constraints such as layer-wise exposure, a full theoretical treatment of multi-layer networks remains mathematically challenging: interactions across layers create highly coupled and unstable dynamics. Nevertheless, the mechanisms identified in the two-layer case can still shed light on deeper models. When the upstream task has high quality data with strong signal and large sample size, deeper models are expected to preserve and propagate this useful structure across layers. Our real experiments results also support this conclusion aligning with our theoretical analysis. More broadly, this example reflects the general philosophy of feature-learning theory. Theoretical models in this area deliberately use simplified architectures to capture **common phenomena** in representation learning, rather than to reproduce every implementation detail of practical systems. Despite their simplicity, such models have repeatedly demonstrated that early learned features and noise can strongly affect downstream performance, even in deep networks with high capacity and nonlinear expressiveness. This explains why analyses based on shallow models remain informative: they isolate fundamental principles of signal learning, noise memorization, and transfer quality that extend to more complex architectures, although in forms that are increasingly difficult to characterize with full mathematical rigor. We have added a discussion to **Line 2407-2418 in Section H**.

---

> ### Author Response · Authors · 2025-11-18
>
> **Q3**: Can a connection to [1] and [2] be established?
>
> **A3**: The formal definition of negative transfer given in [1] is based on the empirical performance on the target data, i.e., using source data leads to worse performance than training only on target data. Reference [1] attributes negative transfer to the large discrepancy between the source and target domains. In contrast, our work analyzes the causes of negative transfer from a theoretical perspective. We also provide conditions for negative transfer from the viewpoint of data rather than empirical results. Our paper thus offers a deeper theoretical explanation behind the phenomenon described in [1].
>
> Reference [2] introduces the notion of the degree of alignment and investigates its relationship with transfer learning performance. It argues that neural networks automatically adjust their representations during training so that the top singular vectors align with the task labels, which is validated by the experiments. Insteadly, our work provides a theoretical explanation for the underlying dynamics. We obtain some similar findings with proof: a neural network memorizes both signal and noise during training, as shown in Lemma A.1 and Lemma A.2.  The transferred parameters therefore retain the shared signal between the two tasks. When the norm of the shared signal becomes too small, negative transfer emerges. Our work theoretically characterizes and explains this dynamical process, while [2] proposes the conjectures and then verifies it from an empirical perspective. The core viewpoints are different but share conceptual similarities.
>
> Reference [1] is cited in **Line 049**. References [2] is cited in **Line 2379**.
>
> **S1**: Have a uniform citation format. Sometimes the page number is included for NeurIPS; sometimes it’s not. Sometimes conference names are abbreviated, and sometimes written in full.
>
> **AS1**: Thank you very much for your thorough review. We have re-examined and revised all the references accordingly again.
>
> **Q4**: Is this theory only valid for classification, or can it be generalized to regression and other tasks?
>
> **A4**: Although we have not provided a rigorous proof, our intuition suggests that this process can be extended from classification to regression. As for other tasks such as segmentation, detection, and image generation, this remains a highly meaningful open question.

---

> > ### Comment · Reviewer_ymzu · 2025-11-20
> > **Reply for related work**
> >
> > Dear authors, thank you for addressing my concern regarding related work. In my opinion, authors justify a connection with [2] appropriately in the reply but do not cite in detail in the main text. as the maintext just reads "Imani et al. (2021) introduces the notion of the degree of alignment and investigates its relationship
> > with transfer learning performance.", while citing this work is good, i would reccomend to cite it and distinguish it from your work as well in the manuscript.  I think it would make sense if authors justify the relationship in the manuscript similarly as they did in reply, considering it's being cited in the appendix, there is no issue of page limit as well.

---

> > > ### Author Response · Authors · 2025-11-20
> > >
> > > Dear Reviewer,
> > >
> > > Thank you very much for your constructive feedback.
> > > In the latest revised manuscript, we have incorporated a more complete discussion of [2] in Section G, following the detailed explanation we provided in our previous response. Additionally, we cite this work in the first sentence of the Introduction.
> > >
> > > We sincerely hope that these revisions address your concern. If you find the changes satisfactory, please reconsider whether the manuscript now meets the criteria of ICLR.

---

> > > > ### Comment · Reviewer_ymzu · 2025-11-23
> > > >
> > > > Thank you
> > > >
> > > > I have increased my score to 6.

---

### Official Review · Reviewer_ZDxq · 2025-11-01

**Soundness:** 3
**Presentation:** 3
**Contribution:** 3
**Rating:** 8
**Confidence:** 3

**Summary:**

This paper provides a theoretical analysis of feature learning in parameter transfer, a core mechanism in transfer learning where parts of a pretrained model are reused for downstream tasks. Focusing on ReLU convolutional neural networks, the authors study when and why partial parameter reuse is beneficial. Their framework reveals how transferred parameters encode universal knowledge that can enhance learning efficiency under certain conditions, while also explaining cases where parameter transfer may hurt downstream performance compared to training from scratch. Theoretical insights are supported by both numerical simulations and real-world experiments, offering a clearer understanding of the mechanisms governing successful parameter transfer.

**Strengths:**

The theoretical derivations appear rigorous, and the topic—providing a theoretical explanation for transfer learning—is both timely and important. The use of feature learning analysis to study parameter transfer offers a novel and interesting perspective.

**Weaknesses:**

Some assumptions are a bit restrictive, such as assuming $u$ is orthogonal to both $v_1$ and $v_2$. In addition, certain terminologies, including "Bayesian optimal" and "sub-Bayesian optimal" as used in Theorems 4.2 and 4.3, require clearer definitions or explanations to ensure accessibility for a broader audience.

**Questions:**

1. In Definitions 3.1 and 3.2, the covariance matrices of the noise are given as $\sigma_{p,1}(I - uu^\top / \|u\|^2 - v_1v_1^\top / \|v_1\|^2)$ and $\sigma_{p,2}(I - uu^\top / \|u\|^2 - v_2v_2^\top / \|v_2\|^2)$, respectively. However, the last paragraph of Page 3 states that "the noise variances in Task 1 and Task 2 are $\sigma_{p,1}$ and $\sigma_{p,2}$." These descriptions seem inconsistent—please clarify the relationship between them.
2. Still in Definitions 3.1 and 3.2, the covariance matrices imply that different elements of the one noise vector may be correlated, with correlations depending on $u$ and $v$. This modeling choice requires justification. Moreover, it should be verified (or stated) that the covariance matrices are always positive semidefinite, as required for valid covariance definitions.
3. When $d = 1$, it is impossible to have $u \perp v_1$ and $u \perp v_2$. It should therefore be noted somewhere that $d \geq 2$ is assumed throughout the analysis.

---

> ### Author Response · Authors · 2025-11-18
>
> **W1**: Some assumptions are a bit restrictive, such as assuming $u$  is orthogonal to both and $v1, v2$.
>
> **A1**: The orthogonal assumptions are just aimed to simplify our theoretical analysis and can be extended to more general cases where the noise may have a non-trivial correlation with the signal part. We have pointed out this in the following paragraph of Definition 3.2 in Section 3. Importantly, these assumptions **do not change the underlying mechanism** we study. Even without exact orthogonality, the inner products between the signal and noise vectors will concentrate near zero, which has negligible impact on our analysis despite some complex
>  but not difficult technical steps.
>
>
> **W2**: In addition, certain terminologies, including "Bayesian optimal" and "sub-Bayesian optimal" as used in Theorems 4.2 and 4.3, require clearer definitions or explanations to ensure accessibility for a broader audience.
>
> **A2**: Thank you for mentioning the vague definition of "Bayesian optimal" and "sub-Bayesian optimal". To avoid any confusion, we have revised this by changing "Bayesian optimal" into "is close to the optimum", and "sub-Bayesian optimal" into "has a gap from the optimum". See Theorem 4.2.
>
> **Q1**: About the covariance matrices of the noise.
>
> **A1**: Here, $I - \mathbf{u}\mathbf{u}^{\top}/\\|\mathbf{u}\\|^{2} - \mathbf{v_1}\mathbf{v_1}^{\top}/\\|\mathbf{v_1}\\|^{2}$ and $I - \mathbf{u}\mathbf{u}^{\top}/\\|\mathbf{u}\\|^{2} - \mathbf{v_2}\mathbf{v_2}^{\top}/\\|\mathbf{v_2}\\|^{2}$
> are designed so that the sampled noise is orthogonal to the signal.  The sampled noise degenerates along the two directions $\mathbf{u}$ and $\mathbf{v_1}$ (or $\mathbf{v_2}$).
> Therefore, although the resulting distribution is not a standard normal distribution, the noise variances in Task 1 and Task 2 remain $\sigma_{p,1}$ and $\sigma_{p,2}$, respectively.
>
> **Q2**: Still in Definitions 3.1 and 3.2, the covariance matrices imply that different elements of the one noise vector may be correlated, with correlations depending on $u$ and $v$. This modeling choice requires justification. Moreover, it should be verified (or stated) that the covariance matrices are always positive semidefinite, as required for valid covariance definitions.
>
> **A2**: In Definitions 3.1 and 3.2, the noise covariance matrices are given by
> $\Sigma_{1} = \sigma_{p,1}(I - \mathbf{u}\mathbf{u}^{\top}/\\|\mathbf{u}\\|^{2} -\mathbf{v_1}\mathbf{v_1}^{\top}/\\|\mathbf{v_1}\\|^{2})$ and
> $\Sigma_{2} = \sigma_{p,2}(I - \mathbf{u}\mathbf{u}^{\top}/\\|\mathbf{u}\\|^{2} - \mathbf{v_2}\mathbf{v_2}^{\top}/\\|\mathbf{v_2}\\|^{2})$,
> where $\mathbf{u}, \mathbf{v_1}, \mathbf{v_2}$ are mutually orthogonal. Each term $\mathbf{u}\mathbf{u}^{\top}/\\|\mathbf{u}\\|^{2}$ and $\mathbf{v_1}\mathbf{v_1}^{\top}/\\|\mathbf{v_1}\\|^{2}$ is the orthogonal projector onto the corresponding direction, so the matrices in parentheses are orthogonal projectors onto the subspaces orthogonal to $\mathrm{span}${$\mathbf{u}, \mathbf{v_1}$} and $\mathrm{span}${$\mathbf{u}, \mathbf{v_2}$}, respectively. Orthogonal projectors are idempotent (see 0.9.13 in [1]) and then each eigenvalue of the idempotent matrix is either 0 or 1 (see 1.1.P5 in [1]). So orthogonal projectors are symmetric positive semidefinite. Hence both covariance matrices are always positive semidefinite.
>
> [1] Horn, Roger A., and Charles R. Johnson. Matrix analysis. Cambridge university press, 2012.
>
> **Q3**: When $d = 1$, it is impossible to have $\mathbf{u} \perp \mathbf{v_1}$ and $\mathbf{u} \perp \mathbf{v_2}$. It should therefore be noted somewhere that $d \geq 2$ is assumed throughout the analysis.
>
> **A3**: Yes. The assumption that $\mathbf{u}$ is orthogonal to $\mathbf{v_1}$ and $\mathbf{v_2}$, which implies that $d \geq 2$.

---

### Official Review · Reviewer_SXqG · 2025-11-05

**Soundness:** 3
**Presentation:** 3
**Contribution:** 2
**Rating:** 6
**Confidence:** 3

**Summary:**

The paper investigates the role of parameters shared between upstream and downstream models. Inheriting parameters can serve as a carrier of general knowledge and is sometimes beneficial for the target task; however, in certain cases, transferring parameters leads to lower accuracy on the target task than training from scratch.

**Strengths:**

1.The paper is highly readable.
2.The theoretical derivations are complete, lending strong credibility.
3.It substantiates a key conclusion: inheriting more parameters, using larger upstream training datasets, and having less noise in upstream tasks can improve downstream model performance. The conclusions in the contributions section are insightful.

**Weaknesses:**

Please refer to the Questions section below.

**Questions:**

1.The case would be stronger with experiments on ViT-based models or VLMs.
2.There is a lack of cross-dataset experiments across different tasks.
3.The paper does not provide sufficient comparisons with existing parameter-transfer or transfer-learning methods.

---

> ### Author Response · Authors · 2025-11-18
>
> **Q1**: The case would be stronger with experiments on ViT-based models or vision-language models (VLMs).
>
> **A1**: We did experiments on ViT-based models, which is shown in **Experiments on Vision Transformers** part in Section 7.
>
> **Q2**: There is a lack of cross-dataset experiments across different tasks.
>
> **A2**: We conduct experiments where the upstream task is image segmentation and the downstream task is image classification. The results are added to Section F. For the upstream segmentation task, we use two models, deeplabv3_resnet50 and deeplabv3_mobilenet_v3_large, whose backbones are resnet50 and mobilenet_v3_large, respectively. For the downstream classification task, we use resnet50, resnet34, and mobilenet_v3_large. The upstream models are pretrained by PyTorch on a subset of the COCO dataset, while the downstream models are trained on CIFAR10 and CIFAR100.
>
> The experimental results are summarized in the table below. Here, **w/o PL** denotes downstream training without parameter transfer, and **w/ PL** denotes downstream training with parameter transfer. The results show that cross-task parameter transfer can also be beneficial, indicating the presence of shared knowledge across different tasks.
>
> | Dataset   | Upstream Model                 | Downstream Model        | Acc (w/o PL) | Acc (w/ PL) |
> |-----------|--------------------------------|--------------------------|--------------|--------------|
> | CIFAR10   | deeplabv3_resnet50             | resnet50                | 89.25        | 94.67        |
> | CIFAR10   | deeplabv3_resnet50             | resnet34                | 90.8         | 93.39        |
> | CIFAR10   | deeplabv3_mobilenet_v3_large   | mobilenet_v3_large      | 83.06        | 89.45        |
> | CIFAR100  | deeplabv3_resnet50             | resnet50                | 70.45        | 75.31        |
> | CIFAR100  | deeplabv3_resnet50             | resnet34                | 68.35        | 73.96        |
> | CIFAR100  | deeplabv3_mobilenet_v3_large   | mobilenet_v3_large      | 66.64        | 72.24        |
>
>
>
> **Q3**: The paper does not provide sufficient comparisons with existing parameter-transfer or transfer-learning methods.
>
> **A3**: Our goal is not to propose a new state-of-the-art parameter-transfer method. Instead, we aim to provide a reasonable explanation for the underlying working mechanism of existing parameter-transfer approaches. So we use a simple parameter-transfer algorithm. The experiments are conducted to validate the soundness of the theoretical framework we propose. We add the discussion about connections between theory and practice in **Section H** in the appendix.

---

### Author Response · Authors · 2025-11-28

Dear Area Chair,

Thanks for rolling back the scores to keep the process fair.

We just want to clarify one thing: the two positive score updates—Reviewer 6DBg 2→4 and Reviewer ymzu 4→6—happened before the bug became a widely publicized issue. We didn’t know the bug existed and simply answered their questions.

We would be grateful if you could take the evolution of the discussion into account. The upward revision of the scores came only after a constructive dialogue, and we feel these later scores are a truer reflection of our paper's contribution.

Thank you for your valuable time.

Best,

The Authors

---

### Note · Authors · 2026-01-26

I have read and agree with the venue's withdrawal policy on behalf of myself and my co-authors.

---

### Meta-Review · Area_Chair_Kcwq · 2026-01-06

**Summary:**

This paper provides a theoretical analysis on the impact of partial parameter transfer on the performance of transfer learning with convolutional neural networks, and particularly characterizes the impact of three factors, including the normal of shared knowledge, the sample size of the source task, and the noise level in the source data.

Strengths:
- The theoretical analysis is solid.

- Characterize the training dynamics for partial parameter transfer in transfer learning.

Weaknesses:

- Some assumptions are restrictive.

- The insights provided by this paper are not surprising and whether they can provide meaningful guidance in practice is not clear.

- Portions of the empirical evaluation do not adequately reflect the theoretical analysis.

**Reviewer Concerns:**

The rebuttal mainly addressed some clarification issues from the reviewers, including the clarification of modeling and citations. While experimental results were provided for cross-dataset experiments, there experiments purely demonstrated the benefit of transfer learning but couldn't reflect any insights obtained from this paper. This problem also existed in section 7 about the experiments on Vision Transformers.

Besides, after reading the paper, while the theoretical analysis is sound, the insights on the impact of the three key factors on the learning performance are indeed intuitive. Although the authors claim that this is the first analysis to show negative transfer, their findings from the theory are not surprising, i.e., negative transfer can happen when the source and the target do not share enough common knowledge. Therefore, it is not clear what meaningful guidance these findings can provide for transfer learning in practice.

The assumption about random parameter selection is made for analytical convenience but not practical. While this may not cause much difference from layer-wise parameter transfer on the simple 2-layer CNN considered in this paper, such a statement can not be justified for more complex models.

**Reviewer Scores:**

Reviewers SXqG and ZDxq may maintain their score as they were positive initially and didn't raise any critical concerns that could change the score.

Reviewer ymzu may increase the score to 6 as their criticism was mainly about presentation issues and addressed after rebuttal.

Reviewer 6DBg may increase the score to 4 but still stay negative due to the weaknesses about assumptions and practical guidance.

---

### Decision · Program_Chairs · 2026-01-26

Reject